# Reconfiguring nucleation for CVD growth of twisted bilayer MoS$_2$ with a wide range of twist angles

Manzhang Xu [1,2,3,10], Hongjia Ji[1,10], Lu Zheng[1,2,3], Weiwei Li[1,2,3], Jing Wang[4], Hanxin Wang [1], Lei Luo[1], Qianbo Lu [1,2,3], Xuetao Gan [4], Zheng Liu [5,6,7], Xuewen Wang [1,2,3] ✉ & Wei Huang [1,2,3,8,9] ✉

Twisted bilayer (TB) transition metal dichalcogenides (TMDCs) beyond TB-graphene are considered an ideal platform for investigating condensed matter physics, due to the moiré superlattices-related peculiar band structures and distinct electronic properties. The growth of large-area and high-quality TB-TMDCs with wide twist angles would be significant for exploring twist angle-dependent physics and applications, but remains challenging to implement. Here, we propose a reconfiguring nucleation chemical vapor deposition (CVD) strategy for directly synthesizing TB-MoS$_2$ with twist angles from 0° to 120°. The twist angles-dependent Moiré periodicity can be clearly observed, and the interlayer coupling shows a strong relationship to the twist angles. Moreover, the yield of TB-MoS$_2$ in bilayer MoS$_2$ and density of TB-MoS$_2$ are significantly improved to 17.2% and 28.9 pieces/mm$^2$ by tailoring gas flow rate and molar ratio of NaCl to MoO$_3$. The proposed reconfiguring nucleation approach opens an avenue for the precise growth of TB-TMDCs for both fundamental research and practical applications.

In recent years, twisted bilayer (TB) graphene, especially magic-angle graphene, has become a fascinating material for advancing the field of twistronics. They provide an ideal platform for the investigation of superconductivity, Mott insulation state, quantum anomalous Hall effect, and quantum critical point[1–4]. The exciting properties of TB-graphene are naturally expected to extend to TB-transition metal dichalcogenides (TB-TMDCs)[5,6]. Similar to TB-graphene, the periodic

Moiré patterns can be formed in TB-TMDCs, which results in the regulation of electronic band structure and further producing a series of interesting physical phenomenon[7–11]. However, these TB-TMDC materials have neither been naturally formed in bulk materials nor synthesized via traditional growth methods due to their unfavorable manner in thermodynamic processes. Therefore, to develop a distinctive approach for the direct synthesis of high-quality and large-area

[1]Frontiers Science Center for Flexible Electronics (FSCFE) & Institute of Flexible Electronics (IFE), Northwestern Polytechnical University, Xi'an 710072, P. R. China. [2]MIIT Key Laboratory of Flexible Electronics (KLoFE), Northwestern Polytechnical University, Xi'an 710072, P. R. China. [3]Shaanxi Key Laboratory of Flexible Electronics (KLoFE), Northwestern Polytechnical University, Xi'an 710072, P. R. China. [4]Key Laboratory of Light Field Manipulation and Information Acquisition, Ministry of Industry and Information Technology, and Shaanxi Key Laboratory of Optical Information Technology, School of Physical Science and Technology, Northwestern Polytechnical University, Xi'an 710129, P. R. China. [5]School of Materials Science and Engineering, Nanyang Technological University, 50 Nanyang Avenue, Singapore 639798, Singapore. [6]CINTRA CNRS/NTU/THALES, UMI 3288, Research Techno Plaza, 50 Nanyang Drive, Border X Block, Level 6, Singapore 637553, Singapore. [7]School of Electrical and Electronic Engineering, Nanyang Technological University, Singapore 639798, Singapore. [8]State Key Laboratory of Organic Electronics and Information Displays, Institute of Advanced Materials (IAM), Nanjing University of Posts & Telecommunications, Nanjing 210023, P. R. China. [9]Key Laboratory of Flexible Electronics (KLoFE) and Institute of Advanced Materials (IAM), Nanjing Tech University (NanjingTech), Nanjing 211800, P. R. China. [10]These authors contributed equally: Manzhang Xu, Hongjia Ji. ✉e-mail: iamxwwang@nwpu.edu.cn; vc@nwpu.edu.cn

TB-TMDCs with a wide range of twist angles from 0° to 120° would be significantly important to investigate their intriguing physical properties and potential applications.

In order to accomplish this, two-step stacking and folding methods have been proposed for artificially controlling the twist angles. However, the polymer residues, time-consuming, low success rate, as well as ultra-low yield result in unavoidable interlayer contamination, which negatively impacts the interlayer coupling[12,13]. On the contrary, one-step chemical vapor deposition (CVD) method offers an approach to directly grow clean TB-TMDCs. Thus, direct growth of TB-TMDCs via CVD is currently regarded as a highly promising technique, owing to its excellent quality and scalability[14,15]. The TB-TMDCs always show non-twisted structure (0°-TB-TMDCs or 60°-TB-TMDCs) due to the energetically favorable bilayer in CVD process[12,16]. It is well known that the rotations between monolayer and bilayer in TB-TMDCs need to overcome a high energy barrier. Therefore, there are fewer reports for the CVD synthesis of TB-TMDCs compared with bilayer TMDCs[17,18]. Sever approaches have been adopted for the synthesis of TB-TMDCs, such as controlling nucleation rate, temperature enhancement, as well as introduction of catalyst. Specifically, Liu et al.[16] reduced the nucleation rate at the initial stage, which can obtain a high proportion of bilayer MoS$_2$, and acquire break trough of high yield of TB-MoS$_2$ in bilayer MoS$_2$ (~5%). Han et al.[19] introduce the Mo foil instead of MoO$_3$, which can reduce the nucleation rate and avoid introducing impurities. Shao et al.[20] introduced the tin (Sn) into the CVD system for reducing the stacking energy of WS$_2$. A similar approach has also been adopted for the synthesis of twisted WSe$_2$/WSe$_2$ homostructure[21]. Zheng et al.[22] enhanced the reaction temperature to 1100 °C to overcome the angle mismatch in bilayer WS$_2$, while only non-twisted structure can be observed under a reaction temperature of 850 °C. However, it is still challenging to prepare large-area and clean TB-TMDCs with a wide range of twist angles by CVD method especially with small twist angles. Therefore, an efficient approach for growing TB-TMDCs, especially with good repeatability, high yield, and diverse twist angles is urgently required. In addition, establishing the correlation between the synthesized results and CVD parameters is still highly desirable but fraught with challenges.

Here, we propose a reconfiguring nucleation strategy for the direct synthesis of clean TB-TMDCs with thermodynamically unfavorable twisted stacking structures. As one example, the synthesized TB-MoS$_2$ has a wide range of twist angles from 0° to 120°. The exceptional crystalline quality of the grown TB-MoS$_2$ is verified by the atomic-resolution lateral force microscopy (LFM) and clear Moiré patterns in scanning transmission electron microscopy (STEM). The Raman and photoluminescence (PL) spectra show a strong relationship between interlayer coupling and the twist angles. In addition, the key parameters for CVD growing of TB-MoS$_2$ have been proposed. This synthesis strategy has been extended for preparing high-quality TB-WS$_2$, sheding light on the precisely controllable growth of other 2D TMDCs with controllable interlayer twists.

## Results
### Synthesis of TB-MoS$_2$
A space-confined CVD method was developed to grow TB-MoS$_2$, as illustrated in Fig. 1a (see Methods for details). The MoO$_3$ and sulfur powder were adopted as Mo and S source, respectively. With the assistance of NaCl, TB-MoS$_2$ is grown on SiO$_2$/Si substrate. As demonstrated in Fig. 1b, the twist angle is defined by the angular difference along the corresponding edges of the top and bottom layers (indicated by dashed green and red lines, respectively). The twist angles are determined from the bright field (BF) optical microscopy (OM) images with an error of about 0.8° (Supplementary Note 1 and Supplementary Fig. 1). The polarization-resolved second harmonic generations (SHG) is a precise technique to determine the orientation of 2D materials. To evaluate the accuracy of twist angle from BF-OM,

the SHG has been recorded in the location of monolayer MoS$_2$ and TB-MoS$_2$ (Fig. 1c–f). Monolayer MoS$_2$ exhibits a three-fold rotational symmetry in its crystalline structure, while the SHG response with a six-fold rotational symmetry with six petals when the polarization of the incident laser is parallel (perpendicular) to the polarization of the SHG signal[23]. The polarization dependence of the SHG intensities in TB-MoS$_2$ with respect parallel components is shown in Fig. 1e. The 0°-TB-MoS$_2$ shows the superposed SHG fields with consistent SHG orientation excepted for the SHG intensity. In the 95.3°-TB-MoS$_2$ (Fig. 1f), the SHG orientation of monolayer MoS$_2$ is consistent with the polarization direction of 0°-TB-MoS$_2$. The polarization direction of each layer is determined by the angle between the respective armchair direction and the pump laser polarization of each layer. The phase difference of SHG in two layers of SHG is determined by the twist angle. Therefore, six-fold rotational symmetry of SHG remains in the twisted area, and the azimuthal angle of SHG is half of the twist angle. The difference of azimuthal angle from SHG is calculated to be 47.7°, which is roughly half of the twist angle for 95.3°-TB-MoS$_2$. More TB-MoS$_2$ with twist angles of 9.2°, 21.0°, 32.7°, 74.9°, 81.0°, and 84.7° show the SHG azimuthal angle of 4.5°, 10.5°, 16.3, 37.3°, 40.3°, and 42.5, respectively (Supplementary Figs. 2 and 3). The SHG results indicate that the measured twist angles from the optical images are accurate and reliable. Due to the convenient operation, the OM was adopted to measure the twist angles of TB-MoS$_2$, based on the sharp edges of the MoS$_2$ crystals. Figure 1g clearly demonstrates that TB-MoS$_2$ with a wide range of twist angles from 0 to 120° are successfully grown by the proposed space-confined CVD method. In addition, the dark-field (DF) OM images confirm the clean surface in the TB-MoS$_2$ (Supplementary Fig. 4).

### Characterization of TB-MoS$_2$
The area of monolayer and bilayer in TB-MoS$_2$ can be clearly identified by the BF-OM, DF-OM, atomic force microscope (AFM), and scanning electron microscope (SEM) images. As noticed in Supplementary Fig. 5, the in situ BF-OM, DF-OM, SEM, and AFM images of a typical TB-MoS$_2$ with a twist angle of 21.9° indicate that clean and homogeneous TB-MoS$_2$ is successfully synthesized. The typical Raman and PL spectra were taken from the 101.3°-TB-MoS$_2$, while the single spectrum of monolayer and bilayer are recorded from the location in the dotted area of TB-MoS$_2$ (Supplementary Fig. 6a). The Raman peaks of monolayer MoS$_2$ are located around 385.8 and 401.6 cm$^{-1}$ (Fig. 2a), which is corresponding to $E^1_{2g}$ mode (the in-plane vibration of Mo and S atoms) and $A_{1g}$ mode (out-of-plane vibration of the S atom), respectively[16,19,24]. Compared to monolayer MoS$_2$, TB-MoS$_2$ shows a higher Raman intensity attributed to the strongly increased Raman intensity resulting from the increase in layer number[25–29]. Besides, the frequency of the $E^1_{2g}$ mode decreases ($\Delta\omega = 2.4$ cm$^{-1}$), while the frequency of the $A_{1g}$ mode increases ($\Delta\omega = 2.1$ cm$^{-1}$). This is caused by the long-range Coulombic interaction between the Mo atoms. In addition, the enhancement of interlayer interactions leads to restoring forces on the increased S atoms[30]. The Raman intensity mappings of $A_{1g}$ and $E^1_{2g}$ with a mapping size of 18 μm × 18 μm are shown in Fig. 2b and Supplementary Fig. 6b, respectively. The monolayer and bilayer MoS$_2$ can be distinguished easily from the Raman mapping due to the strong contrast. The uniform intensity observed throughout the mapping region indicates the crystalline uniformity of monolayer and bilayer in 101.3°-TB-MoS$_2$ sample. In addition, clear contrast is observed in the monolayer and bilayer area in the Raman mapping for the position of maximum of 101.3°-TB-MoS$_2$ in the range around $E^1_{2g}$ and $A_{1g}$, which is attributed to the large shift for the $A_{1g}$ and $E^1_{2g}$ vibration frequency (Fig. 2c, d). Compared to the peak locations for monolayer MoS$_2$, a red shift of $E^1_{2g}$ model (2.3 cm$^{-1}$) and a blue shift of $A_{1g}$ (2.0 cm$^{-1}$) are observed for bilayer MoS$_2$, which is consistent with the Raman spectrum in the literature[31–33]. Moreover, the bilayer MoS$_2$ shows a more homogeneous Raman shift across the whole mapping region, indicating a high consistency of vibration mode in 101°-TB-MoS$_2$.

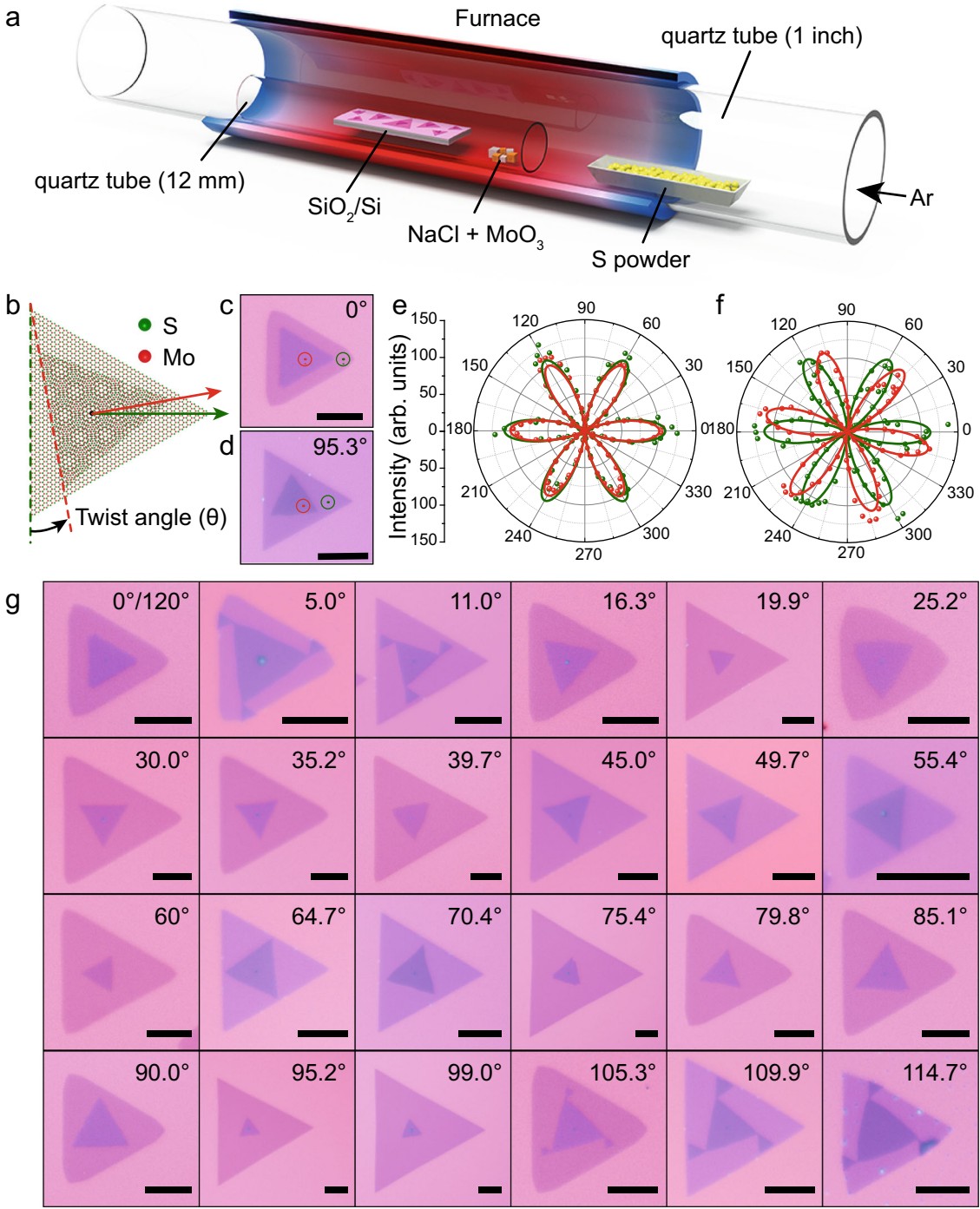

**Fig. 1 | Growth of twisted bilayer MoS₂ (TB-MoS₂). a** Illustration of the chemical vapor deposition (CVD) setup for the synthesis of TB-MoS₂. **b** Schematic atomic configuration of TB-MoS₂ with a twist angle due to a relative lattice alignment between two layers. **c, d** Typical bright field optical microscopy (BF-OM) of 0° and 95.3°-TB-MoS₂, the green and red circles indicate the top and bottom layers, respectively. **e, f** Polar plots of parallel components of second harmonic generations (SHG) intensity of 0°-TB-MoS₂ and 95.3°-TB-MoS₂. The raw data and corresponding fitted curves of the monolayer (greed) and TB-MoS₂ (red) are measured from the circles area in (**c**) and (**d**), respectively. The intensity has been normalized. **g** BF-OM images of the as-grown TB-MoS₂ with twist angles from 0 to 120°. Scale bars: 10 μm in (**c**), (**d**), and (**g**).

The PL spectra and PL mapping of the 101°-TB-MoS₂ are shown in Fig. 2e–g. In the monolayer MoS₂, two peaks are observed at energy levels of 1.81 and 1.95 eV, corresponding to A exciton and B exciton, respectively. For the bilayer MoS₂, a red shift of 0.4 eV is observed for A exciton, while the B exciton remains at 1.95 eV. Compared to monolayer MoS₂, the A exciton exhibits weaker PL intensity, while the B exciton has a consistent PL intensity in the PL intensity mappings of

bilayer MoS₂ (Fig. 2f, g). In addition, a noticeable emission shift is observed around A exciton between monolayer and bilayer, while no significant change is observed in the B exciton, as shown in Supplementary Fig. 6c, d. This is consistent with previous reports[24,34–36]. The blue shift of A exciton can be attributed to a decrease in the binding energy[37]. The reduced emission intensity of A exciton can be caused by the charge redistribution under interlayer interaction at the van der

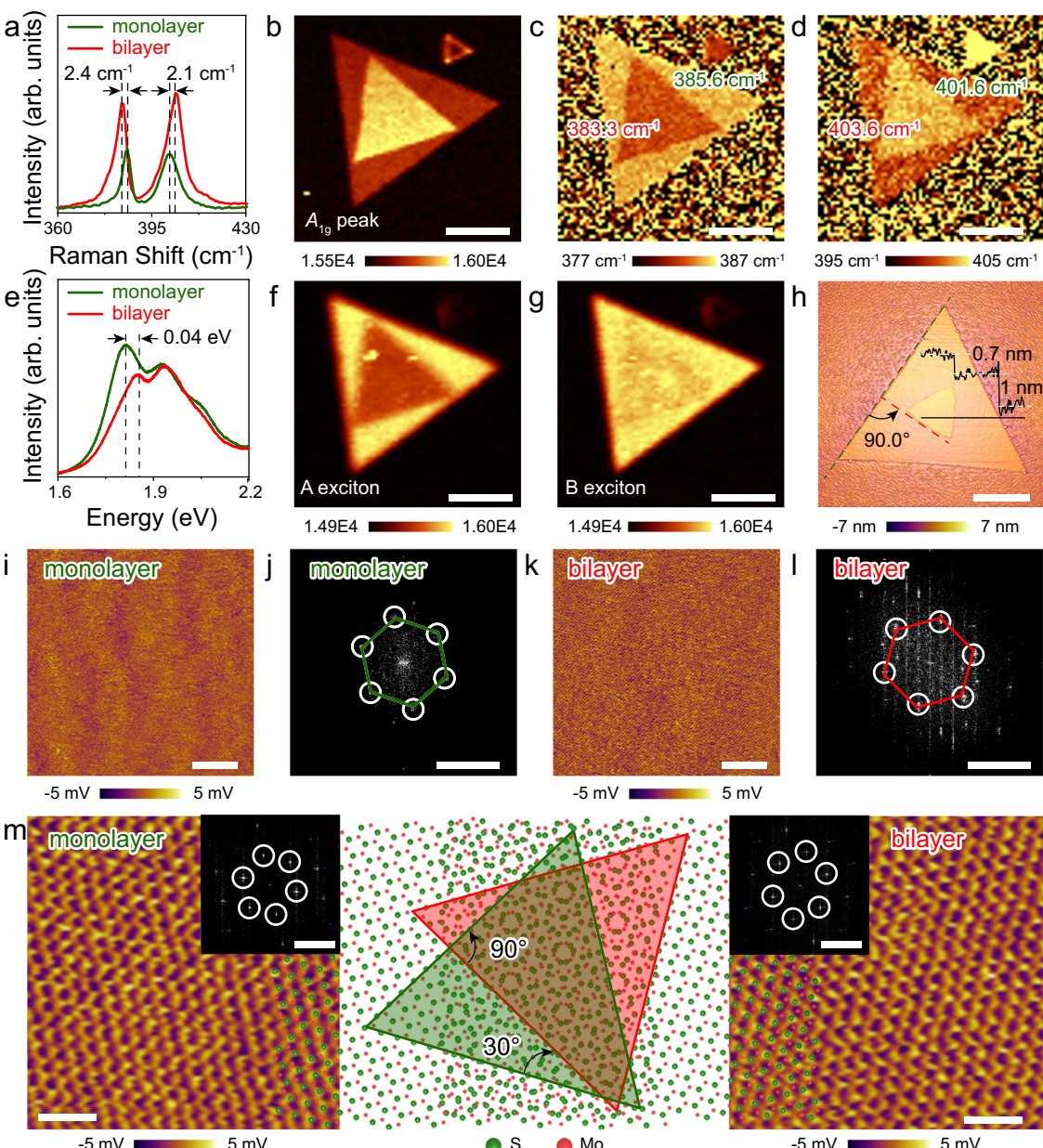

**Fig. 2 | Characterization of TB-MoS₂. a** Typical Raman spectra of monolayer and 101.3°-TB-MoS₂. The dashed lines represent the $E^1_{2g}$ and $A_{1g}$ peak positions of TB-MoS₂. **b** Raman intensity mapping for $A_{1g}$ mode of 101.3°-TB-MoS₂. **c, d** Raman mappings for position of maximum of 101.3°-TB-MoS₂ in the range around $E^1_{2g}$ and $A_{1g}$, respectively. **e** Typical photoluminescence (PL) spectra of monolayer and bilayer in 101.3°-TB-MoS₂, which is taken from the dots in panel (**d**). The dashed lines represent the A exciton position of TB-MoS₂. **f, g** PL intensity mapping for A exciton and B exciton of 101.3°-TB-MoS₂. **h** Atomic force microscope (AFM) image of typical 90.0°-TB-MoS₂. The dashed lines represent the orientation of the bottom (green) and top (red) layer, respectively. The height profile shows the thicknesses of monolayer and bilayer MoS₂. **i, j** High-resolution lateral force microscopy (LFM) raw image and corresponding fast Fourier transform pattern of the monolayer area in 90.0°-TB-MoS₂. **k, l** High-resolution LFM raw image and corresponding fast Fourier transform pattern of the bilayer area in 90.0°-TB-MoS₂. **m** Atomic-resolution LFM raw images of the monolayer and bilayer in 90.0°-TB-MoS₂ and corresponding schematic diagram of atomic lattice of 90.0°-TB-MoS₂. The insets illustrate the fast Flourier transform (FFT) patterns of atomic-resolution LFM raw images. Scale bars: 5 μm in (**b**)–(**d**) and (**f**)–(**h**), 5 nm in (**i**) and (**k**), 5 nm⁻¹ in (**j**) and (**l**), 1 nm in (**m**), and 5 nm⁻¹ in the inset of (**m**).

Waals interface in bilayer MoS₂. Moreover, the constant emission intensity of B exciton may be attributed to the similar hole concentration in the valence state of monolayer and bilayer MoS₂[38]. The Raman and PL results suggest the successful synthesis of high-quality TB-MoS₂.

The ability to visualize atomic lattices and moiré superlattices nondestructively is critical for the rapid development of 2D materials, and the scanning probe approach offers an attractive option for achieving this goal. The LFM, which measures nanoscale friction by using widely available AFM, proves particularly useful for

nondestructive imaging across various types of 2D materials under ambient conditions[39,40]. Figure 2h shows a typical AFM image of 90.0°-TB-MoS₂. The corresponding BF-OM and DF-OM images are shown in Supplementary Fig. 7. The thicknesses of monolayer and bilayer MoS₂ are 1.0 and 0.7 nm, respectively. The LFM raw images of monolayer and bilayer MoS₂ as well as the corresponding fast Flourier transform (FFT) images are shown in Fig. 2i–l. As demonstrated, the periodic lattice structures of monolayer and bilayer are clearly observed from the high-resolution LFM raw image (Fig. 2i, k). The hexagonal points are observed in the corresponding FFT images, as shown in Fig. 2j, l,

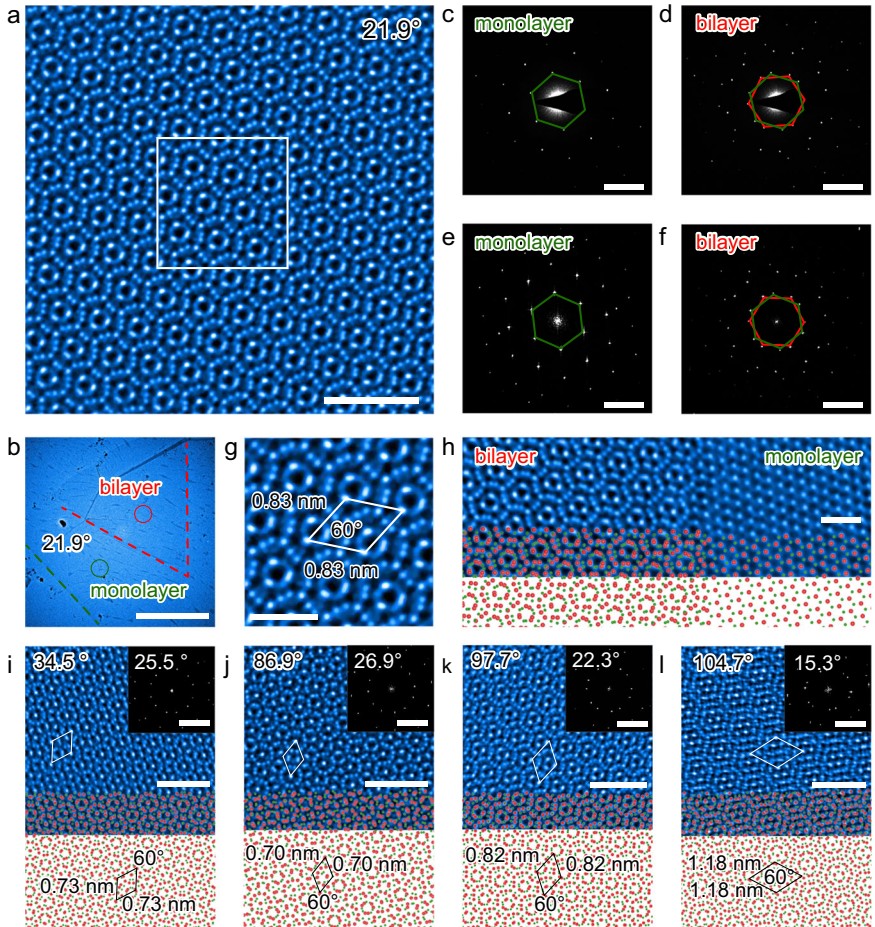

**Fig. 3 | Commensurate superlattices in TB-MoS₂. a** Atomic-resolution high-angle annular dark field (HAADF) scanning transmission electron microscopy (STEM) image of 21.9°-TB-MoS$_2$. **b** HAADF-STEM image of the 21.9°-TB-MoS$_2$. The dashed lines represent the edges of the bottom (green) and top (red) layer of MoS$_2$, respectively. The green and red circles indicate the monolayer and bilayer regions for the measurement of SETM and SAED, respectively. **c–f** SAED and fast Flourier transform (FFT) patterns of monolayer and bilayer in 21.9°-TB-MoS$_2$ in (**b**). The green and red hexagonal points represent the bottom and top layer rotation orientations, respectively. **g** Enlarged white box region in (**a**) showing the detailed atomic features in each Moiré pattern, and the white rhombus indicates a Moiré period of 0.83 nm. **h** Atomic-resolution HAADF-STEM image of the monolayer and bilayer MoS$_2$ at the interface of the 21.9°-TB-MoS$_2$ with the corresponding atom model. **i–l** Atomic-resolution HAADF-STEM images of a series of commensurate superlattices and corresponding atom models of 34.5°, 86.9°, 97.7°, and 104.7°-TB-MoS$_2$, respectively. The insets are the corresponding FFT patterns of TB-MoS$_2$ with measured twisted angles. Scale bars: 2 nm in (**a**), 5 μm in (**b**), 5 nm⁻¹ in (**c**)–(**f**), 1 nm in (**g**) and (**h**), 2 nm in (**i**)–(**l**), and 5 nm⁻¹ in the inset of (**i**)–(**l**).

demonstrating a hexagonal crystalline structure of monolayer and bilayer MoS$_2$. It is also seen that the FFT patterns have different angles, which is ascribed to the different lattice orientations in the monolayer and bilayer of the 90.0°-TB-MoS$_2$. The atomic lattice images of monolayer MoS$_2$ are less sharp compared to the bilayer MoS$_2$, which is caused by the relatively rough surface of the substrate. The accurate atomic-resolution LFM raw image of monolayer and bilayer TB-MoS$_2$ with an angle of 90.0° is shown in Fig. 2m. The FFTs of the monolayer and bilayer MoS$_2$ show a slight deformation due to the thermal drift, which is a typical challenge encountered for AFM system that operate in non-vacuum environments[40]. It should be noted that the Moiré periodicity is not directly observed in the LFM images and the corresponding FFT pattern of the bilayer area in TB-MoS$_2$, which may be attributed to nanoscale friction induced by the interaction between the AFM tip and surface S atoms. In contrast, the twist angle can be identified by generating two sets of hexagonal points from FFT analysis with the twist angle of 29.8° (Supplementary Fig. 8). This is consistent with AFM and OM results for 90.0°-TB-MoS$_2$. Importantly, atomic lattices of monolayer and bilayer MoS$_2$ match well with the LFM results (Fig. 2m), indicating a twist angle between them in TB-MoS$_2$, corroborating the measurement from FFT results and OM images.

## Commensurate superlattices in TB-MoS₂

The twist angles of synthesized TB-MoS$_2$ are further investigated using an aberration-corrected STEM-high-angle annular dark field (HAADF) with an advanced STEM corrector (ASCOR). By employing ASCOR corrector on FEI Themis Z, the atomically fine Moiré features in the TB-MoS$_2$ are recorded. The damage-free imaging condition and high spatial resolution ensure highly precise and accurate Moiré patterns. A representative HAADF-STEM image with Moiré features in TB-MoS$_2$ is shown in Fig. 3a, while its corresponding HAADF-STEM image can be seen in Fig. 3b. The selective area electron diffraction (SAED) is further adopted to characterize the TB-MoS$_2$. As illustrated in Fig. 3c, the SAED of monolayer MoS$_2$ exhibiting hexagonal points, which reflects the single crystalline structure of the monolayer MoS$_2$. The representative SAED pattern of the bilayer MoS$_2$ displays two sets of hexagonal points with a relative rotation (Fig. 3d), confirming the twist angle between the monolayer and bilayer MoS$_2$. The SAED patterns exhibit 6-fold symmetry. In monolayer MoS$_2$, the intensity of $I_{\{110\}}$ is lower than $I_{\{100\}}$, while the opposite intensity can be founded in the TB-MoS$_2$ (Supplementary Fig. 9) due to the relatively weak interactions between the monolayer and bilayer. The FFT patterns of monolayer and bilayer TB-MoS$_2$ are shown in Fig. 3e, f. The FFTs show a consistent orientation

compared with the SAED. The twist angle can be determined from the FFT and SAED patterns and determined to be precisely 21.9° (Supplementary Note 2 and Supplementary Figs. 10 and 11), confirming a 21.9° misorientation between the monolayer and bilayer (Fig. 3d, f). Based on the real-space STEM (Fig. 3g), the Moiré superlattice of the TB-MoS$_2$ can be clearly observed and measured to be 0.83 nm. As noticed, each Moiré superlattice repetition with a 0.83 nm in both $x$ and $y$ directions suggests the presence of rotational symmetry in a commensurate 21.9°-TB-MoS$_2$[41,42]. The STEM image of the interface between the monolayer and bilayer is utilized to study the termination interfaces of the TB-MoS$_2$. As shown in Fig. 3h, the observed Moiré patterns is a direct evidence of epitaxial growth. Note that the periodic Moiré patterns are also noticed at the interface, indicating that the bilayer MoS$_2$ is epitaxially grown on the monolayer MoS$_2$.

More HAADF-STEM and SAED characterizations are performed on the TB-MoS$_2$ with several other twist angles (Fig. 3i–l and Supplementary Fig. 12). It is found that twist angles are measured to be 34.5°, 86.9°, 97.7°, and 104.7°from the HAADF-STEM images. The SAED patterns of 34.5°, 86.9°, 97.7°, and 104.7°-TB-MoS$_2$ show that two groups of hexagonal arrays of diffraction spots are observed with angles of 25.5°, 26.9°, 22.3°, and 15.3°, respectively (Supplementary Fig. 12). Note that the twist angles obtained from the HAADF-STEM image and SAED pattern have a certain relationship due to the 3-fold rotation symmetry of the monolayer MoS$_2$ lattice, as follows:

$$\theta_{SAED} = \theta_{STEM}(0 < \theta_{STEM} \leq 30°) \tag{1}$$

$$\theta_{SAED} = 60° - \theta_{STEM}(30° < \theta_{STEM} \leq 60°) \tag{2}$$

$$\theta_{SAED} = \theta_{STEM} - 60°(60° < \theta_{STEM} \leq 90°) \tag{3}$$

$$\theta_{SAED} = 120° - \theta_{STEM}(90° < \theta_{STEM} \leq 120°) \tag{4}$$

Here, the $\theta_{SAED}$ represents the twist angle between two adjacent diffraction spots in the SAED, while the $\theta_{STEM}$ represents the twist angle obtained from the HAADF-STEM image (same with the OM). Therefore, the calculated $\theta_{SAED}$ from the SAED are 25.5°, 26.9°, 22.3°, and 15.3°, which are consistent with the FFT results from the HAADF-STEM images of 34.5°, 86.9°, 97.7°, and 104.7°-TB-MoS$_2$. Besides, the Moiré patterns of 34.5°, 86.9°, 97.7°, and 104.7°-TB-MoS$_2$ are directly observed from HAADF-STEM images, as shown in Fig. 3i–l. For the TB-MoS$_2$, the Moiré periodicity ($L$) can be calculated based on the following equation[43]:

$$L = \frac{a(1 + \delta)}{\sqrt{2 \times (1 + \delta) \times (1 - \cos\theta) + \delta^2}} \tag{5}$$

while $\theta$ represents the relative twist angle, $a$ is the lattice constant of MoS$_2$, and $\delta$ represents the lattice mismatch between two twisted layers. For TB-MoS$_2$, there is no lattice mismatches. Therefore, the equation can be simplified as:

$$L = \frac{a}{\sqrt{2 \times (1 - \cos\theta)}} \tag{6}$$

For bilayer MoS$_2$, $a$ is equal to 0.3165 nm according to the computational equation of the Moiré periodicity. The Moiré periodicity of 34.5°, 86.9°, 97.7°, and 104.7°-TB-MoS$_2$ can be theoretically determined to be 0.71, 0.68, 0.81, and 1.18 nm, respectively. From the HAADF-STEM images, the FFT patterns of the TB-MoS$_2$ with twist angles of 25.5°, 26.9°, 22.3°, and 15.3° are shown in Fig. 3i–l. The clear Moiré patterns are observed, showing a distinct TB-MoS$_2$ stacking

mode and high crystalline quality. Furthermore, the uniformity of Moiré periodicity across the observed region suggests that 2D interface between TB-MoS$_2$ bilayers is pristine. The moiré periods are measured to be 0.73, 0.70, 0.82, and 1.18 nm in 34.5°, 86.9°, 97.7°, and 104.7°-TB-MoS$_2$, which are close to that of the theoretically calculated values. The TB-MoS$_2$ exhibits rotational order without translational symmetry, indicating the formation of a high-quality quasi-crystalline system. Based on the STEM results, it is thus concluded that the synthesized TB-MoS$_2$ has high crystallinity, displaying clear Moiré periodicity and defined twist angles.

## Evolution of spectral properties of TB-MoS$_2$

The twist angle-dependence spectral properties of TB-MoS$_2$ are further investigated. Due to the $D_{3h}^1$ symmetry of monolayer MoS$_2$ and the $D_{3d}^3$ symmetry of bilayer MoS$_2$, the twist angle is optimized from 0°-120° to 0°-60° (Supplementary Fig. 13). The Raman spectra of the TB-MoS$_2$ at different twist angle in the range of 370–415 cm$^{-1}$ is shown in Fig. 4a. As noticed, the vibration frequency of $E_{2g}^1$ and $A_{1g}$ modes shift under the evolutive of twist angles. The $A_{1g}$ vibration mode shows red shift, while the $E_{2g}^1$ mode has opposite behavior (Fig. 4b). With the increase in the twist angle, the $E_{2g}^1$ vibration frequency first increases and then decreases, exhibiting the lowest vibration frequency at twist angle of 0° and 60°. The $A_{1g}$ vibration mode has the opposite behavior with the highest vibration frequency at twist angle of 0° and 60°. Compared with the 0° (60°)-TB-MoS$_2$, the $A_{1g}$ ($E_{2g}^1$) vibration frequencies at other angles demonstrate a constant red shift (blue shift), which is consistent with previous reports[16,19,24]. The red shift of $E_{2g}^1$ mode is associated with a decrease in the dielectric screening when the twist angle increases. This is caused by the increase of the long-range Coulomb interaction between the Mo atoms, resulting in an increase in the effective restoring forces on the atoms. The $A_{1g}$ mode only involves interlayer vibrations of S atoms, and the slight blue shift is attributed to the weakening of interlayer interactions with the increase of twist angles[30,44,45].

The frequency difference between $E_{2g}^1$ and $A_{1g}$ ($\omega_{A_{1g}} - \omega_{E_{2g}^1}$) is related to the twist angles. As shown in Fig. 4c, the $\omega_{A_{1g}} - \omega_{E_{2g}^1}$ reaches its maximum value at 0° and 60°, and decreases in the ranges of 0°–20° and 60°–40°. A global minimum value is obtained at the twist angle of 30°. The $\omega_{A_{1g}} - \omega_{E_{2g}^1}$ can serve as the effective indicator of interlayer coupling. In other words, the larger separation of the $E_{2g}^1$ and $A_{1g}$ peaks, the stronger is the interlayer coupling. Thus, the interlayer coupling strength is the strongest at 0° and 60° due to the minimum interlayer separation according to DFT calculation[46,47]. The S atoms are almost vertically aligned at 30°. Therefore, the interlayer separation is maximized, resulting in the lowest coupling strength. Except for the phonon energies, the Raman intensity also shows the strong relationship to the twist angle, as shown in Fig. 4d–f. The intensity of $A_{1g}$ ($I_{A_{1g}}$) is compared with intensity of $E_{2g}^1$ ($I_{E_{2g}^1}$). Both peak intensities reach the global maxima at around 0° and 60° and then decrease upon approaching 30°. Besides, the peak intensity ratio of $E_{2g}^1$ to $A_{1g}$ ($I_{E_{2g}^1}/I_{A_{1g}}$) shows a W-shaped change with the twist angles (Fig. 4e), demonstrating a twist angle-dependent behavior.

The evolution of band structure and electronic coupling of TB-MoS$_2$ have also been investigated based on the dependence of the twist angle on the PL spectra for TB-MoS$_2$. As demonstrated in Fig. 4f, all the PL emission spectra of TB-MoS$_2$ are well fitted into three peaks, namely A exciton (1.82 eV), B exciton (1.95 eV), and trion (A$^-$, 1.80 eV). The A and B exciton are combined by the electron and hole, and the corresponding emission process at K point of Brillouin zone is shown in Fig. 4g. The A exciton arises from direct excitonic transitions between the minimum of the conduction band (CBM) and the maximum of the valence band (VBM). The B exciton is derived from direct excitonic transitions between the CBM and the valence-band splitting (VBS). The A$^-$ is ascribed to the excess carrier concentration in MoS$_2$ layers, as illustrated in Fig. 4g[48]. The PL emission energies of the A, B

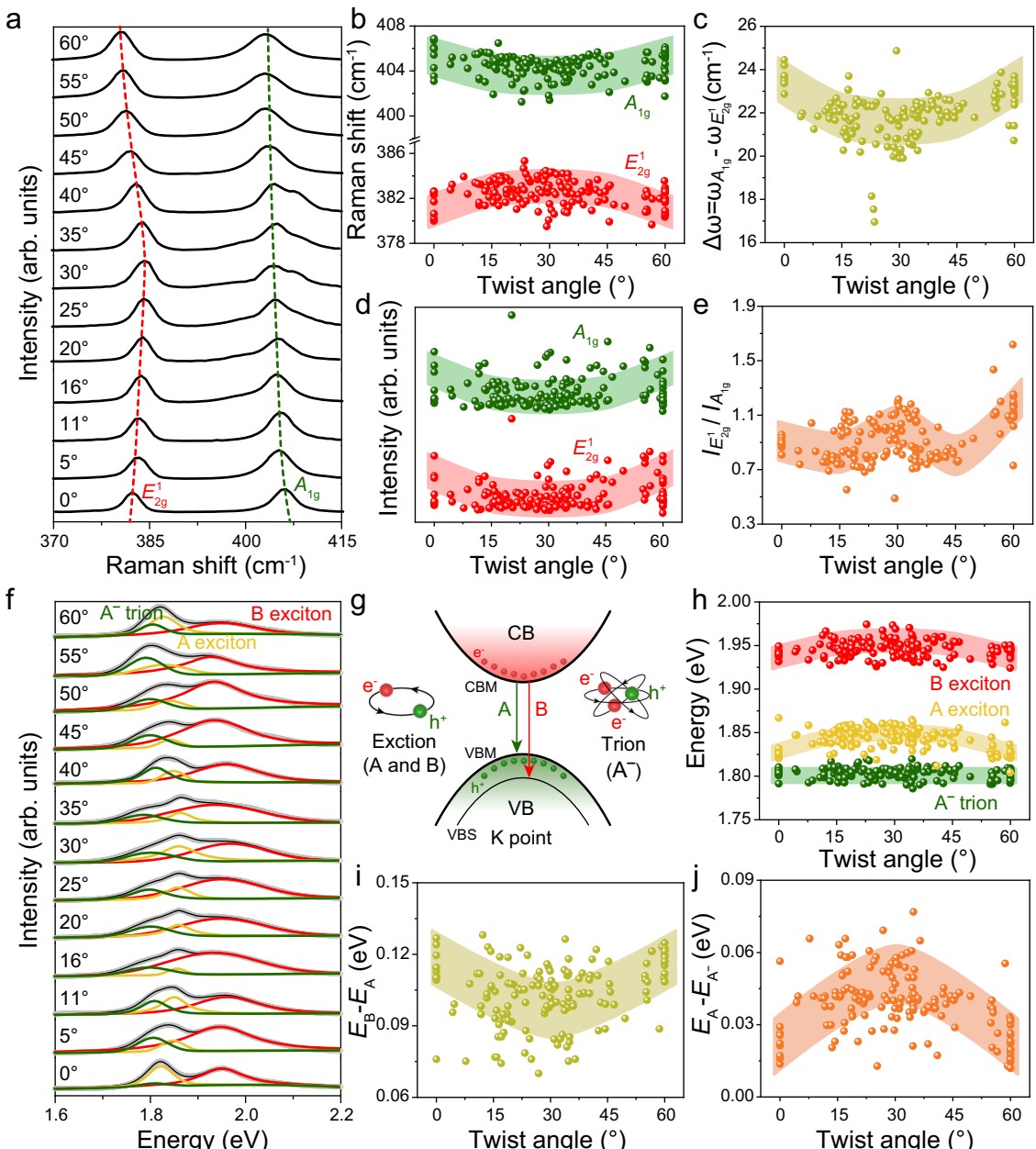

**Fig. 4 | Twist angle-dependent for Raman and PL spectra of TB-MoS₂. a–e** Twist angle-dependent Raman spectra, Raman position ($A_{1g}$ and $E_{2g}^1$), Raman frequency difference ($\omega_{A_{1g}} - \omega_{E_{2g}^1}$), Raman intensity ($A_{1g}$ and $E_{2g}^1$), and Raman intensity ratio ($I_{E_{2g}^1}/I_{A_{1g}}$) in TB-MoS₂, respectively. The colored shaded areas in (**b**)–(**e**) represent a guide to the eye for the evolution of twist angle-dependent Raman spectra. **f** Twist angle-dependent PL emission spectra. **g** Energy band schematic diagram in the vicinity of the Brillouin zone, showing the PL emission process between conduction band (CB) and valence band (VB) in TB-MoS₂. A exciton arises from the minimum of conduction band (CBM) to the maximum of the valence band (VBM), and B exciton derives from CBM to the valence-band splitting (VBS). **h–j** Twist angle-dependent PL emission spectra, PL emission energy (A⁻, A exciton, and B exciton), PL emission energy difference between the B and A excitons ($E_B - E_A$), and the binding energy of A⁻($E_A - E_{A^-}$) in TB-MoS₂, respectively. The colored shaded areas in (**h**)–(**j**) represent a guide to the eye for the evolution of twist angle-dependent PL spectra.

exciton and A⁻ show a strong correlation with the twist angles (Fig. 4h). Compared with the 0° and 60°-TB-MoS₂, the PL emission energies of A and B exciton exhibit a slight red shift (1.82–1.86 eV for A exciton and 1.92–1.96 eV for B exciton) in the ranges of 0°–20° and 60°–40°, while keep constant at around 20°–40°. This indicates that the direct excitonic transitions at K point is slightly sensitive to the twist angle, which is consistent with previous reports[13,19,35,47,49]. The relationship between the twist angles and energy disparity between the B and A excitons ($E_B - E_A$) is shown in Fig. 4i. The $E_B - E_A$ has the maximum difference under the twist angles of 0° and 60°, and then decreases in the range of 0°–20° and 60°–40°. A global minimum difference reaches at 30°. The VBS is strongly dependent on interlayer coupling at the K point. In other words, the VBS increases as the interlayer coupling becomes stronger, which is originating from variations in interlayer separation or interlayer interaction strengths at varying twist angles[50,51]. Thus, the strongest interlayer coupling can be observed in 0° and 60° TB-MoS₂. The increase in interlayer separation reduces the interlayer interaction in angle-dependent TB-MoS₂, resulting in decreased VBS, which is coincident with the Raman results. The A⁻ emission energy shows the constant values of 1.79–1.81 eV from 0° to 60°, suggesting that A⁻ is insensitive to the twist angles (Fig. 4h). But, the trion binding energy (PL emission energy difference between A exciton and trion, $E_A - E_{A^-}$) shows a strong angle-dependence evolution (Fig. 4j). Specifically, due to the smallest interlayer spacing at 0° (60°), the $E_A - E_{A^-}$ has the

minimum value at 0° (60°) and the maximum one at 30°, resulting in easier removal of the electron from the A$^-$ at 0° (60°) than at other angles.

## Key parameters for CVD growing TB-MoS$_2$

Understanding the growth process of the TB-MoS$_2$ is the key for the controllable synthesis of high-quality TB-MoS$_2$, which may inspire the growth strategies for other TB-TMDCs. The CVD setup provides a relative stability gas flow in the inner tube for the growth of TB-MoS$_2$ (Supplementary Note 3 and Supplementary Fig. 14), resulting in excellent reproducibility (Supplementary Fig. 15) and uniformly (Supplementary Fig. 14) for the synthesis of large-area TB-MoS$_2$. Here, the TB-MoS$_2$ can be well synthesized under a narrow reaction temperature range of 770–790 °C, with an optimal temperature of 780 °C (Supplementary Fig. 17). Based on the experimental results, the TB-MoS$_2$ samples are identified by the OM and the corresponding angle distribution is shown in Fig. 5a. About 685 TB-MoS$_2$ samples are collected except 0° (120°) and 60°-TB-MoS$_2$. As noticed, the typical Gaussian distribution is observed both in the range of 0°–60° and 60°–120°. The numbers of twist angles near 30° and 90° are significantly larger than that of the other angles. The similar distribution also can be verified under different growth conditions as shown in Supplementary Fig. 18. The twist angles near 0° (120°) and 60° (in the range of 0°–5°, 55°–65° and 115°–120°) accounting for 3.8% are difficult to grow mainly due to the high stability of 0° and 60°-TB-MoS$_2$[12,16]. The interactions become larger for smaller twist angles, which is consistent with the previously reported TB-Graphene and TB-TMDCs[10,11,15,52]. It is well known that CVD-grown TB-TMDCs with small twist angles is highly challenging due to their unfavorable manner in thermodynamical processes. Thus, based on the twisted nucleation process caused by the introduction of the confined space and NaCl, we realized the reconfiguring nucleation under the assistance of NaCl and confined space, and the TB-MoS$_2$ can be successfully synthesized by CVD. Therefore, some smaller twist angles (such as 1.2° and 58.2°) near magic angle can be well synthesized (Supplementary Fig. 19). Although some smaller twist angles close to 0° (120°) and 60° can be well synthesized, these angles are too small to be identified, because that the angles are determined by the observation from OM with human eyes.

In addition to the superiority of the CVD setup, the key parameters for the synthesis of TB-MoS$_2$ are investigated and shown in Fig. 5b, c. The density and yield are adopted to evaluate the parameters of the CVD method on TB-MoS$_2$. All the TB-MoS$_2$ shows a concentrated distribution with a bottom layer, which ensures a more reliable statistic results on density (Supplementary Fig. 20). As mentioned in Supplementary Note 4, the density is defined as the number of TB-MoS$_2$ per unit area, while the yield is dedicated to the proportion of TB-MoS$_2$ in bilayer MoS$_2$. As shown in Fig. 5b and Supplementary Fig. 21, with the increase in the gas flow rate, the density and the yield of TB-MoS$_2$ first increased and then decreased. The highest yield and density of 17.2% and 28.9 pieces/mm² are obtained under a gas flow rate of 50 sccm, which is much larger than the reported TB-TMDCs (Supplementary Table 1). Besides, the TB-MoS$_2$ can be well synthesized under gas flow rates of 30–90 sccm, indicating that the gas flow rate of 30–90 sccm would be an enhanced growth range. The rapid change in yield and density of TB-MoS$_2$ indicates that the gas flow rate and NaCl are important in the synthesis process, which mainly affects the nucleation density and growth rate of TB-MoS$_2$. This is consistent with the process for the synthesis of bilayer MoS$_2$[53,54]. Apart from the gas flow rate, the molar ratio of NaCl to MoO$_3$ is another key parameter affecting the density and yield of TB-MoS$_2$. To investigate the growth process of TB-MoS$_2$, controlled experiments on the molar ratio of NaCl to MoO$_3$ have been carried out. As shown in Fig. 5c and Supplementary Fig. 22, with the increase in the molar ratio from 0 to 20, the density increases from 0 to 28.9 pieces/mm², while the yield in bilayer MoS$_2$ increases from 0 to 17.2%. The TB-MoS$_2$ cannot be synthesized when the molar ratio of

NaCl to MoO$_3$ reaches 30, indicating that the molar ratio of 10–20 would be a suitable growing parameter range.

The combination of NaCl and confined space promotes the synthesis of the TB-MoS$_2$ (Supplementary Fig. 23). Introducing NaCl can affect the thermodynamics and kinetics in the CVD system (the detailed analysis can be found in Supplementary Note 5)[53]. As shown in Fig. 5d, the MoO$_3$ is insufficient for the reaction without adding NaCl under the low sublimation rate of the sulfur. The MoO$_3$ precursor is evaporated under the temperature. As a result, quadrilateral shapes MoO$_3$ and MoO$_2$ preferentially nucleate and grow on the SiO$_2$/Si substrate (Fig. 5d and Supplementary Figs. 24 and 25). In contrast, the added NaCl can be reacted with the MoO$_3$, forming the oxychloride compounds MoO$_x$Cl$_y$ and enhanced the concentration and the precursor vapor pressure[53,55]. The resultant compounds further react with the sulfur, resulting in the growth of triangular shapes MoS$_2$ crystal on SiO$_2$/Si substrate. The reactant concentrations of MoO$_x$Cl$_y$ are greatly affected by the molar ratio of NaCl to MoO$_3$ at the definite reaction temperature. With a low salt ratio, nuclei with small size are obtained, and consequently, monolayer MoS$_2$ is grown on SiO$_2$/Si substrate (Fig. 5d and Supplementary Fig. 25). With the increase in the molar ratio of NaCl to MoO$_3$, the reactant concentrations of MoO$_x$Cl$_y$ increases, making the nuclei surplus grow larger. As a result, there is a rising number of bilayer MoS$_2$ and the monolayer concomitant on SiO$_2$/Si, and some TB-MoS$_2$ can be observed (Fig. 5d and Supplementary Fig. 25). When the molar ratio of NaCl to MoO$_3$ further increases, the nuclei size becomes even larger, leading to the disappearance of monolayer MoS$_2$. In contrast, the bilayer and even multilayer MoS$_2$ are obtained. The proportion of MoS$_2$ with different layer numbers under different molar ratio of NaCl to MoO$_3$ is shown in Supplementary Fig. 26. The highest proportion of bilayer MoS$_2$ (60.9%) can be obtained under a molar ratio of 20, and the excessive NaCl can result in the growth of multilayer MoS$_2$. Therefore, the TB-MoS$_2$ can be synthesized under the suitable molar ratio of NaCl to MoO$_3$. Based on the calculation results, the 0°-TB-MoS$_2$ and 60°-TB-MoS$_2$ have the lowest energy, which has been widely synthesized by the CVD method[16,47]. Therefore, the TB-MoS$_2$ is hardly grown via a conventional thermodynamic growing method.

Here, introducing a confined space alters the kinetics and consequently realizes anisotropic growth of the TB-MoS$_2$ with the thermodynamically unfavorable twisted stacking structures. The introduction of the confined space can facilitate the synthesis of TB-MoS$_2$ by optimizing the growth kinetics conditions of CVD system including gas velocity, temperature, and turbulent flow. First, the gas velocity can be decreased by the confined space (Supplementary Note 6 and Supplementary Figs. 14 and 27). As we know, the gas flow condition might be an important parameter affecting the sample state in CVD system[53,56]. The low gas velocity in the confined space results in the nuclei surplus with large nuclei. Therefore, the bilayer MoS$_2$ can be easily synthesized and also be verified by the proportion of bilayer MoS$_2$ with the increase in gas flow rate (Supplementary Figs. 14 and 26). The high proportion of bilayer MoS$_2$ (77.6% and 60.9%) can be achieved under the gas flow of 30 sccm and 50 sccm, respectively. We can find that a higher yield of TB-MoS$_2$ can be obtained under a higher proportion of bilayer MoS$_2$. Higher or lower gas flow rates result in a high proportion of multilayer MoS$_2$. Second, with the introduction of the confined space, the temperature is very uniform in the confined space near the SiO$_2$/Si substrate (Supplementary Note 6 and Supplementary Fig. 27). The confined space in the inner tube served as a "heat insulation layer" to prevent the heat diffusion, which benefits the stable synthesis of highly repetitive TB-MoS$_2$. In this growth stage, the in-plane growth is generally faster than out-of-plane growth, resulting in the smaller size of bilayer MoS$_2$ than that of the monolayer (Supplementary Fig. 20).

Most importantly, the turbulent flow changes with the introduction of the confined space. Based on the previous study, the turbulent

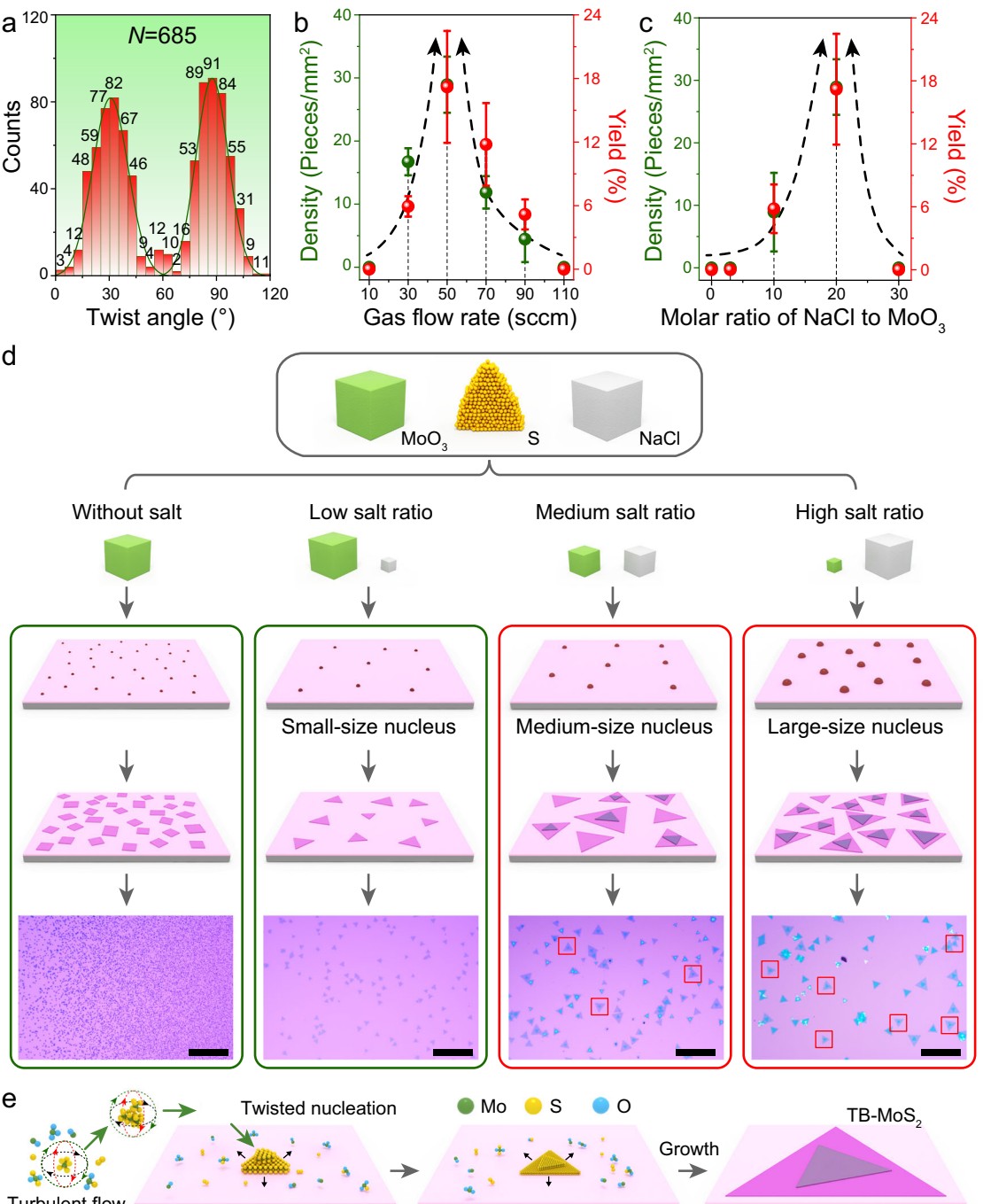

**Fig. 5 | Key parameters for CVD-grown TB-MoS₂.** **a** Statistical distribution and corresponding Gaussian distribution curve of twist angles based on OM of as-grown TB-MoS₂. **b**, **c** Average density, yield of TB-MoS₂, and the standard deviations under different gas flow rate and molar ratio of NaCl to MoO₃, respectively.

**d** Flowchart of the salt-assisted growth process for the reconfiguring nucleation and growth of TB-MoS₂ via the space-confined CVD method, and the boxes represent the location of TB-MoS₂. **e** Growth mechanism diagrammatic of twisted nucleation process under turbulent flow. Scale bar: 100 μm in (**d**).

flow indeed causes the different growth conditions in CVD process, which has been adopted to synthesize 2D materials, such as graphene and TMDCs with different morphologies[15,57,58]. As shown in Supplementary Fig. 27, the components of the velocity vector ($V_y$ and $V_z$) exhibit opposite gas flow directions in the confined space compared to $V_x$, indicating that the backflow gas is generated around the substrate. The deposited precursor molecular clusters are relatively small and limited, which are easily reduced into MoS₂ clusters in the initial growth stage. With the help of the turbulent flow induced by the confined space, the random motion of MoS₂ clusters will be enhanced under the gas phase (Fig. 5e). The collision rate of molecules to

molecules and molecules to substrate will further be increased, which produces enough energy for the in situ twisted nucleation under the gas phase. Then, the MoS₂ clusters can be twisted under the turbulent flow. The size of the twisted nucleus is further enlarged by the interaction with vapor atoms, and consequently deposited randomly on the SiO₂/Si substrate. The twisted MoS₂ nucleus will continuously grow along the edges, leading to the formation of layered MoS₂ flakes (Fig. 5e). Besides, the experimental results (DF-OM) in Supplementary Figs. 4 and 25 show that the MoS₂ layer grows with different rates while sharing the same nucleation site. We believe that the introduced backflow gas through the inner tube drives the entire inner tube out of

equilibrium, resulting in the reduced chance to form more stable 0°- and 60°-TB-MoS$_2$. Therefore, in the high-temperature CVD process, the medium and large nucleus sizes are beneficial for the growth of bilayer MoS$_2$. Based on the twisted nucleation process, we realize that reconfiguring nucleation under the assistance of NaCl and confined space is the key for the successful synthesis of TB-MoS$_2$ by CVD. In addition, it is noteworthy that the proposed CVD technique has been utilized to synthesize other TB-TMDCs. Supplementary Fig. 28 presents the successful growth of TB-WS$_2$ using comparable procedures.

## Discussion

In summary, the reconfiguring nucleation strategy is adopted for the direct synthesis of TB-MoS$_2$ by CVD with twist angles ranging from 0° to 120°, especially some smaller twist angles close to 0° (120°) and 60°. The atomic-resolution LFM results show that the monolayer and bilayer area TB-MoS$_2$ have different orientations, which is used to identify the twist angle in TB-MoS$_2$. The high-quality Tb-MoS$_2$ with various Moiré periodicity and defined twist angles are observed in the STEM. The Raman and PL results indicate the formation of low-defect TB-MoS$_2$ with a strong relationship between interlayer coupling and the twist angle. The key parameters for CVD growing TB-MoS$_2$ are proposed. Increasing molar ratio of NaCl to MoO$_3$ or decreasing the gas flow rate significantly improves the yield and density of TB-MoS$_2$. Although the high yield of TB-MoS$_2$ in bilayer MoS$_2$ (17.2%) and high density (28.9 pieces/mm$^2$) of TB-MoS$_2$ have been achieved, the twist angles are still random and unpredictable. The proportion of small-angle twist angles (in the range of 0°-5°, 55°-65° and 115°-120°) in the total amount of twist angles reaches 3.8%. This work brings inspiration for the controllable growth of TB-MoS$_2$ and other TB-TMDCs, and future works still need to synthesize TB-TMDCs with certain twist angles, especially at small twist angles (near magic angle), possibly by controlling the nucleation process, substrate engineering[59], utilizing the axial screw dislocation[60–62], introducing strain[63], intermediate layers[64,65], screw dislocations[66–68], or non-Euclidean surfaces[69]. Meanwhile, the aid of machine learning is expected to accelerate the optimization process, while image identification is helpful for accurately identifying small twist angles. Both of them will boost the development of TB-TMDCs for twistronics.

## Methods

### Synthesis of TB-MoS$_2$

The TB-MoS$_2$ is grown by atmospheric pressure CVD method on SiO$_2$/Si substrate in a single temperature tube furnace as depicted in Fig. 1a. In a typical synthesis process, two pieces of SiO$_2$/Si substrates (back-to-back, 40 mm in length and 9 mm in diameter) are positioned into a small quartz tube (12 mm in diameter) and the 0.8 mg mixed MoO$_3$/NaCl (the molar ratio of NaCl to MoO$_3$ is 20) are placed around 10 mm from the front of the substrate, and 10 mm from the front opening. Then, the small quartz tube is loaded into a 1-inch quartz tube and placed at the high-temperature range of the furnace. Another crucible containing 0.1 g S powder was placed around 12 cm upstream from the precursors. The CVD process is carried out under ambient pressure and high-purity Ar is used as the carrier gas to facilitate the growth. The tube was cleaned by the 600 sccm Ar for 10 min, and maintained in 50 sccm in the overall process. The temperature of the furnace was gradually increased to 780 °C at a rate of 30 °C/min and held there for 5 min. Subsequently, the furnace was allowed to cool down to room temperature naturally, and the TB-MoS$_2$ was obtained on the SiO$_2$/Si substrate.

### Materials characterization

The morphology and microstructures of TB-MoS$_2$ were characterized by the OM (Nexcope NMM910), FESEM (Gemini SEM 300), STEM (Themis Z), and AFM (Asylum Research Cypher S). The Raman and PL of TB-MoS$_2$ were performed by the WITEC Alpha 300R with a 532 nm laser and calibrated by Raman peak of Si (520 cm$^{-1}$). The atomic-scale LFM images of TB-MoS$_2$ on SiO$_2$/Si substrate were measured by Cypher S AFM under ambient conditions with a temperature of 20 °C and a relative humidity of 30%. An image with 256 × 256 pixels was acquired with a scan rate of 9.77 Hz (high-resolution LFM) and 30 Hz (atomic-resolution LFM) under the contact mode. The SHG characteristics of the TB-MoS$_2$ were evaluated using a home-built vertical microscope setup with the reflection geometry. Similar to our previous work[70], a central wavelength of 1558 nm (PriTel Inc., with a pulse width of 8.8 ps and repetition rate of 18.8 MHz) fiber pulsed laser was used as the fundamental pump light. The pulse laser is focused by the 50× objective (with a numerical aperture of 0.75), and the focused laser spot is about 2 μm. The SHG response was collected using the same objective lens and examined using a spectrometer (Princeton Instruments, SP 2558 and 100BRX) equipped with a cooled silicon CCD camera. To investigate the polarization dependence of the SH radiation, a circularly polarized pump laser was passed through a rotating polarizer to achieve linear polarizations along different directions. Another polarizer was then placed in the signal collection path and rotated accordingly to the pump polarization to gather the parallel components of the SHG signal.

## Data availability

The data that support the findings of this study are available from the corresponding author upon request.

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

## Acknowledgements

The authors gratefully acknowledge financial support by the National Natural Science Foundation of China (62288102, 62304182, 62371397, and 61974120), the Natural Science Foundation of Shaanxi Province (2022JQ-659 and 2023-JC-YB-495), the Practice and Innovation Funds for Graduate Students of Northwestern Polytechnical University (PF2023037), and the Fundamental Research Funds for the Central Universities, the start-up funds from Northwestern Polytechnical Uni-versity (23GH02028). This work was also supported by National Research Foundation–Competitive Research Program (NRF-CRP22-2019-0007 and NRF-CRP21-2018-0007) and A*STAR under its AME IRG Grant (A2083c0052). We would like to thank the Analytical & Testing Center of Northwestern Polytechnical University for STEM, Raman, and PL tests.

## Author contributions

X.W. and W.H. conceived and supervised the project. M.X. designed the experiments. M.X. and H.J. prepared and characterized the samples. L.L. conducted the SEM characterization. J.W. carried out the SHG measurement. Q.L. carried out the simulation of the flow field. M.X., H.J., L.Z., W.L., H.W., J.W., L.L., and Q.L. analyzed the characterization results. M.X. and H.J. wrote the paper. M.X., W.L., X.G., Z.L., X.W., and W.H. revised the paper. All authors discussed the results and commented on the manuscript.

## Competing interests

The authors declare no competing interests.
