## [Peer Review File · Nature Communications]

Reconfiguring nucleation for CVD growth of twisted bilayer MoS₂ with a wide range of twist anglesEditorial Note: Parts of this Peer Review File have been redacted as indicated to remove third-party material where no permission to publish could be obtained.

REVIEWER COMMENTS

Reviewer #1 (Remarks to the Author):

The manuscript reports on the CVD growth of bilayer MoS₂ with entire twist angles (0 to 120°) in a confined space. The controlled growth of bilayer TMDs with specific twisted angles has always been an attractive topic and important challenge in the field. The present work carried out a comprehensive microscopic and optical characterization of the CVD grown bilayer MoS₂. The authors further claimed that density and yield of the twisted bilayer MoS₂ can be tuned by changing the gas flow rate as well as the molar ratio of NaCl to MoO₃. However, several concerns are listed as follows and I would recommend a major revision before the consideration of the present manuscript for publication.

1. Many current progress on the CVD growth of bilayer TMD with controlled twist angles are missing in the introduction section.
2. It is known that the MoS₂ has a hexagonal atomic arrangements and a 6-fold symmetry. There have been several reports published and talked about the CVD growth of bilayer TMDs with a wide twist angle between 0-60°, as well as the change of Raman and PL signals over different twist angles. What is the difference between 0-60 and 60-120° twist angles in this work? The novelty of the present work is arguable.
3. For the CVD growth of binary TMDs, it is important to show the controllability over the twist angles. In the present work, the authors investigated the twist angles of plenty of bilayer MoS₂. It is not surprise that all different twist angles can be found in the sample with preference on several specific angles. Again, novelty of the present work should be clarified and comments on the controllability over the twist angles should be provided.
4. From the low-mag optical images showed in the manuscript, many monolayer and multi-layer MoS₂ flakes can always be found. The selectivity of bilayer MoS₂ by the proposed CVD process should be mentioned and discussed.
5. The authors claimed that the space-confined CVD process can break the thermodynamically unfavorable growth of bilayer MoS₂. However, explanation and discussion on the mechanism is not provided.
6. The flake size of the as-grown bilayer MoS₂ should be clarified when talking about the density of the flakes.

Reviewer #2 (Remarks to the Author):

This manuscript reports a NaCl-assisted space-confined CVD strategy for directly growing twisted bilayer MoS₂. The authors claim that this method can break the thermodynamically unfavourable and achieve entire twist angles from 0 to 120°. Twisted bilayer 2D materials have attracted a lot of interest in recent years, and the development of bottom-up strategies for growing these metastable or thermodynamically unfavourable materials with controllable twist angles is important for this field. There is no evidence in

this manuscript for control over the twist angles of as-grown TB-TMDCs. Nevertheless, the achieved wide range of twist angles and the breaking of the thermodynamically unfavourable stacking order (How and why?) would be a step forward to controllable growth of TB-TMDCs, especially for twist angle control. The study is quite carefully conducted and should be of interest to experts in the field, and can therefore be considered for publication in Nature Communications. However, the paper contains several weak points and issues that are mandatory to be addressed conclusively before acceptance may be considered. A list of the most urgent issues follows below:

1. The term “Reconfiguring nucleation” in the title seems like a proposed strategy, but there is no corresponding statement in the main text. What’s the relationship between the “Reconfiguring nucleation” and the proposed strategy “NaCl-assisted space-confined CVD”?
2. The term “entire twist angles” is oversold. One cannot get all the angles in one paper. The authors didn’t should replace it, maybe with “wide range of twist angles”.
3. The authors paid a lot of attention to characterization but neglected the growth method itself. It is reported by many literatures that halogen salts and the confined space play crucial roles in the CVD growth of TMDCs. How does this combination (NaCl and confined space) break the thermodynamically unfavourable? Both experimental results and theoretical analysis should be strengthened to prove this hypothesis.
4. Figure 5b, it seems that the gas flow rate has a negative effect on the yield of TB-MoS₂. Why? What’s the relationship between this result and the space-confined method? Does the gas turbulence influence on the breaking of the thermodynamically unfavourable? If the confined space changed, what would be happen to the yield curve as a function of the flow rate?
5. The authors claimed that the molar ratio of NaCl and MoO₃ is another key parameter to influence the yield, but show little analysis and corresponding experimental results on the reason.

Reviewer #3 (Remarks to the Author):

The author reports a NaCl-assisted space-confined chemical vapor deposition (CVD) method to direct synthesis bilayer MoS₂ with twist angles ranging from 0 to 120°. The study of twisted transition metal disulfide lasted for more than a decade. Previous studies include ACS Nano 2018, 12, 8, 8770–8780, Nat Commun 5, 4966 (2014), Nano Lett. 2014, 14, 10, 5500–5508, Nat Commun 11, 2153 (2020), Adv. Funct. Mater. 2022, 32, 2111529, ACS Appl. Mater. Interfaces 2023, 15, 3, 4724–4732, CrystEngComm, 2021, 23, 2889–2896, have deeply revealed the optical, electrical, and mechanical properties of both CVD and transferred twisted MoS₂. From this point of view, the characterizations of twisted bilayer MoS₂ in this manuscript are not new. The key point of this manuscript is that the author offers a relatively easy method to grow twisted bilayer MoS₂ with abundant twist angles, which is meaningful for the study of twistronics. However, the research and discussion on the growth parameters are incomplete, and the current manuscript contains defects and mistakes. Hence, I do not support considering this manuscript by Nature Communication at the current stage.

1. The author needs to check the language of this manuscript carefully. It is very hard to read and contains too many grammar problems, such as lines 91: "filed"; 152-155 "may be"; 169 "Six hexagonal points"; 170 "demonstrating which indicate"; 179: "noticed"; and 180: "under".
2. At line 91, "twist angle is defined by the angular difference along the corresponding edges of the top and bottom layers (indicated by dashed green and red lines, respectively). The twist angles are determined from the bright field (BF) optical microscopy (OM) images with an error of about 0.3° (Supplementary Fig. 1)." The author used edges of the top and bottom MoS₂ to determine the twist angle and error. Fig. 1g in the manuscript and Fig. S2, 4, 5, 6, and 7 in the Supplementary information show that most edges of either top or bottom MoS₂ triangle domains are not straight. Thus, just comparing edges is insufficient to determine the twist angle and the error.
3. Line 129-130, "the TB-MoS₂ shows a higher vibration intensity, while the frequency of E_{2g} mode decreases ($\Delta\omega = 2.4 \text{ cm}^{-1}$) and the A_{1g} mode increases ($\Delta\omega = 2.1 \text{ cm}^{-1}$). Please clarify the "vibration intensity" and its relation with the Raman shift.
4. Line 135, the homogenous Raman or PL mappings can not support low defects concentration. The distribution of defects can be uniform.
5. Line 140, "which is consistent with the Raman spectrum in the literature." No citation support.
6. Fig. 2a and e, please mark the "monolayer" and "bilayer" on the spectra.
7. Fig. 3, in the STEM and SAED measurements, it is unclear what method was used to determine twist angles. How much error between the twist angle measured by SAED (STEM) and the OM method?
8. Fig. 5b-c. The statistical detail of the TB-MoS₂ density is not clear. The number of samples (and growth) and the statistic area of each sample are missing. The calculation method of error bars in the figures is ungiven. The twist-angle distribution of TB-MoS₂ under each growth condition is missing (are they the same or not?).
9. The current results show that the density and yield of TB-MoS₂ monopoly increase with the slat ratio and decrease with the flow rate. Is there a limit to this trend?
10. The temperature is also a key parameter in CVD growth but is not involved in the discussion of this research.
11. Line 409, the pressure during the growth is missing.

Response to referees

Response to Reviewer 1

The manuscript reports on the CVD growth of bilayer MoS₂ with entire twist angles (0 to 120°) in a confined space. The controlled growth of bilayer TMDs with specific twisted angles has always been an attractive topic and important challenge in the field. The present work carried out a comprehensive microscopic and optical characterization of the CVD grown bilayer MoS₂. The authors further claimed that density and yield of the twisted bilayer MoS₂ can be tuned by changing the gas flow rate as well as the molar ratio of NaCl to MoO₃. However, several concerns are listed as follows and I would recommend a major revision before the consideration of the present manuscript for publication.

Response: We sincerely appreciate the reviewer for carefully reading our work and providing invaluable suggestions to improve it. In the revised manuscript, we have made our best to address your concerns. Our point-by-point response has been listed below.

Comment 1:

Many current progresses on the CVD growth of bilayer TMD with controlled twist angles are missing in the introduction section.

Response:

Thanks for the valuable comment and suggestion. We have summarized the recently reported CVD growth of TB-TMDCs. Currently, there are three typical strategies reported in the literature for the growth of TB-TMDCs, including control nucleation rate [*Nat. Commun.* **5**, 4966 (2014); *Nanotechnology* **25**, 365603 (2014); *CrystEngComm* **23**, 2889 (2021); *ACS Appl. Mater. Interfaces* **15**, 4724-4732 (2023)], enhanced the growth temperature [*Adv. Opt. Mater.* **3**, 1600-1605 (2015)], as well as the introduction of the catalyst [*Chem. Mater.* **32**, 721-9729 (2020); *J. Semicond.* **44**, 012001 (2023); *Adv. Mater.* **35**, 2210909 (2023)]. Unfortunately, the precise control of the twist angle for TB-TMDCs is not achieved yet. Here, we have added some discussions in the revised manuscript by citing the related literature.

Changes to the revised manuscript are shown below (In Page 4).

It is well known that the rotations between monolayer and bilayer in TB-TMDCs need to overcome a high energy barrier. Therefore, there are fewer reports for the CVD synthesis of TB-TMDCs compared with bilayer TMDCs [*Adv. Mater.* **33**, 2105079 (2021); *J. Am. Chem. Soc.* **144**, 3949-3956 (2022)]. Several approaches have been adopted for the synthesis of TB-TMDCs, such as controlling nucleation rate, temperature enhancement, as well as introduction of catalyst. In specific, Liu et al., [*Nat. Commun.* **5**, 4966 (2014)] reduced the nucleation rate at the initial stage, which can obtain a high proportion of bilayer MoS₂, and acquire break through of high yield of TB-MoS₂ in bilayer MoS₂ (~5%). Han et al., [*CrystEngComm* **23**, 2889-2896 (2021)] introduce the Mo foil instead of MoO₃, which can reduce the nucleation rate and avoid introducing impurities. Shao et al., [*Chem. Mater.* **32**, 9721-9729 (2020)] introduced the tin (Sn) into the CVD system for reducing the stacking energy of WS₂. Similar approach also have been adopted for the synthesis of twisted WSe₂/WSe₂ homostructure [*Adv. Mater.* **35**, 2210909 (2023)]. Zheng et al., [*Adv. Opt. Mater.* **3**, 1600-1605 (2015)] enhanced the reaction temperature to 1100 °C to overcome the angle mismatch in bilayer WS₂, while only non-twisted structure can be observed under a reaction temperature of 850 °C. However, it is still challenging to prepare large-area and clean TB-TMDs with a wide range of twist angles by CVD method especially with small twist angles. Therefore, an efficient approach for growing TB-TMDCs, especially with good repeatability, high yield, and diverse twist angles is urgently required. Additionally, establishing the correlation between the synthesized results and CVD parameters is still highly desirable but fraught with challenges.

Comment 2:

It is known that the MoS₂ has a hexagonal atomic arrangements and a 6-fold symmetry. There have been several reports published and talked about the CVD growth of bilayer TMDs with a wide twist angle between 0-60°, as well as the change of Raman and PL signals over different twist angles. What is the difference between 0-60 and 60-120° twist angles in this work? The novelty of the present work is arguable.

Response:

Thank you for the comment. The point-by-point response has been listed below:

(1) The symmetry analysis in MoS₂.

It is well known that MoS₂ crystals synthesized via CVD method shows different morphologies, such as triangle, truncated triangle, hexagon, 6-pointed stars, 3-pointed stars, and dendrite, as shown in Fig. R1a [*Nat. Mater.* **12**, 554-561 (2013); *Chem. Mater.* **26**, 6371-6379 (2014); *ACS Nano* **15**, 6839-6848 (2021); *ACS Nano*, **12**, 635-643 (2018)]. Under the macroscopic and mesoscopic scales, the symmetry of the film is determined by its geometry instead of the boundary or edge of the material. Here, we choose the triangle, truncated triangle, and hexagon structure as examples to analyze the symmetry, and the corresponding atomic crystalline structure (Fig. R1b). There are two typical edge terminations in MoS₂, namely zigzag-Mo edge and zigzag-S edge. The reported experiment and simulation results indicated that the shape change of MoS₂ is attributed to the concentration ratio of the molybdenum and sulfur source, which has an influence on the kinetic growth dynamics of edges [*Chem. Mater.* **26**, 6371-6379 (2014); *ACS Nano* **15**, 6839-6848 (2021); *ACS Nano* **17**, 127-136 (2023); *ACS Nano*, **12**, 635-643 (2018)]. In specific, the lower S chemical potential leads to a Mo-rich regime, inducing a higher energy of zigzag-Mo edge than that of zigzag-S. Therefore, the equilibrium shape of MoS₂ is expected to be a triangle (**three-fold symmetry**) with zigzag-S edge termination. With the increase in the S chemical potential, the energy of zigzag-S increased, resulting in truncated triangle morphology with the alternating zigzag-S and zigzag-Mo edge (zigzag-S dominant) (**three-fold symmetry**). Importantly, at the crossover from the Mo-rich to S-rich regime, the zigzag-S and zigzag-Mo edges have the same edge energy. As a result, the equilibrium shape of MoS₂ is expected to be a hexagon with alternating zigzag-Mo and zigzag-S edges, which is of **three-fold symmetry**. When the S chemical potential further increases, the equilibrium shape of MoS₂ is expected to be a truncated triangle (**three-fold symmetry**) with alternating zigzag-S and zigzag-Mo edge (zigzag-Mo dominant). In contrast, with a higher S chemical potential (S-rich regime), the equilibrium shape of MoS₂ is expected to be a triangle (**three-fold symmetry**) with zigzag-Mo edge termination. To summarize, with increasing S chemical potential, **the symmetry of MoS₂ is three-fold symmetry**.

The quality of MoS₂ holds significant importance for subsequent investigation and application, because that variation in morphologies lead to distinct crystal qualities. In the CVD synthesized MoS₂, the macroscopical morphology of dendrite (**six-fold symmetry**), 3-pointed stars (**three-fold symmetry**), and corresponding derived structures are not favorable for producing high-quality 2D crystals. In contrast, the single layer MoS₂ with triangle shape

often shows a single crystal. The edge and vertex contain more defect sites and adsorbates due to the edge-dangling bonds. While the other shapes of MoS₂ crystals usually involve a twin crystal growth, especially for the highly symmetric hexagonal MoS₂ (**hexagon and 6-pointed stars structure MoS₂ with six-fold symmetry in macroscopical**), which contains several rotationally symmetric mirror twins, forming a cyclic twin. These mirror twin boundary defects are always existent and non-negligible because they can affect the optical and transport properties of the materials [*Nat. Mater.* **12**, 554-561 (2013)]. The hexagon crystal always possesses a higher density of S vacancies than the triangle because of the rich in-plane defects. Besides, the rotational symmetric growth of hexagonal crystal may also result in abundant S vacancies. [*ACS Appl. Mater. Interfaces* **9**, 63-770 (2017); *J. Mater. Chem. A* **7**, 27603-27611 (2019)].

It is observed that all the bottom-layer and top-layer MoS₂ in TB-MoS₂ are triangle morphology in our manuscript, indicating that single crystal MoS₂ has been successfully synthesized. Therefore, **the three-fold symmetry has been adopted in the analysis of twist angles of TB-MoS₂ based on OM, wherein the twist angle of TB-MoS₂ was calculated from 0 to 120° in our manuscript.**

Fig. R1 The symmetry analysis in CVD synthesized MoS₂. a Typical morphology of CVD synthesized MoS₂. **b** Illustration of the atomic crystalline structure of MoS₂.

(2) The Raman and PL between 0°-60° and 60°-120° twist angles.

As we know, Raman and PL are the micro-area analysis methods, wherein only an area of hundreds of nanometers can be detected under a 532 nm laser. Therefore, we should analyze the symmetry in the micro-area of TB-MoS₂. As we know, the bulk MoS₂ (2H phase) shows the hexagonal Bravais lattice and the space group is $P6_3/mmc$ (D_{6h} a non-symmorphic group). A monolayer MoS₂ contains one Mo and two S atoms. In this case, the inversion symmetry is broken and the space group (more precisely, layer group) is $P\bar{6}m2$ (D_{3h}^1 symmorphic group). As shown in Fig. R2a, the triangle MoS₂ shows a **three-fold symmetry** both in macroscopic OM and hundreds of nanometers area. The bilayer MoS₂ is constructed by adding another S-Mo-S layer, showing the layer group $P\bar{3}m1$ (D_{3d}^3 symmorphic group). Consequently, an odd number of layers has the exact symmetry as the single-layer (absence of inversion symmetry), whereas an even number has the symmetry of a double-layer (with inversion symmetry). Here, 30° and 90°-TB MoS₂ samples were selected to discuss the symmetry of bilayer MoS₂, and a **three-fold symmetry** can be observed in the macroscopic OM (Fig. R2b). As shown in the previous section, the triangle morphology might have zigzag-S or zigzag-Mo edges (Fig. R1). That would mean eight different combinations can be obtained with 30° and 90° twist angles and different zigzag-Mo and zigzag-S edges in the bottom and top-layer, respectively.

Fig. R2 The atom structure of MoS₂. **a** Illustration of the atomic crystalline structure of twist angle MoS₂. **b** BF-OM and DF-OM of 30.0° and 90.0°-TB-MoS₂, scale bars: 10 μm. **c** The corresponding atomic configuration and enlarged area 30.0° and 90.0°-TB-MoS₂ with different zigzag-Mo and zigzag-S edge of the top and bottom layer, respectively.

Fig. R3 Twist angle-dependence for Raman and PL of TB-MoS₂. a-d Twist angle-dependence Raman position (A_{1g} and E_{2g}^1), Raman frequency difference ($\omega_{A_{1g}} - \omega_{E_{2g}^1}$), Raman intensity (A_{1g} and E_{2g}^1), and Raman intensity ratio ($I_{E_{2g}^1}/I_{A_{1g}}$) in TB-MoS₂, respectively. e-g Twist angle-dependence PL emission spectra, PL emission energy (A^- , A exciton, and B exciton), PL emission energy difference between the B and A excitons ($E_B - E_A$), and the binding energy of A^- ($E_A - E_{A^-}$) in TB-MoS₂, respectively.

As shown in Fig. R2c, all the atomic crystalline structures and the enlarged areas of the TB-MoS₂ show a **three-fold symmetry**. All the eight combinations can be classified into two groups. Despite the different twist angles, different zigzag edge was established, and only two groups with different atomic structure was observed. Besides, we can notice that the two groups of atomic structure show **mirror symmetry**, indicating that all the 30° and 90°-TB-

MoS₂ show the same atomic structure or mirror symmetry. Although we only take 30° and 90° -TB-MoS₂ as examples, the mirror symmetry can be observed in all the micro-area of the TB-MoS₂ samples due to the crystal symmetry. Considering the Raman and PL measurement under micro area, the twist angle ($\theta_{\text{Raman-PL}}$) under the Raman and PL can be reduced to 0°~60° based on the following equations by the twist angle (θ_{OM}) measured from OM.

$$\theta_{\text{Raman-PL}} = \theta_{\text{OM}} \quad (0 \leq \theta_{\text{OM}} \leq 60^\circ) \quad (1)$$

$$\theta_{\text{Raman-PL}} = 120^\circ - \theta_{\text{OM}} \quad (60^\circ < \theta_{\text{OM}} \leq 120^\circ) \quad (2)$$

In fact, similar to the relationship between $\theta_{\text{Raman-PL}}$ and θ_{OM} , the evolution of Raman and PL with the twist angle in the range of 60° to 120° is mirror-symmetric with the range of 0° to 60°. The comparisons of the twist angle-dependence Raman and PL of TB-MoS₂ with the range of 0° to 60° and 0° to 120° have been carried out in Fig. R3. It should be noted that the imaginary point at the twist angle of 120° is the same with 0° samples (just to show symmetry). As shown in Fig. R3, all the data exhibits a well mirror symmetry with a center of 60°, which is well-matched with the equations. This phenomenon is also reported in previous papers [*Nano Lett.* **14**, 5500-5508 (2014)]. Most of the twist angle-dependence Raman and PL analysis shows the twist angle range from 0° to 60° by CVD [*Nat. Commun.* **5**, 4966 (2014); *CrystEngComm* **23**, 2889 (2021)], two-step stacking [*Nano Lett.* **14**, 3869-3875 (2014); *Nano Lett.* **16**, 1435-1444 (2016); *ACS Nano* **10**, 2736-2744 (2016)], or folding methods [*ACS Appl. Mater. Interfaces* **13**, 22819-22827 (2021)].

Therefore, the twist range of 0° to 60° had been adopted in the analysis of Raman and PL. To ensure an easy reading, we have added corresponding expressions in the revised manuscript and Supplementary Information.

Changes to the revised manuscript are shown below (In Page 13).

Due to the D_{3h}^1 symmetry of monolayer MoS₂ and the D_{3d}^3 symmetry of bilayer MoS₂, the twist angle is optimized from 0°~120° to 0°~60° (Supplementary Fig. 13).

Changes to the revised Supplementary Information are shown below (In Page 27).

As we know, Raman and PL are micro-area analysis methods, wherein only an area of

hundreds of nanometers can be detected under a 532 nm laser. Due to the D_{3h}^1 symmetry of monolayer MoS₂ and the D_{3d}^3 symmetry of bilayer MoS₂, the twist angle is optimized from 0°~120° to 0°~60°. As shown in Supplementary Fig. 13a, the triangle MoS₂ shows a three-fold symmetry both in macroscopic OM and hundreds of nanometers area. Three-fold symmetry can also be observed for the macroscopic OM of 30° and 90°-TB-MoS₂ (Supplementary Fig. 13b). It should be noted that the triangle morphology might own the zigzag-S or zigzag-Mo edges, which means eight different combinations (Supplementary Fig. 13c) can be obtained with 30° and 90° twist angle and different zigzag-Mo and zigzag-S edge in bottom and top-layer, respectively. Eight combinations can be classified into two groups with different atom structures. Besides, the two groups of atom structure show mirror symmetry, which indicates that all the 30° and 90°-TB-MoS₂ show the same atom structure or mirror symmetry. Although only 30° and 90°-TB-MoS₂ were discussed, the mirror symmetry can be observed in all the micro-areas of the TB-MoS₂ samples due to the crystal symmetry. Therefore, considering the Raman and PL were measured under the micro-area, the twist angle ($\theta_{\text{Raman-PL}}$) under the Raman and PL can be reduced to 0°~60° based on the following equations by the twist angle (θ_{OM}) measured from OM.

$$\theta_{\text{Raman-PL}} = \theta_{\text{OM}} \quad (0 \leq \theta_{\text{OM}} \leq 60^\circ) \quad (14)$$

$$\theta_{\text{Raman-PL}} = 120^\circ - \theta_{\text{OM}} \quad (60^\circ < \theta_{\text{OM}} \leq 120^\circ) \quad (15)$$

Supplementary Fig. 13 The atom structure of MoS₂. **a** Illustration of atomic crystalline structure of twist angle MoS₂. **b** BF-OM and DF-OM of 30.0° and 90.0°-TB-MoS₂, scale bars: 10 μm. **c** The corresponding atomic configuration and enlarged area 30.0° and 90.0°-TB-MoS₂ with different zigzag-Mo and zigzag-S edge of top and bottom layer, respectively.

(3) Comparison with CVD growth of bilayer TMDCs with the previous report.

We have summarized the recent reports on the CVD growth of TB-TMDCs, as shown in Table R1. In terms of synthesis strategy, three typical strategies have been adapted for the growth of TB-TMDCs, including controlling nucleation rate [*Nat. Commun.* **5**, 4966 (2014); *Nanotechnology* **25**, 365603 (2014); *CrystEngComm* **23**, 2889 (2021); *ACS Appl. Mater. Interfaces* **15**, 4724-4732 (2023)], enhancing the growth temperature [*Adv. Opt. Mater.* **3**, 1600-1605 (2015)], as well as introducing catalysts [*Chem. Mater.* **32**, 721–9729 (2020); *J. Semicond.* **44**, 012001 (2023); *Adv. Mater.* **35**, 2210909 (2023)]. However, it is still challenging to grow large-area, clean TB-TMDs with wide range of twist angles by CVD method, especially with small twist angles. Therefore, an efficient approach for growing TB-TMDCs, especially with good repeatability, high yield, and diverse twist angles is urgently required. Additionally, establishing the correlation between the synthesized results and CVD parameters is highly desirable but fraught with challenges.

Compared with previous works, our proposed reconfiguring nucleation strategies can change the thermodynamically and kinetics of the CVD process by introducing NaCl and a confined space. We notice that only two works show the rich twist angles by CVD method [*Nat. Commun.* **5**, 4966 (2014); *CrystEngComm* **23**, 2889 (2021)] (Table R1). Compared with these works, our reconfiguring nucleation strategy achieve a wide range of twist angles from 0 to 120°, a high yield (17.2%), and a high density (28.9 pieces/mm²) of TB-MoS₂ (Fig. R4). It is well known that CVD-grown TB-TMDCs with small twist angles are highly challenging due to their unfavorable manner in thermodynamical processes, and most of the reported twist angles of TB-TMDCs are larger than 10°. Thus, based on the twist nucleation process caused by the introduction of the confined space and NaCl, we realized the reconfiguring nucleation under the assistance of NaCl and a confined space to synthesize TB-MoS₂ by CVD. Therefore, some smaller twist angles (such as 1.2° and 58.2°) near the magic angle can be well synthesized (Fig. R5). In our work, the proportion of small-angle twist angles (in the range of 0~5°, 55~65°, and 115~120°) in the total amount of twist angles can reach to 3.8%. In contrast, previous work hardly obtain the small twist angles of TB-MoS₂.

Most importantly, we find out the key parameters for growing of TB-MoS₂. A systematic analysis and discussion of the growing process, and help establish the correlation between the synthesized results (yield, density, and proportion of bilayer) and CVD parameters (gas flow rate and molar ratio of NaCl to MoO₃). This would be highly desirable for CVD synthesis of

TB-TNDCs, which can be extended to the synthesis of TB-WS₂. It is beneficial for establishing a rich material library. Therefore, we believe that the reconfiguring nucleation strategy and the related growth conditions are important for growing TB-TMDCs, which is novel enough for potential readers in Nature Communications.

Fig. R4 Key parameters for CVD grown TB-MoS₂. **a** Statistical distribution of twist angles based on OM of as-grown TB-MoS₂. **b, c** Density and yield of TB-MoS₂ under different gas flow rate and molar ratio of NaCl to MoO₃, respectively.

Fig. R5 OM of synthesized 1.2° and 58.2°-TB-MoS₂. **a-b** OM images and corresponding twist angles of 1.2° and 58.2°-TB-MoS₂, respectively. Scale bars: 10 μ m.

Table R1 Summary of the CVD preparation for TB-TMDCs.

Materials	Synthesis parameters of TB-TMDCs			Results of TB-TMDCs			Ref.	
	CVD Setup	Precursors	Growth conditions	Twist angle produced	Density	Proportion of bilayer		Yield
MoS ₂	Conventional method , 1-inch quartz tube, mica, fused silica, and SiO ₂ /Si substrate	20 mg MoO ₃ in a crucible, 7 mg S in another crucible	Sit at 105 °C with 500 sccm for 1 h, ramp to 700 °C at 15 °C min ⁻¹ with 10-15 sccm N ₂ , sit at 700 °C for 5-10 min, cool down naturally with 500 sccm N ₂	0° to 60°	—	30%	~5%	Nat. Commun. 5 , 4966 (2014)
MoS ₂	Conventional method , 1-inch quartz tube, SiO ₂ /Si substrate	Mo foil, 600 mg S,	T _S =270 °C, T=795 °C, 70 sccm Ar, t=30 min	0° to 60°	—	—	—	CrystEngComm 23 , 2889 (2021)
MoS ₂	Conventional method , SiO ₂ /Si substrate	8mg NaCl, 20mg MoS ₂	900 °C for 50 min with 60 sccm Ar/H ₂ (95% Ar)	12°	—	—	—	ACS Appl. Mater. Interfaces 15 , 4724-4732 (2023)
MoSe ₂	Conventional method , 1-inch quartz tube, SiO ₂ /Si substrate	MoO ₃ , Se powder	T _{Se} =300 °C, T=700 °C, 65 sccm Ar and 5 sccm H ₂ , t=10 min	θ = 7°, 21°, 25°	—	—	—	Nanotechnology 25 , 365603 (2014)
WS ₂	Conventional method , two-temperature-zone tube, quartz substrate	WO ₃ powder, 0.1 g S powder	200 sccm Ar for 30 min before heating, 20 sccm Ar for ramping to 1100 °C at the rate of 20 °C min ⁻¹ , growing for 20 min	θ = 13°, 30°, 41°, 83°	—	—	—	Adv. Opt. Mater. 3 , 1600-1605 (2015)
WS ₂	Conventional method , SiO ₂ /Si substrate	10 mg SnO ₂ , 5 mg NaCl, 100 mg WO ₃ , 480 mg S powder	180 sccm Ar for 10 min before heating, 180 sccm Ar and 15 sccm H ₂ for ramping to 810 °C in 45 min, growing for 3 min	θ = 10°, 20°, 31°, 36°, 40°, 55°	—	90%	—	Chem. Mater. 32 , 9721-9729 (2020) J. Semicond. 44 , 012001 (2023)
WSe ₂	Conventional method , SiO ₂ /Si substrate	30 mg WO ₃ , 10 mg SnO ₂ , 100 mg Se	810 °C for 10 min with Ar and H ₂	θ = 1.5°, ~24°, ~30°	—	—	—	Adv. Mater. 35 , 2210909 (2023)
MoS ₂	Space-confined method , 1-inch quartz tube, SiO ₂ /Si substrate	0.8 mg mixed MoO₃/NaCl, sufficient S powder	600 sccm for 10 min before heating, 50 sccm Ar for ramping to 780 °C at a rate of 30 °C/min, growing for 5 minutes	0° to 120° (3.8% for small twist angles)	28.9 pieces/mm²	60.9%	17.2%	Our work

Proportion of bilayer: The proportion of bilayer TMDCs in all obtained TMDCs domains.

Yield: The proportion of TB- TMDCs in the bilayer TMDCs.

(4) The novelty and significance of our work.

We would also like to briefly discuss the novelty and significance of our work as follows:

Beyond magic-angle TB-graphene, TB-TMDCs with small twist angles close to 0° (120°) and 60° are also fascinating because of the emerging moiré flat bands by providing an ideal platform for the investigation of intriguing fundamental physical properties. Even though CVD has been widely adopted to grow high-quality 2D materials, controllable synthesis of TB-TMDCs with small twist angles is highly challenging due to their unfavorable manner in thermodynamical processes. Therefore, effective optimal synthesis conditions of TB-TMDCs are urgently required.

In this work, the direct synthesis of TB-MoS₂ with a wide range of twist angles from 0 to 120° , especially in small twist angles such as 1.2° and 58.2° , has been achieved by introducing a confined space and salt to a conventional CVD system. Apart from the direct synthesis of TB-MoS₂, a novel CVD growth mechanism is proposed: the molar ratio of NaCl and metal oxide precursor plays key roles in the growing process. As a result, an ultrahigh density (28.9 pieces/mm²) and high yield (17.2%) of TB-MoS₂ can be easily controlled. More importantly, the proposed strategy has been expanded for the synthesis of high-quality TB-WS₂. Besides, the synthesized TB-MoS₂ shows a strong relationship between interlayer coupling and the twist angles in van der Waals-coupled layers. The angle-dependent valence-band splitting behavior of our CVD-grown TB-MoS₂ is opposite to that of the two-step stacking configuration, which is mainly caused by the incommensurate stacking of in-plane rotational symmetry and translational gliding symmetry. Our work opens an avenue for the precise growth of TB-TMDCs, which has great significance for both fundamental research and practical applications and will boost the development of TB-TMDCs for twistrionics.

Comment 3:

For the CVD growth of binary TMDs, it is important to show the controllability over the twist angles. In the present work, the authors investigated the twist angles of plenty of bilayer MoS₂. It is not surprise that all different twist angles can be found in the sample with preference on several specific angles. Again, novelty of the present work should be clarified and comments on the controllability over the twist angles should be provided.

Response:

Thank you for raising these insightful comments. Indeed, “*controllability over the twist angles*” is the goal of the controlled growth of 2D materials such as TB-graphene and TB-TMDCs. However, twisted configurations are thermodynamically unfavourable, making it challenging for accurate control of twist angles during growth. To be specific, there are significant challenges mainly caused by two TMDCs layers forming energy-favorable 0°- and 60°-TB-TMDCs. The nucleation and growth of twisted bilayers only occur randomly at the secondary layer, resulting in the less accuracy to control twist angles [*Nat. Commun.* **5**, 4966 (2014); *Small Struct.* **2**, 2000153 (2021)]. Here, we proposed a reconfiguring nucleation strategy to enhance the formation of TB-MoS₂. We strongly believe the results bring us one step closer to this goal by overcoming the energy preference of 0°- and 60° TB-MoS₂, and the Reviewer 3 agreed by stating that “*The key point of this manuscript is that the author offers a relatively easy method to grow twisted bilayer MoS₂ with abundant twist angles, which is meaningful for the study of twistronics*”.

As we all know, the ideal way is to directly and precisely control the twist angles of the TB-TMDCs. It is challenging for CVD synthesis of TB-TMDCs, and we are dedicated to achieving this goal. However, for another 2D material of graphene, the controllable angles of the TB-graphene have been preliminarily achieved [*Nat. Mater.* **21**, 1263-1268 (2022)]. Liu Group control the synthesis of TB-graphene with the accurate angle replication from two prerotated single-crystal Cu (111) foils. Through this ingenious strategy, they can obtain bilayer graphene with arbitrary twist angles by artificially replicating the relative rotation angle of Cu foils on a macroscale. While for TMDCs, the uniform nucleation and epitaxy of monolayer and bilayer MoS₂ have been achieved based on the different step orientations of sapphire [*Nature* **605**, 69-75 (2022); *Nat. Nanotechnol.* **16**, 1201-1207 (2021); *Nat. Nanotechnol.* (2023) DOI: 10.1038/s41565-023-01445-9]. Therefore, we are trying to synthesize TB-TMDCs with precise twist angles based on substrate engineering with the two prerotated sapphire substrates. Besides, some other substrates for controlling the anisotropic growth of the 2D TMDCs will also be top-priority.

Furthermore, we can also draw inspiration from the alternative approaches in preparing other twisted structures, with the hope of achieving further controllability of twist angles. In addition to the above-mentioned substrate engineering, utilizing the axial screw dislocation [*Nature* **570**, 358-362 (2019); *Nat. Nanotech.* **3**, 477-481 (2008); *Nature* **570**, 354-357 (2019)],

by introducing strain [*ACS Nano* **15**, 4504-4517 (2021)], intermediate layers [*Science* **361**, 782 (2018); *Nat. Commun.* **10**, 5528 (2019)], screw dislocations [*J. Am. Chem. Soc.* **139**, 3496-3504 (2017); *Nano Lett.* **18**, 3885-3892 (2018); *Nano Lett.* **21**, 7815-7822 (2021)] or non-Euclidean surfaces [*Science* **370**, 442-445 (2020)] are also great chances.

However, before we can achieve precise control of the twist angles of TB-TMDCs, we should first realize the large-area preparation TB-TMDCs with abundant twist angles, especially for small twist angles. It is well known that CVD-grown TB-TMDCs with small twist angles is highly challenging due to their unfavorable manner in thermodynamical processes, and most of the reported twist angles of TB-TMDCs are larger than 10°. Therefore, it is a high priority to break the thermodynamically unfavourable twisted stacking structures. Besides, we can synthesize high-quality large-area TB-TMDCs with various twist angles on the substrate and artificially fast screen and characterization the corresponding samples with the specific twist. Although we cannot precisely control the twist angles in the current stage, we can find samples with any twist angle of interest, especially for the small twist angle samples. Thus, the yield, density, as well as proportion of small-angle twist angles are quite important to evaluate the CVD method.

In this manuscript, the reconfiguring nucleation strategy help enhance the formation of TB-MoS₂. The confined space and NaCl have been introduced to a conventional CVD system for the growth of the TB-MoS₂ with thermodynamically unfavourable twisted stacking structures. That's why the Reviewer 2 commendatory our strategy “*achieved wide range of twist angles and the breaking of the thermodynamically unfavourable stacking order would be a step forward to controllable growth of TB-TMDCs, especially for twist angle control.*”. While for the next stage, we would like to combine the reported approaches for controlling the nucleation process [*Nature* **556**, 355-359 (2018); *ACS Nano* **15**, 4504-4517 (2021); *Nature* **605**, 69-75 (2022); *Nat. Rev. Methods Primers* **1**, 5 (2021)] and our reconfiguring nucleation strategy. It is of high chance to achieve the contraollabile preparation of TB-MoS₂ with higher density, larger domain size, and higher yield in the future. This direction will continue to be explored in our future work. In addition, our approach is expected to be applicable to other 2D TMDCs with interlayer twists, based on the intentional design of the reconfiguring nucleation strategy. Therefore, to address the reviewer's concern, we have added a brief discussion on the controllability over the twist angles in the revised manuscript.

Changes to the revised manuscript are shown below (In Page 5 and Page 22).

This synthesis strategy has been extended for preparing high-quality TB-WS₂, shedding light on the precisely controllable growth of other 2D TMDCs with controllable interlayer twists.

This work brings inspiration for the controllable growth of TB-MoS₂ and other TB-TMDCs, and future works still need to synthesize TB-TMDCs with certain twist angles, especially at small twist angles (near magic angle), possibly by controlling the nucleation process, substrate engineering [*Nat. Mater.* **21**, 1263–1268 (2022)], utilizing the axial screw dislocation [*Nature* **570**, 358-365 (2019); *Nat. Nanotech.* **3**, 477-481 (2008); *Nature* **570**, 354-357 (2019)], introducing strain [*ACS Nano* **15**, 4504-4517 (2021)], intermediate layers [*Science* **361**, 782 (2018); *Nat. Commun.* **10**, 5528 (2019)], screw dislocations [*J. Am. Chem. Soc.* **139**, 3496-3504 (2017); *Nano Lett.* **18**, 3885-3892 (2018); *Nano Lett.* **21**, 7815-7822 (2021)], or non-Euclidean surfaces [*Science* **370**, 442-445 (2020)]. Meanwhile, the aid of machine learning is expected to accelerate the optimization process, while image identification is helpful for accurately identifying small twist angles. Both of them will boost the development of TB-TMDCs for twistrionics.

Comment 4:

From the low-mag optical images showed in the manuscript, many monolayer and multi-layer MoS₂ flakes can always be found. The selectivity of bilayer MoS₂ by the proposed CVD process should be mentioned and discussed.

Response:

We appreciate the reviewer's comment and suggestion. Based on the OM results (Fig. R6 and Fig. R7), we have calculated the different proportions of monolayer, bilayer, and multilayer MoS₂ under different synthesis conditions, as shown in Fig. R8. As noticed, with the increase of gas flow rate from 10 to 30 sccm, the proportion of bilayer increased with a maximum proportion of 77.6% at a gas flow of 30 sccm. However, the proportion of bilayer MoS₂ sustained fall when the gas flow rate is up to 110 sccm. We can obtain a high proportion of bilayer MoS₂ (60.9%) in the optimized gas flow of 50 sccm (with the highest yield and density). Higher or lower gas flow rates all result in a high proportion of multilayer MoS₂ (>90%). And the monolayer MoS₂ shows the highest proportion of 19.7% under the gas flow

of 70 sccm. For the molar ratio of NaCl to MoO₃, we cannot synthesize MoS₂ without NaCl. Interestingly, the high proportion of monolayer MoS₂ (94.0%) can be obtained under a low NaCl to MoO₃ molar ratio of 3. With the increase in the molar ratio of NaCl to MoO₃, the bilayer MoS₂ increases first and then decreases, exhibiting a maximum proportion (60.9%) under the molar ratio of NaCl to MoO₃ of 20.

Fig. R6 Typical BF-OM of synthesized products under different gas flow rate. a-f The typical OM synthesized under different gas flow rate of 110, 90, 70, 50, 30, and 10 sccm. Scale bars: 100 μm for the OM under 20X objective, 10 μm for the small area under 50X objective. The TB-MoS₂ was synthesized with fixed NaCl to MoO₃ of 20.

Fig. R7 Typical BF-OM of synthesized products under different molar ratio of NaCl to MoO₃. a-e The typical OM images of the synthesized MoS₂ under different molar ratio of NaCl to MoO₃ of 30, 20, 10, 3, and 0. Scale bars: 100 μm for the OM under 20X objective, 10 μm for the small area under 50X objective.

Fig. R8 Proportion of monolayer, bilayer, multilayer MoS₂ and non-MoS₂ in the as-grown samples under a different gas flow rate and b molar ratio of NaCl to MoO₃, respectively.

Therefore, to address the reviewer's concern, we have conducted additional experiments and discussions in the revised manuscript and Supplementary Information.

Changes to the revised manuscript are shown below (In Page 19).

When the molar ratio of NaCl to MoO₃ further increases, the nuclei size becomes even larger, leading to the disappearance of monolayer MoS₂. In contrast, the bilayer and even multilayer MoS₂ are obtained. The proportion of MoS₂ with different layer numbers under different molar ratio of NaCl to MoO₃ is shown in Supplementary Fig. 26. The highest proportion of bilayer MoS₂ (60.9%) can be obtained under a molar ratio of 20, and the excessive NaCl can result in the growth of multilayer MoS₂. Therefore, the TB-MoS₂ can be synthesized under the suitable molar ratio of NaCl to MoO₃.

Therefore, the bilayer MoS₂ can be easily synthesized and also be verified by the proportion of bilayer MoS₂ with the increase in gas flow rate (Supplementary Fig. 14 and Supplementary Fig. 26). The high proportion of bilayer MoS₂ (77.6% and 60.9%) can be achieved under the gas flow of 30 sccm and 50 sccm, respectively. Higher or lower gas flow rates result in a high proportion of multi-layer MoS₂.

Changes to the revised Supplementary Information are shown below (In Page 35, Page 36, and Page 40).

Supplementary Fig. 21 Typical BF-OM of synthesized products under different gas flow rate. a-f The typical OM synthesized under different gas flow rate of 110, 90, 70, 50, 30, and

10 sccm. Scale bars: 100 μm for the OM under 20X objective, 10 μm for the small area under 50X objective. The TB-MoS₂ was synthesized with fixed NaCl to MoO₃ of 20.

Supplementary Fig. 22 Typical BF-OM of synthesized products under different molar ratio of NaCl to MoO₃. a-e The typical OM images of the synthesized MoS₂ under different molar ratio of NaCl to MoO₃ of 30, 20, 10, 3, and 0. Scale bars: 100 μm for the OM under 20X objective, 10 μm for the small area under 50X objective.

Supplementary Fig. 26 Proportion of monolayer MoS₂, bilayer MoS₂, multi-layer MoS₂ and non-MoS₂ in the as-grown samples. a-b Under different gas flow rate and molar ratio of NaCl to MoO₃, respectively.

Comment 5:

The authors claimed that the space-confined CVD process can break the thermodynamically unfavorable growth of bilayer MoS₂. However, explanation and discussion on the mechanism is not provided.

Response:

Thank you for raising this insightful concern. Firstly, we apologize for the incurred misunderstanding. In our previous works, we want to point out that the introduction of NaCl and the confined space in CVD system is benefit for CVD growth of the TB-MoS₂ with the thermodynamically unfavourable twisted stacking structures. However, intended meaning of our original statement in the manuscripts was not clearly due to the misleading sentence of “*NaCl-assisted space-confined CVD process to break the thermodynamically unfavorable for the direct synthesis of TB-MoS₂*”. Therefore, we have changed the corresponding description in the revised manuscript. Based on your suggestion, we have carried out additional experiments and theoretical analysis to demonstrate the kinetics effects on CVD synthesis of TB-MoS₂ with the confined space. The point-by-point response has been listed below.

(1) Experimental results on the confined space.

Based on the reviewer’s suggestion, we have carried out the experiments based on the confined space. The typical OMs are shown in Fig. R9. The MoS₂ can be well synthesized with the assistance of NaCl, and the TB-MoS₂ is obtained under the confined space. In contrast, the CVD without a confined space yielded only monolayer crystals, suggesting that the confined space is the key for controlling the morphology of MoS₂ products. Therefore, we believe that the combination of the confined space will affect the kinetics of the CVD process. The detail discussions are shown in below.

Fig. R9 The typical BF-OM of products synthesized TB-MoS₂. **a** with and **b** without confined space. Scale bars: 100 μm for the OM under 20X objective, 10 μm for the small area under 50X objective.

(2) The effect of confined space.

To further clarify the effect of the confined space, we have carried out the computational fluid dynamics simulations of the confined space in CVD. The velocity distribution, temperature distribution, and components of the velocity vector for this CVD setup are shown in Fig. R10. Here, we believe that the confined space will change the gas flow rate, the temperature, and the gas flow direction. The corresponding detail discussion are shown below.

Fig. R10 Velocity distribution, temperature distribution, and velocity vector maps of the CVD setup. **a-d** Velocity and temperature distribution of the CVD setup with and without confined space, respectively. **e-g** Velocity vector maps of the components of the velocity vector of V_x , V_y , and V_z , respectively. All the results were carried out with a gas flow rate of 50 sccm and a temperature of 780 °C.

A. Gas flow rate.

It is observed from the velocity distribution that the gas flow condition affects the sample state in the open tube system (Fig. R10 and Fig. R11). It can be identified that the gas velocity in the inner tube (without sealed-end) is one order of magnitude less than that of the outer tube, indicating that there is a change in the gas fluid velocity on the SiO₂/Si substrate. With the decrease in gas flow rate, the fluid velocity of carrier gas decreases, making the nuclei surplus. As a result, the nuclei will grow larger, which is consistent with the experimental

results (Fig. R6). As we know, small change in the flow regime in CVD significantly influences the reaction products. Although the gas velocity in the inner tube has been greatly reduced, there is a nonnegligible increase. Therefore, there is a rapid change in the yield and density of TB-MoS₂, indicating that the gas flow rate is important in the synthesis process, which mainly affects the nucleation and growth of TB-MoS₂.

Fig. R6 Typical BF-OM of synthesized products under different gas flow rate. a-f The typical OM synthesized under different gas flow rate of 110, 90, 70, 50, 30, and 10 sccm. Scale bars: 100 μm for the OM under 20X objective, 10 μm for the small area under 50X objective. The TB-MoS₂ was synthesized with fixed NaCl to MoO₃ of 20.

Fig. R11 Velocity distribution of the space-confined CVD setup under different gas flow rates. a The velocity distribution with the small inner-tube under gas flow rates of 10 to 110 sccm, respectively. **b** The enlarged velocity distributions of **a**.

B. Temperature

Temperature is another key parameter for CVD process, wherein the different reaction temperatures may result in different morphology [ACS Nano 12, 635-643 (2018); Nat Commun 10, 598 (2019); Nat Commun 12, 809 (2021); ACS Nano 13, 8265-8274 (2019)]. Based on previous reports, the preparation of MoS₂ usually requires a relatively high reaction temperature from 600 °C to 1000 °C [Nature 556, 355–359 (2018); Nat Commun 10, 598 (2019); Nature 605, 69-75 (2022)]. However, based on our experimental results, we find that the TB-MoS₂ can only be well synthesized under a narrow reaction temperature range from 770 °C to 790 °C. As shown in Fig. R12, TB-MoS₂ can be obtained under a reaction temperature of 780 °C. Higher or lower temperatures result in the dramatic decrease in the

yield of TB-MoS₂. For example, large-area monolayer MoS₂ is synthesized under a reaction temperature of 760 °C.

Temperature is the key to affect the thermodynamics and dynamics of the reaction system. Here, we have carried out the simulation on the temperature in the tube, as shown in Fig. R10c-d. As noticed, due to the effect of the gas flow rate, the temperature shows an uneven distribution. However, with the introduction of the confined space, there is a uniform temperature distribution near the SiO₂/Si substrate. The inner tube serves as a “heat insulation layer” to prevent the heat diffusion. Thus, we believe that the confined space plays the key role to stabilize the substrate temperature.

Fig. R12 Typical BF-OM of the synthesized products under different reaction temperatures. a-e Typical OM images of the synthesized MoS₂ under different reaction temperature of 850 °C, 800 °C, 790 °C, 780 °C, 770 °C, and 760 °C. The molar ratio of NaCl to MoO₃ is fixed to 20, and the gas flow rate is fixed to 50 sccm. Scale bars: 100 μm for the OM under 20X objective, 10 μm for the small area under 50X objective.

C. Turbulent flow.

In most cases, laminar gas flow is desirable, but some local areas of turbulent flow may exist. Generally, conventional CVD that uses tubular horizontal reactors possesses laminar flow

under any conditions. A high pressure or flow rate is required to produce turbulent flow in the reactor, which exceeds the flow rate used in CVD. To induce turbulent flow, some form of physical disturbance of the gas flow is needed in the CVD setup. In this work, we introduce the confined space to CVD to induce the turbulent flow in the inner tube. This design creates a circumfluent flow, wherein the gas flows around the chamber and backflows toward the substrate. This significantly reduces the gas flow velocity on the substrate to create an unsteady gas flow.

It is reported that the turbulent flow indeed causes the different growth conditions in CVD process. The turbulent flow has been adopted for the synthesis of 2D materials, such as graphene and TMDCs, which effectively changes the morphology of the products [*Science* **267**, 222-225 (1995); *ACS Omega* **7**, 39362-39369 (2022); *Nat. Commun.* **12**, 2391 (2021)]. The components of the velocity vector V_x , V_y , and V_z in this confined space are shown in Fig. R10. As noticed, V_x shows a positive value (Fig. R10e), while V_y and V_z exhibit opposite gas flow directions (Fig. R10f-g). These results indicate that the backflow gas is generated around the substrate. In other words, no extreme conditions are required to generate the turbulent flow in the inner tube. The turbulent flow is supposed to enhance the MoS₂ nucleation in the carrier gas because backflows increase the collision rate of molecules to the substrate by producing enough energy for the twist nucleation, resulting in an alteration of the synthesis conditions.

We believe that the introduced backflow gas through the inner tube drives the entire inner tube out of equilibrium to reduce the chance to form more stable 0°- and 60°-TB-MoS₂, which is expected to be the most important reason for the synthesis of TB-MoS₂.

Therefore, to address the reviewer's concern, we have added the explanation and discussion on the mechanism in the revised manuscript and Supplementary Information.

Changes to the revised manuscript are shown below (In Page 18-21).

The combination of NaCl and confined space promotes the synthesis of the TB-MoS₂ (Supplementary Fig. 23). Introducing NaCl can affect the thermodynamics and kinetics in the CVD system (the detailed analysis can be found in Supplementary Note 5).

Here, introducing a confined space provides controllability of kinetics and consequently realizes CVD anisotropic growth of the TB-MoS₂ with the thermodynamically unfavourable

twisted stacking structures. The introduction of the confined space can facilitate the synthesis of TB-MoS₂ by optimize the growth kinetics conditions of CVD system including gas velocity, temperature, and turbulent flow. First, the gas velocity can be decreased by the confined space (Supplementary Note 6, Supplementary Fig. 14, and Supplementary Fig. 27). As we know, the gas flow condition might be an important parameter affecting the sample state in CVD system [*Nature* **556**, 355-359 (2018)]. The low gas velocity in the confined space results in the nuclei surplus with large nuclei. Therefore, the bilayer MoS₂ can be easily synthesized and also be verified by the proportion of bilayer MoS₂ with the increase in gas flow rate (Supplementary Fig. 14 and Supplementary Fig. 26). The high proportion of bilayer MoS₂ (77.6% and 60.9%) can be achieved under the gas flow of 30 sccm and 50 sccm, respectively. Higher or lower gas flow rates result in a high proportion of multi-layer MoS₂. Second, with the introduction of the confined space, the temperature is very uniform in the confined space near the SiO₂/Si substrate (Supplementary Note 6 and Supplementary Fig. 27). The confined space in the inner tube served as a “heat insulation layer” to prevent the heat diffusion, which benefits the stable synthesis of highly repetitive TB-MoS₂. In this growth stage, the in-plane growth is generally faster than out-of-plane growth, resulting in the smaller size of bilayer MoS₂ than that of the monolayer (Supplementary Fig. 20).

Most importantly, the turbulent flow change with the introduction of the confined space. Based on the previous study, the turbulent flow indeed causes the different growth conditions in CVD process, which has been adopted to synthesize 2D materials, such as graphene and TMDCs with different morphologies [*Science* **267**, 222-225 (1995); *ACS Omega* **7**, 39362-39369 (2022); *Nat. Commun.* **12**, 2391 (2021)]. As shown in Supplementary Fig. 27, the components of the velocity vector (V_y and V_z) exhibit opposite gas flow directions in the confined space compared to V_x , indicating that the backflow gas is generated around the substrate. The turbulent flow is supposed to enhance the MoS₂ nucleation in the carrier gas because that backflows increase the collision rate of molecules to the substrate and thus produces enough energy for the *in-situ* twist nucleation. As a result, there is an alteration of the synthesis conditions. The experimental results (DF-OM) in Supplementary Fig. 4 and Supplementary Fig. 25 show that the MoS₂ layer grows with different rates while sharing the same nucleation site. We believe that the introduced backflow gas through the inner tube drives the entire inner tube out of equilibrium, resulting in the reduced chance to form more stable 0°- and 60°-TB-MoS₂. Therefore, in the high-temperature CVD process, the medium and large nucleus sizes are beneficial for the growth of bilayer MoS₂. Based on the twist

nucleation process, we realize that the reconfiguring nucleation under the assistance of NaCl and confined space is the key for the successful synthesis of TB-MoS₂ by CVD. In addition, it is noteworthy that the proposed CVD technique has been utilized to synthesize other TB-TMDCs. Supplementary Fig. 28 presents the successful growth of TB-WS₂ using comparable procedures.

Changes to the revised Supplementary Information are shown below (In Page 11-13).

Supplementary Note 6. Kinetics analysis of confined space.

To further clarify the effect of the confined space, we have carried out the computational fluid dynamics simulations of the confined space in CVD. The velocity distribution, temperature distribution, and components of the velocity vector for this CVD setup are shown in Supplementary Fig. 27. Here, we believe that the confined space will change the gas flow rate, the temperature, and the gas flow direction. The corresponding detail discussion are shown below.

(1) Gas flow rate.

It is observed from the velocity distribution that the gas flow condition affects the sample state in the open tube system (Supplementary Fig. 14 and Supplementary Fig. 27). It can be identified that the gas velocity in the inner tube (without sealed-end) is one order of magnitude less than that of the outer tube, indicating that there is a change in the gas fluid velocity on the SiO₂/Si substrate. With the decrease in gas flow rate, the fluid velocity of carrier gas decreases, making the nuclei surplus. As a result, the nuclei will grow larger, which is consistent with the experimental results (Supplementary Fig. 21). As we know, small change in the flow regime in CVD significantly influences the reaction products. Although the gas velocity in the inner tube has been greatly reduced, there is a nonnegligible increase. Therefore, there is a rapid change in the yield and density of TB-MoS₂, indicating that the gas flow rate is important in the synthesis process, which mainly affects the nucleation and growth of TB-MoS₂.

(2) Temperature.

Temperature is another key parameter for CVD process, wherein the different reaction temperatures may result in different morphology [*ACS Nano* **12**, 635-643 (2018); *Nat Commun* **10**, 598 (2019); *Nat Commun* **12**, 809 (2021); *ACS Nano* **13**, 8265-8274 (2019)].

Based on previous reports, the preparation of MoS₂ usually requires a relatively high reaction temperature from 600 °C to 1000 °C [*Nature* **556**, 355–359 (2018); *Nat Commun* **10**, 598 (2019); *Nature* **605**, 69-75 (2022)]. However, based on our experimental results, we find that the TB-MoS₂ can only be well synthesized under a narrow reaction temperature range from 770 °C to 790 °C. As shown in Supplementary Fig. 17, TB-MoS₂ can be obtained under a reaction temperature of 780 °C. Higher or lower temperatures result in the dramatic decrease in the yield of TB-MoS₂. For example, large-area monolayer MoS₂ is synthesized under a reaction temperature of 760 °C.

Temperature is the key to affect the thermodynamics and dynamics of the reaction system. Here, we have carried out the simulation on the temperature in the tube, as shown in Supplementary Fig. 27c-d. As noticed, due to the effect of the gas flow rate, the temperature shows an uneven distribution. However, with the introduction of the confined space, there is a uniform temperature distribution near the SiO₂/Si substrate. The inner tube serves as a “heat insulation layer” to prevent the heat diffusion. Thus, we believe that the confined space plays the key role to stabilize the substrate temperature.

(3) Turbulent flow.

In most cases, laminar gas flow is desirable, but some local areas of turbulent flow may exist. Generally, conventional CVD that uses tubular horizontal reactors possesses laminar flow under any conditions. A high pressure or flow rate is required to produce turbulent flow in the reactor, which exceeds the flow rate used in CVD. To induce turbulent flow, some form of physical disturbance of the gas flow is needed in the CVD setup. In this work, we introduce the confined space to CVD to induce the turbulent flow in the inner tube. This design creates a circumfluent flow, wherein the gas flows around the chamber and backflows toward the substrate. This significantly reduces the gas flow velocity on the substrate to create an unsteady gas flow.

It is reported that the turbulent flow indeed causes the different growth conditions in CVD process. The turbulent flow has been adopted for the synthesis of 2D materials, such as graphene and TMDCs, which effectively changes the morphology of the products [*Science* **267**, 222-225 (1995); *ACS Omega* **7**, 39362-39369 (2022); *Nat. Commun.* **12**, 2391 (2021)]. The components of the velocity vector V_x , V_y , and V_z in this confined space are shown in Supplementary Fig. 27. As noticed, V_x shows a positive value (Supplementary Fig. 27e),

while V_y and V_z exhibit opposite gas flow directions (Supplementary Fig. 27f-g). These results indicate that the backflow gas is generated around the substrate. In other words, no extreme conditions are required to generate the turbulent flow in the inner tube. The turbulent flow is supposed to enhance the MoS₂ nucleation in the carrier gas because backflows increase the collision rate of molecules to the substrate by producing enough energy for the twist nucleation, resulting in an alteration of the synthesis conditions. We believe that the introduced backflow gas through the inner tube drives the entire inner tube out of equilibrium to reduce the chance to form more stable 0°- and 60°-TB-MoS₂, which is expected to be the most important reason for the synthesis of TB-MoS₂.

Supplementary Fig. 14 Velocity distribution of the space-confined CVD setup under different gas flow rates. a The velocity distribution with the small inner-tube under gas flow rates of 10 to 110 sccm, respectively. **b** The enlarged velocity distributions of a.

Supplementary Fig. 17 Typical BF-OM of the synthesized products under different reaction temperatures. a-e Typical OM images of the synthesized MoS₂ under different reaction temperature of 850 °C, 800 °C, 790 °C, 780 °C, 770 °C, and 760 °C. The molar ratio of NaCl to MoO₃ is fixed to 20, and the gas flow rate is fixed to 50 sccm. Scale bars: 100 μm for the OM under 20X objective, 10 μm for the small area under 50X objective.

Supplementary Fig. 27 Velocity distribution, temperature distribution, and velocity vector maps of the CVD setup. a-d Velocity and temperature distribution of the CVD setup with and without confined space, respectively. **e-g** Velocity vector maps of the components of the velocity vector of V_x , V_y , and V_z , respectively. All the results were carried out with a gas flow rate of 50 sccm and a temperature of 780 °C.

Comment 6:

The flake size of the as-grown bilayer MoS₂ should be clarified when talking about the density of the flakes.

Response:

Thanks for the valuable comment and suggestion. The typical BF-OMs of synthesized TB-

MoS₂ under different gas flow rate and different molar ratio of NaCl to MoO₃ are shown in Fig. R6 and Fig. R7. As noticed, all the TB-MoS₂ samples with a uniform bottom-layer size in bilayer MoS₂ (as shown in Fig. R6b-e, Fig. R7b-c). To further clarify the flake size, we have counted the flake size of the bottom-layer and top-layer in the bilayer TB-MoS₂ sample.

The frequency distribution histograms of bottom-layer size, top-layer size, and bottom-top-layer size ratio are shown in Fig. R13. The gamma distribution fitting curve can well fit all the frequency distribution plots, which is similar to the size of our CVD-synthesized WTe₂ NRs [*J. Am. Chem. Soc.* **143**, 18103-18113 (2021)]. The average flake size and standard deviation of the bottom-layer in TB-MoS₂ are calculated to be 28.56 μm and 7.43 μm , respectively, which indicates that the size distributes mainly in ~ 30 μm . This concentrated distribution indicated that our statistics on density are reliable. The flake size of the top-layer in TB-MoS₂ can be calculated to be 9.00 μm , and the bottom-top-layer size ratio in TB-MoS₂ shows a relatively concentrated distribution around 0.33, indicating the uniformity of the samples.

Fig. R6 Typical BF-OM of synthesized products under different gas flow rate. a-f The typical OM synthesized under different gas flow rate of 110, 90, 70, 50, 30, and 10 sccm. Scale bars: 100 μm for the OM under 20X objective, 10 μm for the small area under 50X objective. The TB-MoS₂ was synthesized with fixed NaCl to MoO₃ of 20.

Fig. R7 Typical BF-OM of synthesized products under different molar ratio of NaCl to MoO₃. a-e The typical OM images of the synthesized MoS₂ under different molar ratio of NaCl to MoO₃ of 30, 20, 10, 3, and 0. Scale bars: 100 μm for the OM under 20X objective, 10 μm for the small area under 50X objective.

Fig. R13 Frequency distribution histograms of flake size of TB-MoS₂ and corresponding gamma distribution fitting curve (with the mean value (E) and standard deviation (σ)). a Flake size of bottom-layer in TB-MoS₂. b Flake size of top-layer in TB-MoS₂. c The bottom-top-layer size ratio in TB-MoS₂.

Changes to the revised manuscript are shown below (In Page 17).

The density and yield are adopted to evaluate the parameters of the CVD method on TB-MoS₂. All the TB-MoS₂ shows a concentrated distribution with a bottom layer, which ensures a more reliable statistic results on density (Supplementary Fig. 20). As mentioned in Supplementary Note 4, the density is defined as the number of TB-MoS₂ per unit area, while the yield is dedicated to the proportion of TB-MoS₂ in the bilayer MoS₂.

Changes to the revised Supplementary Information are shown below (In Page 34).

The frequency distribution histograms of bottom-layer size, top-layer size, and bottom-top-layer size ratio are shown in Supplementary Fig. 20. The gamma distribution fitting curve can well fit all the frequency distribution plots. The average flake size and standard deviation of the bottom-layer in TB-MoS₂ are calculated to be 28.56 μm and 7.43 μm , respectively, which indicates that the size distributes mainly in ~ 30 μm . This concentrated distribution indicated that our statistics on density are reliable. The flake size of the top-layer in TB-MoS₂ can be calculated to be 9.00 μm , and the bottom-top-layer size ratio in TB-MoS₂ shows a relatively concentrated distribution around 0.33, indicating the uniformity of the samples.

Supplementary Fig. 20 Frequency distribution histograms of flake size of TB-MoS₂ and corresponding gamma distribution fitting curve (with the mean value (E) and standard deviation (σ)). a Flake size of bottom-layer in TB-MoS₂. **b** Flake size of top-layer in TB-MoS₂. **c** The bottom-top-layer size ratio in TB-MoS₂.

Response to Reviewer 2

This manuscript reports a NaCl-assisted space-confined CVD strategy for directly growing twisted bilayer MoS₂. The authors claim that this method can break the thermodynamically unfavourable and achieve entire twist angles from 0 to 120°. Twisted bilayer 2D materials have attracted a lot of interest in recent years, and the development of bottom-up strategies for growing these metastable or thermodynamically unfavourable materials with controllable twist angles is important for this field. There is no evidence in this manuscript for control over the twist angles of as-grown TB-TMDCs. Nevertheless, the achieved wide range of twist angles and the breaking of the thermodynamically unfavourable stacking order (How and why?) would be a step forward to controllable growth of TB-TMDCs, especially for twist angle control. The study is quite carefully conducted and should be of interest to experts in the field, and can therefore be considered for publication in Nature Communications. However, the paper contains several weak points and issues that are mandatory to be addressed conclusively before acceptance may be considered. A list of the most urgent issues follows below:

Response:

We appreciate for the reviewer' positive comments on the quality of our work and explicit recommendation of publication. We have addressed the reviewer's comments point by point as follows.

Comment 1:

The term “Reconfiguring nucleation” in the title seems like a proposed strategy, but there is no corresponding statement in the main text. What's the relationship between the “Reconfiguring nucleation” and the proposed strategy “NaCl-assisted space-confined CVD”?

Response:

We appreciate the reviewer's comments and we apologize for our ambiguous expressions. Indeed, “reconfiguring nucleation” is the strategy we proposed, while “NaCl-assisted space-confined CVD” is the method for implementing of the strategy. The assistance of NaCl and a confined space in CVD system are beneficial for CVD growth of the TB-MoS₂ with the

thermodynamically unfavourable twisted stacking structures (synthesis of non-twisted structure such as 0° or 60° -TB-TMDCs), and the TB-TMDCs can be well synthesized by the reconfiguring nucleation in CVD process. Accordingly, we have clarified corresponding expressions of “reconfiguring nucleation” and “NaCl-assisted space-confined CVD” in the revised manuscript.

Changes to the revised manuscript are shown below (In Page 2, Page 4, and Page 21).

Here, we proposed a reconfiguring nucleation strategy by introducing NaCl and a confined space to the chemical vapor deposition (CVD) system for the direct synthesis of thermodynamically unfavourable twisted stacking structures MoS_2 with a wide twist angle from 0° to 120° .

The proposed reconfiguring nucleation approach has been extended to synthesize high-quality TB- WS_2 , opening an avenue for the precise growth of TB-TMDCs, which has great significance for both fundamental research and practical applications.

Here, we propose a reconfiguring nucleation strategy for the direct synthesis of clean TB-TMDCs with thermodynamically unfavourable twisted stacking structures.

Based on the twist nucleation process, we realize the reconfiguring nucleation under the assistance of NaCl and confined space is the key for the successful synthesis of TB- MoS_2 by CVD.

In summary, the reconfiguring nucleation strategy is adopted for the direct synthesis of TB- MoS_2 by CVD with twist angles ranging from 0° to 120° , especially some smaller twist angles close to 0° (120°) and 60° .

Comment 2:

The term “entire twist angles” is oversold. One cannot get all the angles in one paper. The authors didn’t should replace it, maybe with “wide range of twist angles”.

Response:

Thank you for your comment. We have changed the “entire twist angles” into “wide range of

twist angles” in the revised manuscript and Supplementary Information.

Comment 3:

The authors paid a lot of attention to characterization but neglected the growth method itself. It is reported by many literatures that halogen salts and the confined space play crucial roles in the CVD growth of TMDCs. How does this combination (NaCl and confined space) break the thermodynamically unfavourable? Both experimental results and theoretical analysis should be strengthened to prove this hypothesis.

Response:

Thank you for raising this insightful concern. Firstly, we apologize for the incurred misunderstanding. In our previous works, we want to point out that the introduction of NaCl and the confined space in CVD system is benefit for CVD growth of the TB-MoS₂ with the thermodynamically unfavourable twisted stacking structures. However, intended meaning of our original statement in the manuscripts was not clearly due to the misleading sentence of “*NaCl-assisted space-confined CVD process to break the thermodynamically unfavorable for the direct synthesis of TB-MoS₂*”. Therefore, we have changed the corresponding description in the revised manuscript. Based on your suggestion, we have carried out additional experiments and theoretical analysis to demonstrate the thermodynamically and kinetics effects on CVD synthesis of TB-MoS₂ with the assistant of NaCl and confined space. The point-by-point response has been listed below.

(1) Experimental results on combination of NaCl and confined space.

Based on the reviewer’s suggestion, we have carried out a serious of experiments on the combination of NaCl and confined space. Specifically, four combinations of NaCl and confined space have been repeatedly conducted, including the combination of without NaCl and with confined space, with NaCl and with confined space, without NaCl and without confined space, as well as with NaCl and without confined space. The typical OMs of the results is shown in Fig. R14. NaCl is the important parameter for the synthesis of MoS₂, and the MoS₂ cannot be well synthesized without NaCl in CVD process, as shown in Fig. R14a and Fig. R14c. As noticed, the 2D quadrilateral shape or 1D wires shape MO_x (MoO₃ and MoO₂) was synthesized without NaCl. While the MoS₂ can be well synthesized with the NaCl,

as shown in Fig. R14c and Fig. R14d. The TB-MoS₂ is successfully synthesized under the confined space. In contrast, the CVD without confined space yielded only monolayer crystals, suggesting that confined space is a key for controlling the TB-MoS₂ of the CVD products. Therefore, we believe that the combination of the NaCl and the confined space will affect the thermodynamically and kinetics of the CVD process. The detail discussions are shown in below.

Fig. R14 Typical BF-OM of synthesized products with/without NaCl and confined space.
a Without NaCl, with confined space. **b** With NaCl, with confined space. **c** Without NaCl, without confined space. **d** With NaCl, without confined space. Scale bars: 100 μm for the OM under 20X objective, 10 μm for the small area under 50X objective.

(2) The effect of NaCl.

In the previous works, there are detailed discussion for the influence of the salt on the CVD growth of TMDCs via thermodynamics and kinetics [*Nature* **556**, 355-359 (2018); *J. Am. Chem. Soc.* **143**, 18103-18113 (2021)]. Here, we briefly discussed the vapor pressure of reaction precursors. As shown in Fig. R15, the vapor pressure of NaCl and MoO₃ can be found around ~10 Pa and ~10⁻⁹ Pa under 780 °C (1053 K), respectively. Considering that the solubility of metal oxide is at the order of ppm, the metal oxide and salt will dramatically increase the vapor pressure of metal precursors by a few orders.

[REDACTED]

Fig. R15 Temperature dependence of vapor pressure of transitional metal oxides and salt.

(Taken from Supplementary Fig. 35 in [*Nature* **556**, 355-359 (2018)])

In terms of affecting thermodynamics, NaCl can be reacted with MoO₃ to form MoO_xCl_y. Notably, the degrees of freedom changed by reacting with MoO_xCl_y, resulting in the change of entropy. Compared with the CVD system without NaCl, the participation of MoO_xCl_y may affect the Gibbs free energy ($\Delta G = \Delta H - T\Delta S$) by neglecting the change of pressure and volume.

While for the kinetics process, NaCl will affect the nucleation process and dominate the geometries of MoS₂ layers. The nucleation rate with and without NaCl can be roughly written as:

$$\frac{\dot{N}(MoO_3 + NaCl)}{\dot{N}(MoO_3)} \sim \frac{P_i(NaCl)}{P_i(\text{without NaCl})} \quad (3)$$

where \dot{N} is the nucleation rate, P_i is partial pressure of precursors. From this equation, we can clearly find that the partial pressure P_i is the dominate factor for the nucleation rate. As mentioned before, the vapor pressure of NaCl is much higher than that of MoO₃ precursors. Besides, in case of reaction between MoO₃ and NaCl, MoO_xCl_y will be formed, resulting in a high P_i due to the high volatility nature of MoO_xCl_y. Therefore, the NaCl will result in a higher nucleation rate.

Fig. R7 Typical BF-OM of synthesized products under different molar ratio of NaCl to MoO₃. a-e The typical OM images of the synthesized MoS₂ under different molar ratio of NaCl to MoO₃ of 30, 20, 10, 3, and 0. Scale bars: 100 μm for the OM under 20X objective, 10 μm for the small area under 50X objective.

Without the addition of NaCl, the MoO₃ precursor is evaporated under the temperature. As a result, quadrilateral shapes MoO₃ and MoO₂ preferentially nucleate and grow on the SiO₂/Si substrate (Fig. R7). Based on the previous discussion, we can notice that the solubility of metal oxide is at the order of ppm. Therefore, the reactant concentrations of MoO_xCl_y were greatly affected by the molar ratio of NaCl to MoO₃ at the definite reaction temperature. With the increase of NaCl to MoO₃ molar ratio, the reactant concentrations of MoO_xCl_y increased. Therefore, increasing the molar ratio of NaCl to MoO₃ makes the nuclei surplus grow larger,

resulting in the arising number of bilayer MoS₂ and the monolayer concomitant on SiO₂/Si, as shown in Fig. R7b-c. Besides, the thickness of MoS₂ under the different molar ratio of NaCl to MoO₃ is shown in Fig. R16, indicating that the high proportion of bilayer MoS₂ (60.9%) can be obtained under high molar ratio of NaCl to MoO₃ (20). With the introduction of the confined space, the growth kinetics was beneficial for CVD growth of the TB-MoS₂ with the thermodynamically unfavourable twisted stacking structures. Therefore, The TB-MoS₂ can be synthesized under the suitable molar ratio of NaCl to MoO₃.

Fig. R16 Proportion of monolayer MoS₂, bilayer MoS₂, multilayer MoS₂ and non-MoS₂ in as-grown samples under different molar ratio of NaCl to MoO₃, respectively.

(3) The effect of confined space.

To further clarify the effect of the confined space, we have carried out the computational fluid dynamics simulations of the confined space in CVD. The velocity distribution, temperature distribution, and components of the velocity vector for this CVD setup are shown in Fig. R10. Here, we believe that the confined space will change the gas flow rate, the temperature, and the gas flow direction. The corresponding detail discussion are shown below.

A. Gas flow rate.

It is observed from the velocity distribution that the gas flow condition affects the sample state in the open tube system (Fig. R10 and Fig. R11). It can be identified that the gas velocity in the inner tube (without sealed-end) is one order of magnitude less than that of the outer tube, indicating that there is a change in the gas fluid velocity on the SiO₂/Si substrate. With the decrease in gas flow rate, the fluid velocity of carrier gas decreases, making the nuclei surplus. As a result, the nuclei will grow larger, which is consistent with the experimental results (Fig. R6). As we know, small change in the flow regime in CVD significantly

influences the reaction products. Although the gas velocity in the inner tube has been greatly reduced, there is a nonnegligible increase. Therefore, there is a rapid change in the yield and density of TB-MoS₂, indicating that the gas flow rate is important in the synthesis process, which mainly affects the nucleation and growth of TB-MoS₂.

Fig. R10 Velocity distribution, temperature distribution, and velocity vector maps of the CVD setup. a-d Velocity and temperature distribution of the CVD setup with and without confined space, respectively. **e-g** Velocity vector maps of the components of the velocity vector of V_x , V_y , and V_z , respectively. All the results were carried out with a gas flow rate of 50 sccm and a temperature of 780 °C.

Fig. R6 Typical BF-OM of synthesized products under different gas flow rate. a-f The typical OM synthesized under different gas flow rate of 110, 90, 70, 50, 30, and 10 sccm. Scale bars: 100 μm for the OM under 20X objective, 10 μm for the small area under 50X objective. The TB-MoS₂ was synthesized with fixed NaCl to MoO₃ of 20.

B. Temperature.

Temperature is another key parameter for CVD process, wherein the different reaction temperatures may result in different morphology [ACS Nano 12, 635-643 (2018); Nat Commun 10, 598 (2019); Nat Commun 12, 809 (2021); ACS Nano 13, 8265-8274 (2019)]. Based on previous reports, the preparation of MoS₂ usually requires a relatively high reaction temperature from 600 °C to 1000 °C [Nature 556, 355-359 (2018); Nat Commun 10, 598 (2019); Nature 605, 69-75 (2022)]. However, based on our experimental results, we find that the TB-MoS₂ can only be well synthesized under a narrow reaction temperature range from 770 °C to 790 °C. As shown in Fig. R12, TB-MoS₂ can be obtained under a reaction temperature of 780 °C. Higher or lower temperatures result in the dramatic decrease in the yield of TB-MoS₂. For example, large-area monolayer MoS₂ is synthesized under a reaction temperature of 760 °C.

Temperature is the key to affect the thermodynamics and dynamics of the reaction system. Here, we have carried out the simulation on the temperature in the tube, as shown in Fig.

R10c-d. As noticed, due to the effect of the gas flow rate, the temperature shows an uneven distribution. However, with the introduction of the confined space, there is a uniform temperature distribution near the SiO₂/Si substrate. The inner tube serves as a “heat insulation layer” to prevent the heat diffusion. Thus, we believe that the confined space plays the key role to stabilize the substrate temperature.

Fig. R11 Velocity distribution of the space-confined CVD setup under different gas flow rates. a The velocity distribution with the small inner-tube under the gas flow rates of 10 to 110 sccm, respectively. **b** The enlarged velocity distributions of **a**.

Fig. R12 Typical BF-OM of the synthesized products under different reaction temperatures. a-e Typical OM images of the synthesized MoS₂ under different reaction temperature of 850 °C, 800 °C, 790 °C, 780 °C, 770 °C, and 760 °C. The molar ratio of NaCl to MoO₃ is fixed to 20, and the gas flow rate is fixed to 50 sccm. Scale bars: 100 μm for the OM under 20X objective, 10 μm for the small area under 50X objective.

C. Turbulent flow.

In most cases, laminar gas flow is desirable, but some local areas of turbulent flow may exist. Generally, conventional CVD that uses tubular horizontal reactors possesses laminar flow under any conditions. A high pressure or flow rate is required to produce turbulent flow in the reactor, which exceeds the flow rate used in CVD. To induce turbulent flow, some form of physical disturbance of the gas flow is needed in the CVD setup. In this work, we introduce the confined space to CVD to induce the turbulent flow in the inner tube. This design creates a circumfluent flow, wherein the gas flows around the chamber and backflows toward the substrate. This significantly reduces the gas flow velocity on the substrate to create an unsteady gas flow.

It is reported that the turbulent flow indeed causes the different growth conditions in CVD process. The turbulent flow has been adopted for the synthesis of 2D materials, such as graphene and TMDCs, which effectively changes the morphology of the products [Science

267, 222-225 (1995); *ACS Omega* 7, 39362-39369 (2022); *Nat. Commun.* 12, 2391 (2021)]. The components of the velocity vector V_x , V_y , and V_z in this confined space are shown in Fig. R10. As noticed, V_x shows a positive value (Fig. R10e), while V_y and V_z exhibit opposite gas flow directions (Fig. R10f-g). These results indicate that the backflow gas is generated around the substrate. In other words, no extreme conditions are required to generate the turbulent flow in the inner tube. The turbulent flow is supposed to enhance the MoS₂ nucleation in the carrier gas because backflows increase the collision rate of molecules to the substrate by producing enough energy for the twist nucleation, resulting in an alteration of the synthesis conditions. We believe that the introduced backflow gas through the inner tube drives the entire inner tube out of equilibrium to reduce the chance to form more stable 0°- and 60°-TB-MoS₂, which is expected to be the most important reason for the synthesis of TB-MoS₂.

Therefore, to address the reviewer's concern, we have added the explanation and discussions on the mechanism in the revised manuscript and Supplementary Information.

Changes to the revised manuscript are shown below (In Page 18-21).

The combination of NaCl and confined space promotes the synthesis of the TB-MoS₂ (Supplementary Fig. 23). Introducing NaCl can affect the thermodynamics and kinetics in the CVD system (the detailed analysis can be found in Supplementary Note 5). As shown in Fig. 5d, the MoO₃ is insufficient for the reaction without adding NaCl under the low sublimation rate of the sulfur. The MoO₃ precursor is evaporated under the temperature. As a result, quadrilateral shapes MoO₃ and MoO₂ preferentially nucleate and grow on the SiO₂/Si substrate (Fig. 5d, Supplementary Fig. 24, and Supplementary Fig. 25). In contrast, the added NaCl can be reacted with the MoO₃, forming the oxychloride compounds MoO_xCl_y and enhanced the concentration and the precursor vapor pressure [*Nature* 556, 355-359 (2018); *J. Am. Chem. Soc.* 143, 18103-18113 (2021)]. The resultant compounds further react with the sulfur, resulting in the growth of triangular shapes MoS₂ crystal on SiO₂/Si substrate. The reactant concentrations of MoO_xCl_y are greatly affected by the molar ratio of NaCl to MoO₃ at the definite reaction temperature. With a low salt ratio, nuclei with small size are obtained, and consequently monolayer MoS₂ is grown on SiO₂/Si substrate (Fig. 5d and Supplementary Fig. 25). With the increase in the molar ratio of NaCl to MoO₃, the reactant concentrations of MoO_xCl_y increases, making the nuclei surplus grow larger. As a result, there is a rising number of bilayer MoS₂ and the monolayer concomitant on SiO₂/Si, and some TB-MoS₂ can

be observed (Fig. 5d and Supplementary Fig. 25). When the molar ratio of NaCl to MoO₃ further increases, the nuclei size becomes even larger, leading to the disappearance of monolayer MoS₂. In contrast, the bilayer and even multilayer MoS₂ are obtained. The proportion of MoS₂ with different layer numbers under different molar ratio of NaCl to MoO₃ is shown in Supplementary Fig. 26. The highest proportion of bilayer MoS₂ (60.9%) can be obtained under a molar ratio of 20, and the excessive NaCl can result in the growth of multilayer MoS₂. Therefore, the TB-MoS₂ can be synthesized under the suitable molar ratio of NaCl to MoO₃. Based on the calculation results, the 0°-TB-MoS₂ and 60°-TB-MoS₂ have the lowest energy, which has been widely synthesized by the CVD method [*Nat. Commun.* **5**, 4966 (2014); *Nano Lett.* **14**, 3869-3875 (2014)]. Therefore, the TB-MoS₂ is hardly grown via a conventional thermodynamic growing method.

Here, introducing a confined space provides controllability of kinetics and consequently realizes CVD anisotropic growth of the TB-MoS₂ with the thermodynamically unfavourable twisted stacking structures. The introduction of the confined space can facilitate the synthesis of TB-MoS₂ by optimize the growth kinetics conditions of CVD system including gas velocity, temperature, and turbulent flow. First, the gas velocity can be decreased by the confined space (Supplementary Note 6, Supplementary Fig. 14, and Supplementary Fig. 27). As we know, the gas flow condition might be an important parameter affecting the sample state in CVD system [*Nature* **556**, 355-359 (2018)]. The low gas velocity in the confined space results in the nuclei surplus with large nuclei. Therefore, the bilayer MoS₂ can be easily synthesized and also be verified by the proportion of bilayer MoS₂ with the increase in gas flow rate (Supplementary Fig. 14 and Supplementary Fig. 26). The high proportion of bilayer MoS₂ (77.6% and 60.9%) can be achieved under the gas flow of 30 sccm and 50 sccm, respectively. Higher or lower gas flow rates result in a high proportion of multi-layer MoS₂. Second, with the introduction of the confined space, the temperature is very uniform in the confined space near the SiO₂/Si substrate (Supplementary Note 6 and Supplementary Fig. 27). The confined space in the inner tube served as a “heat insulation layer” to prevent the heat diffusion, which benefits the stable synthesis of highly repetitive TB-MoS₂. In this growth stage, the in-plane growth is generally faster than out-of-plane growth, resulting in the smaller size of bilayer MoS₂ than that of the monolayer (Supplementary Fig. 20).

Most importantly, the turbulent flow change with the introduction of the confined space. Based on the previous study, the turbulent flow indeed causes the different growth conditions in CVD process, which has been adopted to synthesize 2D materials, such as graphene and

TMDCs with different morphologies [*Science* **267**, 222-225 (1995); *ACS Omega* **7**, 39362-39369 (2022); *Nat. Commun.* **12**, 2391 (2021)]. As shown in Supplementary Fig. 27, the components of the velocity vector (V_y and V_z) exhibit opposite gas flow directions in the confined space compared to V_x , indicating that the backflow gas is generated around the substrate. The turbulent flow is supposed to enhance the MoS₂ nucleation in the carrier gas because that backflows increase the collision rate of molecules to the substrate and thus produces enough energy for the *in-situ* twist nucleation. As a result, there is an alteration of the synthesis conditions. The experimental results (DF-OM) in Supplementary Fig. 4 and Supplementary Fig. 25 show that the MoS₂ layer grows with different rates while sharing the same nucleation site. We believe that the introduced backflow gas through the inner tube drives the entire inner tube out of equilibrium, resulting in the reduced chance to form more stable 0°- and 60°-TB-MoS₂. Therefore, in the high-temperature CVD process, the medium and large nucleus sizes are beneficial for the growth of bilayer MoS₂. Based on the twist nucleation process, we realize that the reconfiguring nucleation under the assistance of NaCl and confined space is the key for the successful synthesis of TB-MoS₂ by CVD. In addition, it is noteworthy that the proposed CVD technique has been utilized to synthesize other TB-TMDCs. Supplementary Fig. 28 presents the successful growth of TB-WS₂ using comparable procedures.

Changes to the revised Supplementary Information are shown below (In Page 10-13).

Supplementary Note 5. Thermodynamically and kinetics analysis of NaCl.

In our previous works, we have discussed the effects of salt on the CVD growth of TMDCs via thermodynamics and kinetics [*Nature* **556**, 355-359 (2018); *J. Am. Chem. Soc.* **143**, 18103-18113 (2021)]. In terms of affecting thermodynamics, NaCl can be reacted with MoO₃ for the synthesis of MoO_xCl_y in first step. Notably, the degrees of freedom changed by reacting with MoO_xCl_y results in the change of entropy. Compared with the CVD system without NaCl, the participation of MoO_xCl_y may affect the Gibbs free energy ($\Delta G = \Delta H - T\Delta S$) by neglecting the change of pressure and volume. For the kinetics process, NaCl will affect the nucleation process and dominate the geometries of MoS₂ layers. The nucleation rate with and without NaCl can be roughly written as:

$$\frac{\dot{N}(\text{MoO}_3 + \text{NaCl})}{\dot{N}(\text{MoO}_3)} \sim \frac{P_i(\text{NaCl})}{P_i(\text{without NaCl})} \quad (13)$$

From this equation, we can find that the partial pressure P_i is the dominate factor for the nucleation rate. As we mentioned before [*Nature* **556**, 355-359 (2018)], the vapor pressure of NaCl is much higher than that of MoO₃ precursors. Besides, the formation of MoO_xCl_y induces a high P_i due to the high volatility nature of MoO_xCl_y. Therefore, the NaCl will result in a higher nucleation rate. The vapor pressure of NaCl and MoO₃ can be found to be around ~10 Pa and ~10⁻⁹ Pa under 780 °C (1053 K), respectively. Considering that the solubility of metal oxide is at the order of ppm, the metal oxide and salt will dramatically increase the vapor pressure of metal precursors of a few orders.

Supplementary Note 6. Kinetics analysis of confined space.

To further clarify the effect of the confined space, we have carried out the computational fluid dynamics simulations of the confined space in CVD. The velocity distribution, temperature distribution, and components of the velocity vector for this CVD setup are shown in Supplementary Fig. 27. Here, we believe that the confined space will change the gas flow rate, the temperature, and the gas flow direction. The corresponding detail discussion are shown below.

(1) Gas flow rate.

It is observed from the velocity distribution that the gas flow condition affects the sample state in the open tube system (Supplementary Fig. 14 and Supplementary Fig. 27). It can be identified that the gas velocity in the inner tube (without sealed-end) is one order of magnitude less than that of the outer tube, indicating that there is a change in the gas fluid velocity on the SiO₂/Si substrate. With the decrease in gas flow rate, the fluid velocity of carrier gas decreases, making the nuclei surplus. As a result, the nuclei will grow larger, which is consistent with the experimental results (Supplementary Fig. 21). As we know, small change in the flow regime in CVD significantly influences the reaction products. Although the gas velocity in the inner tube has been greatly reduced, there is a nonnegligible increase. Therefore, there is a rapid change in the yield and density of TB-MoS₂, indicating that the gas flow rate is important in the synthesis process, which mainly affects the nucleation and growth of TB-MoS₂.

(2) Temperature.

Temperature is another key parameter for CVD process, wherein the different reaction temperatures may result in different morphology [*ACS Nano* **12**, 635-643 (2018); *Nat Commun* **10**, 598 (2019); *Nat Commun* **12**, 809 (2021); *ACS Nano* **13**, 8265-8274 (2019)]. Based on previous reports, the preparation of MoS₂ usually requires a relatively high reaction temperature from 600 °C to 1000 °C [*Nature* **556**, 355–359 (2018); *Nat Commun* **10**, 598 (2019); *Nature* **605**, 69-75 (2022)]. However, based on our experimental results, we find that the TB-MoS₂ can only be well synthesized under a narrow reaction temperature range from 770 °C to 790 °C. As shown in Supplementary Fig. 17, TB-MoS₂ can be obtained under a reaction temperature of 780 °C. Higher or lower temperatures result in the dramatic decrease in the yield of TB-MoS₂. For example, large-area monolayer MoS₂ is synthesized under a reaction temperature of 760 °C.

Temperature is the key to affect the thermodynamics and dynamics of the reaction system. Here, we have carried out the simulation on the temperature in the tube, as shown in Supplementary Fig. 27c-d. As noticed, due to the effect of the gas flow rate, the temperature shows an uneven distribution. However, with the introduction of the confined space, there is a uniform temperature distribution near the SiO₂/Si substrate. The inner tube serves as a “heat insulation layer” to prevent the heat diffusion. Thus, we believe that the confined space plays the key role to stabilize the substrate temperature.

(3) Turbulent flow.

In most cases, laminar gas flow is desirable, but some local areas of turbulent flow may exist. Generally, conventional CVD that uses tubular horizontal reactors possesses laminar flow under any conditions. A high pressure or flow rate is required to produce turbulent flow in the reactor, which exceeds the flow rate used in CVD. To induce turbulent flow, some form of physical disturbance of the gas flow is needed in the CVD setup. In this work, we introduce the confined space to CVD to induce the turbulent flow in the inner tube. This design creates a circumfluent flow, wherein the gas flows around the chamber and backflows toward the substrate. This significantly reduces the gas flow velocity on the substrate to create an unsteady gas flow.

It is reported that the turbulent flow indeed causes the different growth conditions in CVD process. The turbulent flow has been adopted for the synthesis of 2D materials, such as

graphene and TMDCs, which effectively changes the morphology of the products [*Science* **267**, 222-225 (1995); *ACS Omega* **7**, 39362-39369 (2022); *Nat. Commun.* **12**, 2391 (2021)]. The components of the velocity vector V_x , V_y , and V_z in this confined space are shown in Supplementary Fig. 27. As noticed, V_x shows a positive value (Supplementary Fig. 27e), while V_y and V_z exhibit opposite gas flow directions (Supplementary Fig. 27f-g). These results indicate that the backflow gas is generated around the substrate. In other words, no extreme conditions are required to generate the turbulent flow in the inner tube. The turbulent flow is supposed to enhance the MoS₂ nucleation in the carrier gas because backflows increase the collision rate of molecules to the substrate by producing enough energy for the twist nucleation, resulting in an alteration of the synthesis conditions. We believe that the introduced backflow gas through the inner tube drives the entire inner tube out of equilibrium to reduce the chance to form more stable 0°- and 60°-TB-MoS₂, which is expected to be the most important reason for the synthesis of TB-MoS₂.

Supplementary Fig. 14 Velocity distribution of the space-confined CVD setup under different gas flow rates. a The velocity distribution with the small inner-tube under gas flow rates of 10 to 110 sccm, respectively. **b** The enlarged velocity distributions of **a**.

Supplementary Fig. 17 Typical BF-OM of the synthesized products under different reaction temperatures. a-e Typical OM images of the synthesized MoS₂ under different reaction temperature of 850 °C, 800 °C, 790 °C, 780 °C, 770 °C, and 760 °C. The molar ratio of NaCl to MoO₃ is fixed to 20, and the gas flow rate is fixed to 50 sccm. Scale bars: 100 μm for the OM under 20X objective, 10 μm for the small area under 50X objective.

Supplementary Fig. 21 Typical BF-OM of synthesized products under different gas flow rate. a-f The typical OM synthesized under different gas flow rate of 110, 90, 70, 50, 30, and 10 sccm. Scale bars: 100 μm for the OM under 20X objective, 10 μm for the small area under 50X objective. The TB-MoS₂ was synthesized with fixed NaCl to MoO₃ of 20.

Supplementary Fig. 22 Typical BF-OM of synthesized products under different molar ratio of NaCl to MoO₃. **a-e** The typical OM images of the synthesized MoS₂ under different molar ratio of NaCl to MoO₃ of 30, 20, 10, 3, and 0. Scale bars: 100 μm for the OM under 20X objective, 10 μm for the small area under 50X objective.

Supplementary Fig. 23 Typical BF-OM of synthesized products with/without NaCl and confined space. **a** Without NaCl, with confined space. **b** With NaCl, with confined space. **c** Without NaCl, without confined space. **d** With NaCl, without confined space. Scale bars: 100 μm for the OM under 20X objective, 10 μm for the small area under 50X objective.

Supplementary Fig. 27 Velocity distribution, temperature distribution, and velocity vector maps of the CVD setup. a-d Velocity and temperature distribution of the CVD setup with and without confined space, respectively. **e-g** Velocity vector maps of the components of the velocity vector of V_x , V_y , and V_z , respectively. All the results were carried out with a gas flow rate of 50 sccm and a temperature of 780 °C.

Comment 4:

Figure 5b, it seems that the gas flow rate has a negative effect on the yield of TB-MoS₂. Why? What's the relationship between this result and the space-confined method? Does the gas turbulence influence on the breaking of the thermodynamically unfavourable? If the confined space changed, what would be happen to the yield curve as a function of the flow rate?

Response:

Thank you for raising these insightful questions. The point-by-point response has been listed below.

(1) The influence of the gas flow rate to the yield of TB-MoS₂.

Combining the valuable comments and suggestions of reviewer 3 (Comment 9), we have carried out a series of experiment results based on the gas flow rate, as shown in Fig. R6. As noticed, when the gas flow rate is higher than 110 sccm or lower than 10 sccm, we cannot obtain the TB-MoS₂ samples on a substrate, indicating that higher or lower gas flow rates cannot synthesize TB-MoS₂. In other words, the suitable gas flow rate of 30 sccm to 90 sccm would be an enhanced growth range of TB-MoS₂. While the yield and density of TB-MoS₂ increased with the gas flow increased from 10 sccm to 50 sccm, and then decreased from 50 sccm to 110 sccm, and we can get the highest yield and density of 17.2% and 28.9 pieces/mm² (Fig. R17).

Based on the velocity field distribution results, it is observed that the gas flow condition affects the sample state in the open tube system (Fig. R11). It can be identified that the gas velocity in the inner tube (without sealed-end) is one order of magnitude less than that out of the inner tube, indicating that there is a change in the gas fluid field velocity on the SiO₂/Si substrate. With the decrease in gas flow rate, the gas fluid field velocity decreases. Decreasing the flow rate of carrier gas can make the nuclei surplus, and the nuclei will grow large, which is consistent with the experiment results (Fig. R6). As we know, any small change in the flow regime in CVD significantly influences the reaction products. Although the gas velocity in the inner tube has been greatly reduced, there is a nonnegligible increase. Therefore, the rapid change in yield and density of TB-MoS₂ indicates that the gas flow rate is important in the synthesis process, which mainly effects the nucleation and growth of TB-MoS₂.

Fig. R6 Typical BF-OM of synthesized products under different gas flow rate. a-f The typical OM synthesized under different gas flow rate of 110, 90, 70, 50, 30, and 10 sccm. Scale bars: 100 μm for the OM under 20X objective, 10 μm for the small area under 50X objective. The TB-MoS₂ was synthesized with fixed NaCl to MoO₃ of 20.

Fig. R17 Density and yield of TB-MoS₂ under different gas flow rate.

Fig. R11 Velocity distribution of the space-confined CVD setup under different gas flow rates. a The velocity distribution with the small inner-tube under gas flow rates of 10 to 110 sccm, respectively. **b** The enlarged velocity distributions of **a**.

(2) The backflow gas influence on the kinetics.

In most cases, laminar gas flow is desirable, but some local areas of turbulent flow may exist. Generally, conventional CVD that uses tubular horizontal reactors possesses laminar flow under any conditions. A high pressure or flow rate is required to produce turbulent flow in the reactor, which exceeds the flow rate used in CVD. To induce turbulent flow, some form of physical disturbance of the gas flow is needed in the CVD setup. In this work, we introduce the confined space to CVD to induce the turbulent flow in the inner tube. This design creates a circumfluent flow, wherein the gas flows around the chamber and backflows toward the substrate. This significantly reduces the gas flow velocity on the substrate to create an unsteady gas flow.

It is reported that the turbulent flow indeed causes the different growth conditions in CVD process. The turbulent flow has been adopted for the synthesis of 2D materials, such as graphene and TMDCs, which effectively changes the morphology of the products [*Science* **267**, 222-225 (1995); *ACS Omega* **7**, 39362-39369 (2022); *Nat. Commun.* **12**, 2391 (2021)]. The components of the velocity vector V_x , V_y , and V_z in this confined space are shown in Fig. R10. As noticed, V_x shows a positive value (Fig. R10e), while V_y and V_z exhibit opposite gas flow directions (Fig. R10f-g). These results indicate that the backflow gas is generated around the substrate. In other words, no extreme conditions are required to generate the turbulent flow in the inner tube. The turbulent flow is supposed to enhance the MoS₂ nucleation in the carrier gas because backflows increase the collision rate of molecules to the substrate by producing enough energy for the twist nucleation, resulting in an alteration of the synthesis conditions.

We believe that the turbulent flow based kinetic would drive the entire inner tube out of equilibrium to reduce the chance to form more stable 0°- and 60°-TB-MoS₂ due to the backflow gas introduced through the inner tube, which would be the most important reasons for preparation of TB-MoS₂. However, apart from kinetic factors, we believe thermodynamic factors introduced by the NaCl and confined space would be other important parameters for the interlayer twist. The discussion on the influence of NaCl and confined space on the thermodynamically and kinetics was listed in Comment 3.

Fig. R10 Velocity distribution, temperature distribution, and velocity vector maps of the CVD setup. **a-d** Velocity and temperature distribution of the CVD setup with and without confined space, respectively. **e-g** Velocity vector maps of the components of the velocity vector of V_x , V_y , and V_z , respectively. All the results were carried out with a gas flow rate of 50 sccm and a temperature of 780 °C.

(3) The yield curve evolution based on the confined space.

We believe that a confined space is the key factor for the synthesis of TB-MoS₂. As shown in Fig. R9, with the confined space, TB-MoS₂ can be well synthesized. In contrast, only monolayer MoS₂ is observed on SiO₂/Si substrate (Fig. R18) when we remove the confined space, suggesting that confined space is a effective for controlling the TB-MoS₂ of the CVD products.

Fig. R9 The typical BF-OM of products synthesized TB-MoS₂. **a** with and **b** without confined space. Scale bars: 100 μm for the OM under 20X objective, 10 μm for the small area under 50X objective.

Fig. R18 Density and yield of TB-MoS₂ with/without confined space.

Therefore, to address the reviewer's concern, we have added the explanation and discussion on the mechanism in the revised manuscript and Supplementary Information.

Changes to the revised manuscript are shown below (In Page 19-21).

Here, introducing a confined space provides controllability of kinetics and consequently realizes CVD anisotropic growth of the TB-MoS₂ with the thermodynamically unfavourable twisted stacking structures. The introduction of the confined space can facilitate the synthesis of TB-MoS₂ by optimize the growth kinetics conditions of CVD system including gas velocity, temperature, and turbulent flow. First, the gas velocity can be decreased by the confined space (Supplementary Note 6, Supplementary Fig. 14, and Supplementary Fig. 27). As we know, the

gas flow condition might be an important parameter affecting the sample state in CVD system [Nature 556, 355-359 (2018)]. The low gas velocity in the confined space results in the nuclei surplus with large nuclei. Therefore, the bilayer MoS₂ can be easily synthesized and also be verified by the proportion of bilayer MoS₂ with the increase in gas flow rate (Supplementary Fig. 14 and Supplementary Fig. 26). The high proportion of bilayer MoS₂ (77.6% and 60.9%) can be achieved under the gas flow of 30 sccm and 50 sccm, respectively. Higher or lower gas flow rates result in a high proportion of multi-layer MoS₂. Second, with the introduction of the confined space, the temperature is very uniform in the confined space near the SiO₂/Si substrate (Supplementary Note 6 and Supplementary Fig. 27). The confined space in the inner tube served as a “heat insulation layer” to prevent the heat diffusion, which benefits the stable synthesis of highly repetitive TB-MoS₂. In this growth stage, the in-plane growth is generally faster than out-of-plane growth, resulting in the smaller size of bilayer MoS₂ than that of the monolayer (Supplementary Fig. 20).

Most importantly, the turbulent flow change with the introduction of the confined space. Based on the previous study, the turbulent flow indeed causes the different growth conditions in CVD process, which has been adopted to synthesize 2D materials, such as graphene and TMDCs with different morphologies [Science 267, 222-225 (1995); ACS Omega 7, 39362-39369 (2022); Nat. Commun. 12, 2391 (2021)]. As shown in Supplementary Fig. 27, the components of the velocity vector (V_y and V_z) exhibit opposite gas flow directions in the confined space compared to V_x , indicating that the backflow gas is generated around the substrate. The turbulent flow is supposed to enhance the MoS₂ nucleation in the carrier gas because that backflows increase the collision rate of molecules to the substrate and thus produces enough energy for the *in-situ* twist nucleation. As a result, there is an alteration of the synthesis conditions. The experimental results (DF-OM) in Supplementary Fig. 4 and Supplementary Fig. 25 show that the MoS₂ layer grows with different rates while sharing the same nucleation site. We believe that the introduced backflow gas through the inner tube drives the entire inner tube out of equilibrium, resulting in the reduced chance to form more stable 0°- and 60°-TB-MoS₂. Therefore, in the high-temperature CVD process, the medium and large nucleus sizes are beneficial for the growth of bilayer MoS₂. Based on the twist nucleation process, we realize that the reconfiguring nucleation under the assistance of NaCl and confined space is the key for the successful synthesis of TB-MoS₂ by CVD. In addition, it is noteworthy that the proposed CVD technique has been utilized to synthesize other TB-TMDCs. Supplementary Fig. 28 presents the successful growth of TB-WS₂ using comparable

procedures.

Changes to the revised Supplementary Information are shown below (In Page 11-13).

Supplementary Note 6. Kinetics analysis of confined space.

To further clarify the effect of the confined space, we have carried out the computational fluid dynamics simulations of the confined space in CVD. The velocity distribution, temperature distribution, and components of the velocity vector for this CVD setup are shown in Supplementary Fig. 27. Here, we believe that the confined space will change the gas flow rate, the temperature, and the gas flow direction. The corresponding detail discussion are shown below.

(1) Gas flow rate.

It is observed from the velocity distribution that the gas flow condition affects the sample state in the open tube system (Supplementary Fig. 14 and Supplementary Fig. 27). It can be identified that the gas velocity in the inner tube (without sealed-end) is one order of magnitude less than that of the outer tube, indicating that there is a change in the gas fluid velocity on the SiO₂/Si substrate. With the decrease in gas flow rate, the fluid velocity of carrier gas decreases, making the nuclei surplus. As a result, the nuclei will grow larger, which is consistent with the experimental results (Supplementary Fig. 21). As we know, small change in the flow regime in CVD significantly influences the reaction products. Although the gas velocity in the inner tube has been greatly reduced, there is a nonnegligible increase. Therefore, there is a rapid change in the yield and density of TB-MoS₂, indicating that the gas flow rate is important in the synthesis process, which mainly affects the nucleation and growth of TB-MoS₂.

(2) Temperature.

Temperature is another key parameter for CVD process, wherein the different reaction temperatures may result in different morphology [*ACS Nano* **12**, 635-643 (2018); *Nat Commun* **10**, 598 (2019); *Nat Commun* **12**, 809 (2021); *ACS Nano* **13**, 8265-8274 (2019)]. Based on previous reports, the preparation of MoS₂ usually requires a relatively high reaction temperature from 600 °C to 1000 °C [*Nature* **556**, 355–359 (2018); *Nat Commun* **10**, 598 (2019); *Nature* **605**, 69-75 (2022)]. However, based on our experimental results, we find that the TB-MoS₂ can only be well synthesized under a narrow reaction temperature range from

770 °C to 790 °C. As shown in Supplementary Fig. 17, TB-MoS₂ can be obtained under a reaction temperature of 780 °C. Higher or lower temperatures result in the dramatic decrease in the yield of TB-MoS₂. For example, large-area monolayer MoS₂ is synthesized under a reaction temperature of 760 °C.

Temperature is the key to affect the thermodynamics and dynamics of the reaction system. Here, we have carried out the simulation on the temperature in the tube, as shown in Supplementary Fig. 27c-d. As noticed, due to the effect of the gas flow rate, the temperature shows an uneven distribution. However, with the introduction of the confined space, there is a uniform temperature distribution near the SiO₂/Si substrate. The inner tube serves as a “heat insulation layer” to prevent the heat diffusion. Thus, we believe that the confined space plays the key role to stabilize the substrate temperature.

(3) Turbulent flow.

In most cases, laminar gas flow is desirable, but some local areas of turbulent flow may exist. Generally, conventional CVD that uses tubular horizontal reactors possesses laminar flow under any conditions. A high pressure or flow rate is required to produce turbulent flow in the reactor, which exceeds the flow rate used in CVD. To induce turbulent flow, some form of physical disturbance of the gas flow is needed in the CVD setup. In this work, we introduce the confined space to CVD to induce the turbulent flow in the inner tube. This design creates a circumfluent flow, wherein the gas flows around the chamber and backflows toward the substrate. This significantly reduces the gas flow velocity on the substrate to create an unsteady gas flow.

It is reported that the turbulent flow indeed causes the different growth conditions in CVD process. The turbulent flow has been adopted for the synthesis of 2D materials, such as graphene and TMDCs, which effectively changes the morphology of the products [*Science* **267**, 222-225 (1995); *ACS Omega* **7**, 39362-39369 (2022); *Nat. Commun.* **12**, 2391 (2021)]. The components of the velocity vector V_x , V_y , and V_z in this confined space are shown in Supplementary Fig. 27. As noticed, V_x shows a positive value (Supplementary Fig. 27e), while V_y and V_z exhibit opposite gas flow directions (Supplementary Fig. 27f-g). These results indicate that the backflow gas is generated around the substrate. In other words, no extreme conditions are required to generate the turbulent flow in the inner tube. The turbulent flow is supposed to enhance the MoS₂ nucleation in the carrier gas because backflows

increase the collision rate of molecules to the substrate by producing enough energy for the twist nucleation, resulting in an alteration of the synthesis conditions. We believe that the introduced backflow gas through the inner tube drives the entire inner tube out of equilibrium to reduce the chance to form more stable 0°- and 60°-TB-MoS₂, which is expected to be the most important reason for the synthesis of TB-MoS₂.

Comment 5:

The authors claimed that the molar ratio of NaCl and MoO₃ is another key parameter to influence the yield, but show little analysis and corresponding experimental results on the reason.

Response:

Thank you for raising this comment. We believe that the molar ratio of NaCl to MoO₃ is one of the key parameters to influence the yield and density. The related experiment results and analysis are shown below.

(1) Experimental results and analysis on molar ratio of NaCl to MoO₃.

Here, additional experiments on the molar ratio of NaCl to MoO₃ is carried out and the typical OM is shown in Fig. R7. Based on the results, we calculate the density and yield of all the TB-MoS₂ samples, as shown in Fig. R22. With the increase in molar ratio of NaCl to MoO₃ from 0 to 20, the density increased from 0 to 28.9 pieces/mm², while the yield increases from 0 to 17.2%. However, TB-MoS₂ cannot be synthesized with a molar ratio of over 30, indicating that the molar ratio of NaCl to MoO₃ of 3 to 20 would be a suitable range.

Here, the molar ratio of NaCl to MoO₃ dependent experiments is carried out and the typical OM is shown in Fig. R7. Based on the results of the experiment, we calculated the density and yield of all the TB-MoS₂ samples, as shown in Fig. R22. While, with the increase of molar ratio of NaCl to MoO₃ from 0 to 20 the yield and density increased from 0 to 28.9 pieces/mm², while the yield increases from 0 to 17.2%, respectively. While the TB-MoS₂ cannot be synthesized when the molar ratio of NaCl to MoO₃ reach to 30, indicating that the molar ratio of NaCl to MoO₃ of 3 to 20 would be a suitable growing parameter range.

Fig. R7 Typical BF-OM of synthesized products under different molar ratio of NaCl to MoO₃. a-e The typical OM images of the synthesized MoS₂ under different molar ratio of NaCl to MoO₃ of 30, 20, 10, 3, and 0. Scale bars: 100 μm for the OM under 20X objective, 10 μm for the small area under 50X objective.

Fig. R19 Density and yield of TB-MoS₂ under different molar ratio of NaCl to MoO₃.

Besides, all the TB-MoS₂ samples have a uniform bottom-layer size in bilayer MoS₂ (as seen in Fig. R7b-c). In order to further clarify the flake size, we have counted the flake size of the bottom-layer and top-layer in the bilayer TB-MoS₂ sample. The frequency distribution histograms of bottom-layer size, top-layer size, and bottom-top-layer size ratio are shown in Fig. R13. The gamma distribution fitting curve can well fit all the frequency distribution plots,

which is similar to the size of our CVD-synthesized WTe₂ NRs [*J. Am. Chem. Soc.* **143**, 18103-18113 (2021)]. The average flake size and standard deviation of the bottom-layer in TB-MoS₂ are calculated to be 28.56 μm and 7.43 μm , respectively, which indicates that the size distributes mainly in ~ 30 μm . This concentrated distribution indicated that our statistics on density are reliable. The flake size of the top-layer in TB-MoS₂ can be calculated to be 9.00 μm , and the bottom-top-layer size ratio in TB-MoS₂ shows a relatively concentrated distribution around 0.33, indicating the uniformity of the samples.

Fig. R16 Proportion of monolayer MoS₂, bilayer MoS₂, multilayer MoS₂ and non-MoS₂ in as-grown samples under different molar ratio of NaCl to MoO₃, respectively.

Fig. R13 Frequency distribution histograms of flake size of TB-MoS₂ and corresponding gamma distribution fitting curve (with the mean value (E) and standard deviation (σ)). a Flake size of bottom-layer in TB-MoS₂. b Flake size of top-layer in TB-MoS₂. c The bottom-top-layer size ratio in TB-MoS₂.

The selectivity of bilayer MoS₂ also was discussed based on the experiment results of different molar ratio of NaCl to MoO₃, as shown in Fig. R16. For the molar ratio of NaCl to

MoO₃, we cannot synthesize MoS₂ without NaCl. Interestingly, the high proportion of monolayer MoS₂ (94.0%) can be obtained under a low NaCl to MoO₃ molar ratio of 3. With the increase in the molar ratio of NaCl to MoO₃, the bilayer MoS₂ increases first and then decreases, exhibiting a maximum proportion (60.9%) under the molar ratio of NaCl to MoO₃ of 20.

(2) The thermodynamically and kinetics on NaCl in CVD process.

In the previous works, there are detailed discussion for the influence of the salt on the CVD growth of TMDCs via thermodynamics and kinetics [*Nature* **556**, 355-359 (2018); *J. Am. Chem. Soc.* **143**, 18103-18113 (2021)]. Here, we briefly discussed the vapor pressure of reaction precursors. As shown in Fig. R15, the vapor pressure of NaCl and MoO₃ can be found around ~10 Pa and ~10⁻⁹ Pa under 780 °C (1053 K), respectively. Considering that the solubility of metal oxide is at the order of ppm, the metal oxide and salt will dramatically increase the vapor pressure of metal precursors by a few orders.

[REDACTED]

Fig. R15 Temperature dependence of vapor pressure of transitional metal oxides and salt.
(Taken from Supplementary Fig. 35 in [*Nature* **556**, 355-359 (2018)])

In terms of affecting thermodynamics, NaCl can be reacted with MoO₃ to form MoO_xCl_y. Notably, the degrees of freedom changed by reacting with MoO_xCl_y, resulting in the change of entropy. Compared with the CVD system without NaCl, the participation of MoO_xCl_y may affect the Gibbs free energy ($\Delta G = \Delta H - T\Delta S$) by neglecting the change of pressure and volume.

While for the kinetics process, NaCl will affect the nucleation process and dominate the

geometries of MoS₂ layers. The nucleation rate with and without NaCl can be roughly written as:

$$\frac{\dot{N}(\text{MoO}_3 + \text{NaCl})}{\dot{N}(\text{MoO}_3)} \sim \frac{P_i(\text{NaCl})}{P_i(\text{without NaCl})} \quad (4)$$

where \dot{N} is the nucleation rate, P_i is partial pressure of precursors. From this equation, we can clearly find that the partial pressure P_i is the dominate factor for the nucleation rate. As mentioned before, the vapor pressure of NaCl is much higher than that of MoO₃ precursors. Besides, in case of reaction between MoO₃ and NaCl, MoO_xCl_y will be formed, resulting in a high P_i due to the high volatility nature of MoO_xCl_y. Therefore, the NaCl will result in a higher nucleation rate.

Without the addition of NaCl, the MoO₃ precursor is evaporated under the temperature. As a result, quadrilateral shapes MoO₃ and MoO₂ preferentially nucleate and grow on the SiO₂/Si substrate (Fig. R7). Based on the previous discussion, we can notice that the solubility of metal oxide is at the order of ppm. Therefore, the reactant concentrations of MoO_xCl_y were greatly affected by the molar ratio of NaCl to MoO₃ at the definite reaction temperature. With the increase of NaCl to MoO₃ molar ratio, the reactant concentrations of MoO_xCl_y increased. Therefore, increasing the molar ratio of NaCl to MoO₃ makes the nuclei surplus grow larger, resulting in the arising number of bilayer MoS₂ and the monolayer concomitant on SiO₂/Si, as shown in Fig. R7b-c. Besides, the thickness of MoS₂ under the different molar ratio of NaCl to MoO₃ is shown in Fig. R16, indicating that the high proportion of bilayer MoS₂ (60.9%) can be obtained under high molar ratio of NaCl to MoO₃ (20). With the introduction of the confined space, the growth kinetics was beneficial for CVD growth of the TB-MoS₂ with the thermodynamically unfavourable twisted stacking structures. Therefore, The TB-MoS₂ can be synthesized under the suitable molar ratio of NaCl to MoO₃.

Fig. R20 Flow chart of the salt-assisted growth process for the reconfiguring nucleati on and growth of TB-MoS₂ via the space-confined CVD method. Scale bars: 100 μm.

Therefore, to address the reviewer’s concern, we have added the results and discussion on the molar ratio of NaCl to MoO₃ in the revised manuscript and Supplementary Information.

Changes to the revised manuscript are shown below (In Page 17-19).

Apart from the gas flow rate, the molar ratio of NaCl to MoO₃ is another key parameter affecting the density and yield of TB-MoS₂. To investigate the growth process of TB-MoS₂, controlled experiments on the molar ratio of NaCl to MoO₃ have been carried out. As shown in Fig. 5c and Supplementary Fig. 22, with the increase in the molar ratio from 0 to 20, the density increases from 0 to 28.9 pieces/mm², while the yield increases from 0 to 17.2%, respectively. The TB-MoS₂ cannot be synthesized when the molar ratio of NaCl to MoO₃ reaches 30, indicating that the molar ratio 3 to 20 would be a suitable growing parameter range.

In contrast, the added NaCl can be reacted with the MoO₃, forming the oxychloride compounds MoO_xCl_y and enhanced the concentration and the precursor vapor pressure [*Nature* **556**, 355-359 (2018); *J. Am. Chem. Soc.* **143**, 18103-18113 (2021)]. The resultant compounds further react with the sulfur, resulting in the growth of triangular shapes MoS₂

crystal on SiO₂/Si substrate. The reactant concentrations of MoO_xCl_y are greatly affected by the molar ratio of NaCl to MoO₃ at the definite reaction temperature. With a low salt ratio, nuclei with small size are obtained, and consequently monolayer MoS₂ is grown on SiO₂/Si substrate (Fig. 5d and Supplementary Fig. 25). With the increase in the molar ratio of NaCl to MoO₃, the reactant concentrations of MoO_xCl_y increases, making the nuclei surplus grow larger. As a result, there is a rising number of bilayer MoS₂ and the monolayer concomitant on SiO₂/Si, and some TB-MoS₂ can be observed (Fig. 5d and Supplementary Fig. 25). When the molar ratio of NaCl to MoO₃ further increases, the nuclei size becomes even larger, leading to the disappearance of monolayer MoS₂. In contrast, the bilayer and even multilayer MoS₂ are obtained. The proportion of MoS₂ with different layer numbers under different molar ratio of NaCl to MoO₃ is shown in Supplementary Fig. 26. The highest proportion of bilayer MoS₂ (60.9%) can be obtained under a molar ratio of 20, and the excessive NaCl can result in the growth of multilayer MoS₂. Therefore, the TB-MoS₂ can be synthesized under the suitable molar ratio of NaCl to MoO₃. Based on the calculation results, the 0°-TB-MoS₂ and 60°-TB-MoS₂ have the lowest energy, which has been widely synthesized by the CVD method [*Nat. Commun.* **5**, 4966 (2014); *Nano Lett.* **14**, 3869-3875 (2014)]. Therefore, the TB-MoS₂ is hardly grown via a conventional thermodynamic growing method.

Fig. 5 | Key parameters for CVD grown TB-MoS₂. **a** Statistical distribution of twist angles based on OM of as-grown TB-MoS₂. **b-c** Density and yield of TB-MoS₂ under different gas flow rate and molar ratio of NaCl to MoO₃, respectively. **d** Flow chart of the salt-assisted growth process for the reconfiguring nucleation and growth of TB-MoS₂ via the space-confined CVD method. Scale bars: 100 μm in **d**.

Changes to the revised Supplementary Information are shown below (In Page 36, Page 39-40).

Supplementary Fig. 22 Typical BF-OM of synthesized products under different molar ratio of NaCl to MoO₃. a-e The typical OM images of the synthesized MoS₂ under different molar ratio of NaCl to MoO₃ of 30, 20, 10, 3, and 0. Scale bars: 100 μm for the OM under 20X objective, 10 μm for the small area under 50X objective.

Supplementary Fig. 25 The typical BF-OM of products synthesized under different salt ratio. **a-d** The typical OM images of the synthesized materials without NaCl, with low salt ratio, medium salt ratio, and high salt ratio. Scale bars: 100 μm for the OM under 20X objective, 10 μm for the small area under 50X objective.

Supplementary Fig. 26 Proportion of monolayer MoS₂, bilayer MoS₂, multi-layer MoS₂ and non-MoS₂ in the as-grown samples. **a-b** Under different gas flow rate and molar ratio of NaCl to MoO₃, respectively.

Response to Reviewer 3

The author reports a NaCl-assisted space-confined chemical vapor deposition (CVD) method to direct synthesis bilayer MoS₂ with twist angles ranging from 0 to 120°. The study of twisted transition metal disulfide lasted for more than a decade. Previous studies include ACS Nano 2018, 12, 8, 8770-8780, Nat Commun 5, 4966 (2014), Nano Lett. 2014, 14, 10, 5500-5508, Nat Commun 11, 2153 (2020), Adv. Funct. Mater. 2022, 32, 2111529, ACS Appl. Mater. Interfaces 2023, 15, 3, 4724–4732, CrystEngComm, 2021,23, 2889-2896, have deeply revealed the optical, electrical, and mechanical properties of both CVD and transferred twisted MoS₂. From this point of view, the characterizations of twisted bilayer MoS₂ in this manuscript are not new. The key point of this manuscript is that the author offers a relatively easy method to grow twisted bilayer MoS₂ with abundant twist angles, which is meaningful for the study of twistrionics. However, the research and discussion on the growth parameters are incomplete, and the current manuscript contains defects and mistakes. Hence, I do not support considering this manuscript by Nature Communication at the current stage.

Response:

We sincerely appreciate the reviewer for carefully reading our work and providing us with invaluable suggestions to improve it. In the revised manuscript, we have made our best to address your concerns. Our point-by-point responses are shown below.

Comment 1:

The author needs to check the language of this manuscript carefully. It is very hard to read and contains too many grammar problems, such as lines 91: “filed”; 152-155 “may be”; 169 “Six hexagonal points”; 170 “demonstrating which indicate”; 179: “noticed”; and 180: “under”.

Response:

Thank you for your constructive suggestions. We have corrected the grammatical mistakes in the revised manuscript. Furthermore, we have sought the assistance of a native English speaker to refine the sentences. The revised portions are indicated in **blue** in the revised manuscript.

Comment 2:

At line 91, “twist angle is defined by the angular difference along the corresponding edges of the top and bottom layers (indicated by dashed green and red lines, respectively). The twist angles are determined from the bright field (BF) optical microscopy (OM) images with an error of about 0.3° (Supplementary Fig. 1).” The author used edges of the top and bottom MoS₂ to determine the twist angle and error. Fig. 1g in the manuscript and Fig. S2, 4, 5, 6, and 7 in the Supplementary information show that most edges of either top or bottom MoS₂ triangle domains are not straight. Thus, just comparing edges is insufficient to determine the twist angle and the error.

Response:

Thank you for raising these insightful comments. In this work, we have used the OM, SHG, AFM, TEM, SAED, and STEM to determine the twist angle. We agree with the reviewer that measuring the edges of TB-MoS₂ by OM to determine the twist angle is not a precise method compared to the SAED pattern under TEM. Owing to direct growth of the TB-MoS₂ on SiO₂/Si substrate, it is an efficient way to estimate the twist angle using the domain edge, which is also compatible with future device fabrication. Here, the high-quality triangle morphology TB-MoS₂ samples with the most stable zigzag edge in our synthesis parameters and the zigzag edge were widely used to roughly determine the orientation of TMDCs [*Nat. Nanotechnol.* (2023). DOI: 10.1038/s41565-023-01456-6; *Nat. Nanotechnol.* **10**, 407–411 (2015); *Nature* **514**, 470–474 (2014)]. The OM method was also widely adopted for the determination of twist angles [*Adv. Mater.* **35**, 2210909 (2023); *Nano Lett.* **14**, 5500-5508 (2014); *Nat. Commun.* **5**, 4966 (2014); *Nano Lett.* **14**, 3869-3875 (2014); *Nano Lett.* **16**, 1435-1444 (2016)].

SHG method is another optically method to be widely used to determine the twist angles of 2D materials such as TB-graphene and TB-TMDCs from the SHG polarization dependence with an uncertainty of 0.3° . [*Nature* **567**, 76-80 (2019); *Nature* **567**, 81-86 (2019); *Nat. Mater.* **19**, 861-866 (2020)]. In our initial manuscript, the OM measurement results matched well with the SHG results, as shown in Fig. R21 and Fig. R22. As noticed, the SHG azimuthal angle was calculated to be 0° , 4.5° , 10.5° , 16.3° , 37.3° , 40.3° , 42.5° , 47.7° corresponding to the 0° , 9.2° , 21.0° , 32.7° , 47.9° , 81.0° , 84.7° , and 95.3° -TB-MoS₂ samples from Fig. R21. The two-fold SHG azimuthal is highly consistent with the twist angle measurement from OM,

which indicates the accuracy of the twist angle measurement from OM.

Fig. R21 The BF-OM, DF-OM and the corresponding polar plots of parallel components of SHG intensity of TB-MoS₂. **a-h** The SHG BF-OM, DF-OM, and SHG results of 0°, 9.2°, 21.0°, 32.7°, 47.9°, 81.0°, 84.7°, and 95.3°-TB-MoS₂, respectively. The corresponding SHG azimuthal angles can be measured to be 0°, 4.5°, 10.5°, 16.3°, 37.3°, 40.3°, 42.5°, 47.7°, respectively. Scale bars: 10 μ m.

Fig. R22 The relationship between the SHG azimuthal angle and twist angle measured from OM in TB-MoS₂. **a** The relationship between the azimuthal angle measured from the SHG and the twist angles measured from the OM. **b** The twist angle errors between the twist angle and the twofold SHG azimuthal angle. The SHG azimuthal angle was calculated to be 0°, 4.5°, 10.5°, 16.3°, 37.3°, 40.3°, 42.5°, 47.7° corresponding to the 0°, 9.2°, 21.0°, 32.7°, 47.9°, 81.0°, 84.7°, and 95.3°-TB-MoS₂ samples from Supplementary Fig. 2. The twofold SHG azimuthal is highly consistent with the twist angle measured from OM, which indicated that the accuracy of twist angle measurement from OM.

Based on the reviewer's suggestion, we systematically compared the twist angle derived from the edge direction of the top and bottom layer on OM and corresponding SHG results to improve the accuracy of the OM-based method. As the reviewer mentioned, some edges of top layer are not straight. This indeed results in uncertainty in the measurement of the twist angles. To evaluate the error of the twist angle obtained based on the domain edge, we established rules for measuring twist angle based on the edge of OM to ensure the closer corresponding result to the derived results of SHG. Based on the OM of TB-MoS₂, we summarized three types of TB-MoS₂ morphologies that we usually obtained as well as the rules for measuring the twist angles, as shown in Fig. R23.

Fig. R23 Rules for measuring the twist angles of TB-MoS₂. **a** Triangle with straight edge (Type I). **b** Triangle surrounded by satellite-triangle (Type II). **c** Triangle with the **not** straight edge (Type III).

Type I: Triangle with straight edge. In this case, the MoS₂ shows the triangle morphology (Fig. R24a). As shown in Fig. R24b, we take two regular triangles for each layer, and the red and green triangles matched well with the triangle edges of the DF-OM. Based on OM, three twist angles were measured and listed in Fig. R24c, and the twist angle can be calculated to be 21.0°. Fig. R24d shows the *in-situ* SHG result of the samples. By measuring the crystal orientations between the SHG pattern of each layer, we can obtain the twist angle of 21.1°.

Fig. R24 Type I. **a** Typical BF-OM and DF-OM of MoS₂ with the straight edge. **b** Sematic diagrams for measuring the twist angles. **c** Measured results of the twist angles. **d** SHG results of TB-MoS₂ taken from the regions marked in **a**. Scale bars: 10 μm .

Type II: Triangle surrounded by satellite-triangle. In this case, the MoS₂ shows the triangle surrounded by satellite-triangle morphology (Fig. R25a). The pattern can be simplified, and the largest triangle was selected for further measurement (Fig. R23b and Fig. R25b). Therefore, two regular triangles for each layer can be established. The red and green triangles matched well with the triangle edges of the DF-OM. Based on OM, three twist angles were

measured and listed in Fig. R25c with the calculated value of 9.2° . Fig. R25d shows the *in-situ* SHG result of the samples. By measuring the crystal orientations between the SHG pattern of each layer, we can obtain the twist angle of 9.0° , similar to the results from OM.

Fig. R25 Type II. **a** Typical BF-OM and DF-OM of MoS₂ with the triangle surrounded by satellite-triangles. **b** Sematic diagrams for measuring the twist angles. **c** Measured results of the twist angles. **d** SHG results of TB-MoS₂ taken from the regions marked in **a**. Scale bars: 10 μm .

Type III: Triangle with the irregular edges. There are some MoS₂ samples with rough edge (Fig. R23c). We need to find out the alternative line to a straight edge. Firstly, we need to outline the stroke of the uneven edge. Then, the external regular triangle and inscribed regular triangle can be obtained. Here, the edges of the two regular triangles was alternatives for the twist angle measurement. Based on the simplified model, we can efficiently calculate the twist angle. Take one TB-MoS₂ as an example, the typical OM is shown in Fig. R26a. As noticed in Fig. R26b, the outline stroke of the uneven edge is well matched with the triangle edges of the DF-OM. Then, two regular triangles can be obtained. Based on the OM measurement, six twist angles can be measured as shown in Fig. R26c. The twist angle is 81.0° based on OM, corresponding to 80.7° from the SHG (Fig. R26d).

Fig. R26 Type III. a Typical BF-OM and DF-OM of MoS₂ with the **not** straight edge. **b** Sematic diagrams for measuring the twist angles. **c** Measured results of the twist angles. **d** SHG results of TB-MoS₂ taken from the regions marked in **a**. Scale bars: 10 μm.

As we discussed above, the SAED would be the best way to determine the twist angle. While the above rule is applicable for the measurement of TB-MoS₂ in this work. We can notice that the twist angle measured by OM method is close to that obtained from SHG patterns. To better understand the rules for measuring the twist angles of TB-MoS₂, we have added the corresponding details in the revised manuscript and Supplementary Information.

Changes to the revised Supplementary Information are shown below (In Page 4).

Supplementary Note 1. Characterization of the twist angle based on OM.

The TB-MoS₂ shows the triangle morphology in this work, and the monolayer and bilayer share the same nucleation site. Therefore, an edge of the monolayer (dashed green line) rotates counterclockwise to meet one side of the bilayer (dashed red line). The rotated angle is defined as the twist angle θ ($0^\circ < \theta < 120^\circ$) (Supplementary Fig. 1a). Measuring the edges of MoS₂ to estimate the twist angle is an efficient way to calculate the twist angles of all bilayer-MoS₂. Here, we summarized three types of TB-MoS₂ morphologies and the corresponding rules to measure the twist angles, as shown in Supplementary Fig. 1a-c.

The twist angles of the TB-MoS₂ can be directly measured based on the regular triangles from the OM when the triangle with regular edge (Type I). Three twist angles can be measured and the average values are calculated (Supplementary Fig. 1a). The triangle surrounded by satellite-triangle (Type II) samples should be measured by the largest triangle (Supplementary Fig. 1b). For the triangle with irregular edges, the triangle can be replaced by the external inscribed regular triangle for the twist angle measurement of TB-MoS₂ (Supplementary Fig. 1c). The average of six twist angles (with three θ_{out} and three θ_{in}) is considered to be the twist angle of TB-MoS₂.

Here, a TB-MoS₂ sample was used to calculate the twisted angle by OM. As shown in Supplementary Fig. 1b, the twist angle can be measured to be 22.0°, 21.5°, and 22.4°. The average value of 22.0° with a standard deviation of about 0.35° was used as the twist angle (Supplementary Fig. 1d). Actually, the error is inevitable when measuring the orientations of each edge (Supplementary Fig. 1f) due to the limited resolution (the minimum resolution σ for an optical microscope is around 200 nm, which responds to the size of fuzzy region) of

OM images. The errors (φ_{OM}) of each edge can be determined as:

$$\varphi_{OM} = \arctan \frac{\sigma}{L} \quad (1)$$

where L is the length of the triangle, σ is the minimum resolution of OM. Here, we can notice that the φ_{OM} is dependent on the length of the triangle, thus we use the average length of TB-MoS₂ to calculate the error. The average L is calculated to be 28.56 μm (Supplementary Fig. 20) and the error (φ_{OM}) is 0.4°. Due to the twist angle (θ) being determined by measuring two edges of the monolayer and bilayer, the OM based error of the twist angle should be twice the value of the error (φ_{OM}):

$$\text{Error}_{\theta-OM} = 2 \times \varphi_{OM} \approx 0.8^\circ \quad (2)$$

As shown in Supplementary Fig. 1, the standard deviation is about 0.35°, smaller than the $\text{Error}_{\theta-OM}$ of 0.8°.

Supplementary Fig. 1 Rules for measuring the twist angles and corresponding error analysis of TB-MoS₂ based on OM method. **a** Schematic diagram for measuring the TB-MoS₂ with straight triangle edge (Type I), **b** triangle surrounded by satellite-triangle (Type II), and **c** triangle with irregular edge (Type III). **d** The typical OM image of TB-MoS₂. **e** The measured twist angles based on the OM from the TB-MoS₂ in **d** (scale bar: 20 μm). **f** The origin of error during the measurement.

Comment 3:

Line 129-130, “.....the TB-MoS₂ shows a higher vibration intensity, while the frequency of E_{2g} mode decreases ($\Delta\omega = 2.4 \text{ cm}^{-1}$) and the A_{1g} mode increases ($\Delta\omega = 2.1 \text{ cm}^{-1}$)”. Please clarify the “vibration intensity” and its relation with the Raman shift.

Response:

We are grateful for the reviewer’s valuable comment. We have provided a more detailed explanation regarding the vibration intensity and Raman shift between monolayer and bilayer of TB-MoS₂ as well as the relationships as follows.

Firstly, during Raman spectroscopy measurements, the presence of multiple media (air, MoS₂, SiO₂, and Si) leads to multiple reflections and refractions of the incident light at the boundaries between MoS₂ and SiO₂ layers. As a result, the optical path of both the excitation and scattering light becomes highly intricate. For such multilayer structures, a multi-reflection model is usually constructed to quantify the Raman intensity of ultrathin flakes of two-dimensional layered materials [*Nano Lett.* **15**, 1356-1361 (2015); *Phys. Rev. B* **80**, 125422 (2009); *Nanoscale* **7**, 8135-8141 (2015); *ACS Nano* **6**, 7381-7388 (2012)]. In this model, the laser excitation and Raman scattering processes are treated separately, so the output Raman intensity (I) from the top MoS₂ layer (total thickness d) can be expressed as:

$$I = \int_0^d |E_{ab}(x)E_{em}(x)|^2 dx \quad (5)$$

where the $E_{ab}(x)$ and $E_{em}(x)$ are the electric field amplitudes within the flakes and light emitting out of the flakes, respectively. The relative emission efficiency of scattered light and the relative electric-field strength will be fixed between Si and a specific MoS₂ layer (less than 6-layer, 3.9 nm) [*Phys. Rev. B* **97**, 165409 (2018)]. Bilayer MoS₂ only has a thickness of 1.7 nm, which is much smaller than the laser wavelength. Therefore, when the thickness of MoS₂ increases from monolayer to bilayer, the Raman intensity will be enhanced.

Furthermore, with the increase in layer number for MoS₂, the long-range Coulombic interaction between the Mo atoms decreases and the enhancement of interlayer interactions leads to restoring forces on the S atoms increasing. So, compared to monolayer MoS₂, the E_{2g}^1 mode decreases and the A_{1g} mode increases for TB-MoS₂ [*J. Raman Spectrosc.* **44**, 92-

96 (2013)].

Therefore, to be more precise, we have revised the corresponding discussions in the revised manuscript.

Changes to the revised manuscript are shown below (In Page 7).

Compared to monolayer MoS₂, TB-MoS₂ shows a higher vibration intensity attributed to the strongly increased Raman intensity resulting from the increase in layer number [*Phys. Rev. B* **97**, 165409 (2018); *Nano Lett.* **15**, 1356-1361 (2015); *Phys. Rev. B* **80**, 125422 (2009); *Nanoscale* **7**, 8135-8141 (2015); *ACS Nano* **6**, 7381-7388 (2012)]. Besides, the frequency of the E_{2g}^1 mode decreases ($\Delta\omega = 2.4 \text{ cm}^{-1}$), while the frequency of the A_{1g} mode increases ($\Delta\omega = 2.1 \text{ cm}^{-1}$). This is caused by the long-range Coulombic interaction between the Mo atoms. Additionally, the enhancement of interlayer interactions leads to restoring forces on the increased S atoms [*J. Raman Spectrosc.* **44**, 92-96 (2013)].

Comment 4:

Line 135, the homogenous Raman or PL mappings can not support low defects concentration. The distribution of defects can be uniform.

Response:

We appreciate your comment. We agree with the reviewer's viewpoint. Raman can only represent the uniformity of the materials. In fact, homogeneous defects will also result in homogeneous Raman mappings. Consequently, we have made necessary adjustments to the relevant section in the revised manuscript.

Changes to the revised manuscript are shown below (In Page 7).

The uniform intensity observed throughout the mapping region indicates the crystalline uniformity of monolayer and bilayer in 101.3°-TB-MoS₂ sample.

Comment 5:

Line 140, “.....which is consistent with the Raman spectrum in the literature.” No citation support.

Response:

We apologize for the missing citations. According to the reviewer’s suggestions, we cited corresponding literature accordingly in the revised manuscript.

Changes to the revised manuscript are shown below (In Page 7-8).

Compared to the peak locations for monolayer MoS₂, a red shift of E_{2g}^1 mode (2.3cm⁻¹) and a blue shift of A_{1g} (2.0 cm⁻¹) are observed for bilayer MoS₂, which is consistent with the Raman spectrum in the literature [*ACS Nano* **4**, 2695-2700 (2010); *Nat Commun* **9**, 4778 (2018); *Nature* **605**, 69-75 (2022)]. Moreover, the bilayer MoS₂ shows a more homogeneous Raman shift across the whole mapping region, indicating a high consistency of vibration mode in 101°-TB-MoS₂.

Comment 6:

Fig. 2a and e, please mark the “monolayer” and “bilayer” on the spectra.

Response:

Thanks for the comment. We have added the labels of “monolayer” and “bilayer” on Fig. 2a and 2e.

Changes to the revised manuscript are shown below (In Page 33).

Previous version:

Revised version:

Comment 7:

Fig. 3, in the STEM and SAED measurements, it is unclear what method was used to determine twist angles. How much error between the twist angle measured by SAED (STEM) and the OM method?

Response:

Thank you for raising this question. The point-by-point response has been listed as below.

(1) Rule for measuring the twist angle of TB-MoS₂ based on STEM (SAED) method.

The SAED is an accurate method for measuring the twist angle of TB-MoS₂, which is widely used to determine the twist angle of 2D materials such as TB-graphene and TB-TMDCs [*Nano Lett.* **22**, 203-210 (2022); *Nano Lett.* **21**, 3262-3270 (2021); *Nat. Commun.* **12**, 2391 (2021); *Nat. Mater.* **21**, 1263-1268 (2022)]. Firstly, based on the STEM-HAADF image of TB-MoS₂, we can preliminarily identify the twist angle θ_{STEM} range belong to 0°~30°, 30°~60°, 60°~90°, or 90°~120°. Then, we measured the direction of two proximal diffraction points of each layer. The twist angle θ_{SAED} can be measured in the range of 0° to 30°. Based

on the following equations, the twist angle θ_{STEM} can finally be determined.

$$\theta_{\text{SAED}} = \theta_{\text{STEM}} \quad (0 < \theta_{\text{STEM}} \leq 30^\circ) \quad (6)$$

$$\theta_{\text{SAED}} = 60^\circ - \theta_{\text{STEM}} \quad (30^\circ < \theta_{\text{STEM}} \leq 60^\circ) \quad (7)$$

$$\theta_{\text{SAED}} = \theta_{\text{STEM}} - 60^\circ \quad (60^\circ < \theta_{\text{STEM}} \leq 90^\circ) \quad (8)$$

$$\theta_{\text{SAED}} = 120^\circ - \theta_{\text{STEM}} \quad (90^\circ < \theta_{\text{STEM}} \leq 120^\circ) \quad (9)$$

Taking a 21.9° -TB-MoS₂ sample as an example, the typical STEM image and corresponding SAED patterns are shown in Fig. R27. Firstly, based on the STEM-HAADF image, θ_{STEM} can be identified in the range of $0^\circ \sim 30^\circ$. Then, the twist angle θ_{SAED} between the two proximal diffraction points is measured to be 21.9° . Thus, the twist angle θ_{STEM} of this sample was determined to be 21.9° .

Fig. R27 Rule for measuring the TB-MoS₂ based on STEM and SAED. **a** STEM-HAADF image of the 21.9° -TB-MoS₂. **b-d** SAED patterns of monolayer and bilayer in 21.9° -TB-MoS₂ in **a**. Scale bar: $5 \mu\text{m}$ in **a**, 5nm^{-1} in **b-e**.

(2) Error analysis of twist angle measurement based on SAED.

The diffraction points become blurred after amplification, resulting in inevitable errors in the measurement. Taking a SAED pattern as an example (Fig. R28a), the weight lines lotted connecting the opposite diffraction points intersect each other at a specific twist angle. The error (φ_{SAED}) based on SAED pattern can be determined by the full width at half maximum (FWHM) of the blurred diffraction points and the distance between each opposite bright point in a single SAED pattern (Fig. R28b).

$$\varphi_{\text{SAED}i} = \arctan \frac{\text{FWHM}_i}{D_i} \quad (10)$$

where i is the diffraction order ($i=1,2,3,4\dots$), D_i is the distance from the two opposite bright point under the diffraction order of i . To calculate the error based on SAED method, we measured FWHMs of the four groups of the position dependent diffraction intensity plot (1st, 2nd, 3rd, and 5th, Fig. R28c-f) with the average FWHMs of 0.0049 nm^{-1} . While the distance from the two opposite bright points with the diffraction order i can be measured to be ($D_1 = 7.356 \text{ nm}^{-1}$, $D_2 = 12.542 \text{ nm}^{-1}$, $D_3 = 14.666 \text{ nm}^{-1}$, and $D_5 = 21.986 \text{ nm}^{-1}$), indicating that the error will be smaller if the higher order diffraction points was selected (Fig. R28g). Thus, the error of twist angle ($\text{Error}_{\theta\text{-SAED}}$) should be twice of the value of $\varphi_{\text{SAED}i}$ due to the two measurements. In our work, the 2nd diffraction point was selected, and the $\text{Error}_{\theta\text{-SAED}}$ can be canulated to be 0.44° , which is much accurate than that of the OM measurement (0.8°).

$$\text{Error}_{\theta\text{-SAED}} = 2 \times \varphi_{\text{SAED}2} \approx 0.44^\circ \quad (11)$$

Fig. R28 Error analysis of TB-MoS₂ twist angle measurement based on SAED method. **a** Typical SAED pattern of the TB-MoS₂. **b** Illustration of the error determined by the SAED points. **c-f** Position dependent diffraction intensity of four diffraction points shown in **a**. **g** The errors results of SAED method. Scale bar 5 nm^{-1} .

Compared with the above error analysis based on SAED and OM, we can obtain the $\text{Error}_{\theta\text{-OM}}$ of 0.8° by OM (Comment 2) and $\text{Error}_{\theta\text{-SAED}}$ of 0.44° by SAED (Comment 7). As we discussed, the error of OM and SAED is mainly dependent on the size of the TB-MoS₂ domain (L) and distance from the two opposite bright points (D_i). Thus, the increase in the size or the selection of higher diffraction order can result in the higher accuracy of the measurement of the twist angle. Therefore, we have modified the corresponding descriptions and added the calculation process in the revised manuscript and Supplementary Information.

Changes to the revised Supplementary Information are shown below (In Page 6-7).

Supplementary Note 2. Characterizing the twist angle based on SAED.

The SAED is an accurate method for measuring the twist angle of TB-MoS₂, which is widely used to determine the twist angle of 2D materials, such as TB-graphene and TB-TMDCs. Firstly, based on the STEM-HAADF image of TB-MoS₂, we can preliminarily identify the twist angle θ_{STEM} range belonging to $0^\circ\sim 30^\circ$, $30^\circ\sim 60^\circ$, $60^\circ\sim 90^\circ$, or $90^\circ\sim 120^\circ$. Then, we measure the direction of two opposite diffraction points of each layer MoS₂, and the twist angle θ_{SAED} can be measured in the range of 0° to 30° . Based on the following equations, the twist angle θ_{STEM} can be finally determined.

$$\theta_{\text{SAED}} = \theta_{\text{STEM}} \quad (0 < \theta_{\text{STEM}} \leq 30^\circ) \quad (3)$$

$$\theta_{\text{SAED}} = 60^\circ - \theta_{\text{STEM}} \quad (30^\circ < \theta_{\text{STEM}} \leq 60^\circ) \quad (4)$$

$$\theta_{\text{SAED}} = \theta_{\text{STEM}} - 60^\circ \quad (60^\circ < \theta_{\text{STEM}} \leq 90^\circ) \quad (5)$$

$$\theta_{\text{SAED}} = 120^\circ - \theta_{\text{STEM}} \quad (90^\circ < \theta_{\text{STEM}} \leq 120^\circ) \quad (6)$$

Based on the measurement, the direction of two opposite diffraction points of each MoS₂ layer is much higher than the OM method as discussed in Supplementary Note 1. Taking a SAED pattern of 21.9° -TB- MoS₂ as an example (as shown Supplementary Fig. 10a), the weight lines connecting the opposite diffraction points intersect with each other at a specific twist angle.

The error (φ_{SAED}) based on the SAED pattern can be determined by the full width at half

maximum (FWHM) of the blurred diffraction points and the distance between each opposite bright point in a single SAED pattern (Supplementary Fig. 10b).

$$\varphi_{\text{SAED}i} = \arctan \frac{FWHM_i}{D_i} \quad (7)$$

where i is the diffraction order ($i=1,2,3,4\dots$), D_i is the distance from the two opposite bright points under the diffraction order of i . In order to calculate the error based on the SAED method, we measure FWHMs of four groups of the position dependent diffraction intensity plot (1st, 2nd, 3rd, and 5th, Supplementary Fig. 10c-f) with the average FWHMs of 0.0049 nm^{-1} . While the distance from the two opposite bright point with the diffraction order i can be measured to be ($D_1 = 7.356 \text{ nm}^{-1}$, $D_2 = 12.542 \text{ nm}^{-1}$, $D_3 = 14.666 \text{ nm}^{-1}$, and $D_5 = 21.986 \text{ nm}^{-1}$), indicating that the error will be smaller if higher order diffraction points are selected. Thus, the error of twist angle ($\text{Error}_{\theta\text{-SAED}}$) should be the twice of the value of $\varphi_{\text{SAED}i}$ due to the measurement for two times. In our work, the 2nd diffraction point is selected, and the $\text{Error}_{\theta\text{-SAED}}$ is calculated to be 0.44° , which is more accurate than the $\text{Error}_{\theta\text{-OM}}$ of 0.8° .

$$\text{Error}_{\theta\text{-SAED}} = 2 \times \varphi_{\text{SAED}2} \approx 0.44^\circ \quad (8)$$

Supplementary Fig. 10 Error analysis of twist angle measurement based on SAED method. **a** Typical SAED pattern of the TB-MoS₂. **b** Illustration of the error determined by the SAED points. **c-f** Position dependent diffraction intensity of four diffraction points shown in **a**. **g** The errors results of SAED method. Scale bar 5 nm^{-1} .

Comment 8:

Fig.5b-c. The statistical detail of the TB-MoS₂ density is not clear. The number of samples (and growth) and the statistic area of each sample are missing. The calculation method of error bars in the figures is ungiven. The twist-angle distribution of TB-MoS₂ under each growth condition is missing (are they the same or not?).

Response: We understand the reviewer's concern about the calculation of density, yield, as well as corresponding error bar, and appreciate much for the valuable suggestion. The point-by-point response has been listed as below.

(1) The calculation method of TB-MoS₂ density and error bar.

To calculate the density of TB-MoS₂, the OM under the 20X objective was selected due to TB-MoS₂ can be easily identified under the 20X objective. The number of TB-MoS₂ and bilayer MoS₂ based on the OM was calculated. The consistent statistic area was calculated by the OM with the scale bar. The density (D_{TB}) and yield (Y_{TB}) can be calculated by the total number of TB-MoS₂ samples (N_{TB}), the total number of bilayer MoS₂ samples (N_{BL}), and the total area (S).

$$D_{TB} = \frac{N_{TB}}{S} \quad (12)$$

$$Y_{TB} = \frac{N_{TB}}{N_{BL}} \times 100\% \quad (13)$$

To make the results accurate, five samples were adopted under the same growth conditions. The mean value of density (\bar{D}), yield (\bar{Y}), and corresponding standard deviation (δ) were calculated by the following equations.

$$\bar{D} = \frac{\sum_{i=1}^5 D_{TBi}}{5} \quad (14)$$

$$\delta_D = \sqrt{\frac{\sum_{i=1}^5 (D_i - \bar{D})^2}{5}} \quad (15)$$

$$\bar{Y} = \frac{\sum_{i=1}^5 Y_{TBi}}{5} \quad (16)$$

$$\delta_Y = \sqrt{\frac{\sum_{i=1}^5 (Y_i - \bar{Y})^2}{5}} \quad (17)$$

Therefore, the average density and yield with corresponding error bars (standard deviation) can be obtained. We have added the calculation process in the revised manuscript and Supplementary Information.

Changes to the revised manuscript are shown below (In Page 17).

The density and yield are adopted to evaluate the parameters of the CVD method on TB-MoS₂. All the TB-MoS₂ shows a concentrated distribution with a bottom layer, which ensures a more reliable statistic results on density (Supplementary Fig. 20). As mentioned in Supplementary Note 4, the density is defined as the number of TB-MoS₂ per unit area, while the yield is dedicated to the proportion of TB-MoS₂ in the bilayer MoS₂.

Changes to the revised Supplementary Information are shown below (In Page 9).

Supplementary Note 4. The density and yield calculation of TB-MoS₂.

The OM under the 20X objective was selected due to the TB-MoS₂ can be easily identified under the 20X objective. The number of TB-MoS₂ and bilayer MoS₂ based on the OM under the 20X objective was calculated. The density (D_{TB}) and yield (Y_{TB}) can be calculated by the total number of TB-MoS₂ samples (N_{TB}), the total number of bilayer MoS₂ samples (N_{BL}), and the total area (S).

$$D_{TB} = \frac{N_{TB}}{S} \quad (9)$$

$$Y_{TB} = \frac{N_{TB}}{N_{BL}} \times 100\% \quad (10)$$

To make the results more accurate, five samples were adopted under the same growth conditions. The average density (\bar{D}), standard deviation (δ_D , density error), yield (\bar{Y}), and corresponding standard deviation (δ_Y , yield error) were calculated by the following equations.

$$\bar{D} = \frac{\sum_{i=1}^5 D_{TBi}}{5}, \delta_D = \sqrt{\frac{\sum_{i=1}^5 (D_i - \bar{D})^2}{5}} \quad (11)$$

$$\bar{Y} = \frac{\sum_{i=1}^5 Y_{TBi}}{5}, \delta_Y = \sqrt{\frac{\sum_{i=1}^5 (Y_i - \bar{Y})^2}{5}} \quad (12)$$

(2) Twist-angle distribution of TB-MoS₂ under different growth conditions.

The statistical distribution of twist angles under different growth conditions is shown in Fig. R29. As noticed, all the samples show a similar Gaussian distribution both in the range of 0° to 60° and 60° to 120°. The twist angles near 30° and 90° are significantly larger than that of the other angles. We have added the missing distribution and description in the revised manuscript and Supplementary Information.

Fig. R29 Statistical distribution of twist angles under different growth conditions. a The TB-MoS₂ synthesized under a gas flow rate of 90 sccm and a NaCl to MoO₃ molar ratio of 20.

b The TB-MoS₂ synthesized under a gas flow rate of 70 sccm and a NaCl to MoO₃ molar ratio of 20. **c** The TB-MoS₂ synthesized under a gas flow rate 30 sccm and a NaCl to MoO₃ molar ratio of 20. **d** The TB-MoS₂ synthesized under a gas flow rate of 50 sccm and a NaCl to MoO₃ molar ratio of 10.

Changes to the revised manuscript are shown below (In Page 16).

As noticed, the typical Gaussian distribution is observed both in the range of 0° to 60° and 60° to 120°. The numbers of twist angles near 30° and 90° are significantly larger than that of the other angles. The similar distribution also can be verified under different growth conditions as shown in Supplementary Fig. 18.

Changes to the revised Supplementary Information are shown below (In Page 32).

Supplementary Fig. 18 Statistical distribution of twist angles under different growth conditions. **a** The TB-MoS₂ synthesized under a gas flow rate of 90 sccm and a NaCl to MoO₃ molar ratio of 20. **b** The TB-MoS₂ synthesized under a gas flow rate of 70 sccm and a

NaCl to MoO₃ molar ratio of 20. **c** The TB-MoS₂ synthesized under a gas flow rate 30 sccm and a NaCl to MoO₃ molar ratio of 20. **d** The TB-MoS₂ synthesized under a gas flow rate of 50 sccm and a NaCl to MoO₃ molar ratio of 10.

Comment 9:

The current results show that the density and yield of TB-MoS₂ monopoly increase with the slat ratio and decrease with the flow rate. Is there a limit to this trend?

Response:

Thank you for raising this insight question. Based on your suggestion, we have carried out a series of experiments on the gas flow rate and molar ratio of NaCl to MoO₃. Based on the experiments results, we calculated the density and yield of all the TB-MoS₂ samples, as shown in Fig. R30. The results indicated that both parameters have a limit to this trend.

Fig. R30 Key parameters for CVD grown TB-MoS₂. **a, b** Density and yield of TB-MoS₂ under different gas flow rate and molar ratio of NaCl to MoO₃, respectively.

Fig. R6 Typical BF-OM of synthesized products under different gas flow rate. a-f The typical OM synthesized under different gas flow rate of 110, 90, 70, 50, 30, and 10 sccm. Scale bars: 100 μm for the OM under 20X objective, 10 μm for the small area under 50X objective. The TB-MoS₂ was synthesized with fixed NaCl to MoO₃ of 20.

Fig. R7 Typical BF-OM of synthesized products under different molar ratio of NaCl to MoO₃. a-e The typical OM images of the synthesized MoS₂ under different molar ratio of NaCl to MoO₃ of 30, 20, 10, 3, and 0. Scale bars: 100 μm for the OM under 20X objective, 10 μm for the small area under 50X objective.

For the gas flow rate, the suitable value of 30 sccm to 90 sccm would be an enhanced growth range of TB-MoS₂. We can notice that the yield and density show the highest under a gas flow rate of 50 sccm, exhibiting a highest yield and density of 17.2% and 28.9 pieces/mm², respectively (Fig. R30a). The high nucleation density under 30 sccm gas flow rate results in a low yield of TB-MoS₂ compared with 70 sccm. When the gas flow rate is higher than 110 sccm or lower than 10 sccm, we cannot obtain the TB-MoS₂ samples (Fig. R6). With the increase of NaCl to MoO₃ molar ratio from 0 to 20, the yield and density increased (Fig. R30b). When the molar ratio of NaCl to MoO₃ up to 30, we cannot synthesize the TB-MoS₂, and the MoS₂ shows a very large size compared with samples synthesized under smaller molar ratios of NaCl to MoO₃ (Fig. R7).

Therefore, we can notice that both parameters have a limit to the trend of yield and density. We have added the corresponding discussions in the revised manuscript.

Changes to the revised manuscript are shown below (In Page 17-18).

As shown in Fig. 5b and Supplementary Fig. 21, with the increase in the gas flow rate, the density and the yield of TB-MoS₂ first increased and then decreased. The highest yield and density of 17.2% and 28.9 pieces/mm² are obtained under a gas flow rate of 50 sccm, which is much larger than the reported TB-TMDCs (Supplementary Table 1). Besides, the TB-MoS₂ can be well synthesized under gas flow rates of 30 to 90 sccm, indicating that the gas flow rate of 30 sccm to 90 sccm would be an enhanced growth range.

As shown in Fig. 5c and Supplementary Fig. 22, with the increase in the molar ratio from 0 to 20, the density increases from 0 to 28.9 pieces/mm², while the yield increases from 0 to 17.2%, respectively. The TB-MoS₂ cannot be synthesized when the molar ratio of NaCl to MoO₃ reaches 30, indicating that the molar ratio 3 to 20 would be a suitable growing parameter range.

Comment 10:

The temperature is also a key parameter in CVD growth but is not involved in the discussion of this research.

Response: Thank you for raising this question. As we know, temperature is also a key parameter in CVD growth, wherein the different reaction temperatures may result in different morphology [ACS Nano 12, 635-643 (2018); Nat Commun 10, 598 (2019); Nat Commun 12, 809 (2021); ACS Nano 13, 8265-8274 (2019)]. Based on previous report, the preparation of MoS₂ usually requires a large reaction temperature range of 600 °C to 1000 °C [Nature 556, 355–359 (2018); Nat Commun 10, 598 (2019); Nature 605, 69-75 (2022)]. However, based on our experiment results, we find that the TB-MoS₂ can only be well synthesized under a very small reaction temperature range from 770 °C to 790 °C. As shown in Fig. R12, TB-MoS₂ can be well synthesized under the reaction temperature of 780 °C. With the reaction temperature increase to 790 °C, the sample of TB-MoS₂ dramatically decreases. There has no TB-MoS₂ at higher temperatures than 800 °C. When the reaction temperature decreases to 770 °C, the numbers of TB-MoS₂ also dramatically decreases. Large-area monolayer MoS₂ can be well synthesized under 760 °C. Thus, we selected the reaction temperature of 780 °C as the optimized synthesis parameter.

Fig. R12 Typical BF-OM of the synthesized products under different reaction

temperatures. a-e Typical OM images of the synthesized MoS₂ under different reaction temperature of 850 °C, 800 °C, 790 °C, 780 °C, 770 °C, and 760 °C. The molar ratio of NaCl to MoO₃ is fixed to 20, and the gas flow rate is fixed to 50 sccm. Scale bars: 100 μm for the OM under 20X objective, 10 μm for the small area under 50X objective.

Therefore, to address the reviewer's concern, we have added corresponding experiment results and discussion in the revised manuscript and Supplementary Information.

Changes to the revised manuscript are shown below (In Page 16).

Here, the TB-MoS₂ can be well synthesized under a narrow reaction temperature range of 770 °C to 790 °C, with an optimal temperature of 780 °C (Supplementary Fig. 17).

Changes to the revised Supplementary Information are shown below (In Page 31).

Supplementary Fig. 17 Typical BF-OM of the synthesized products under different reaction temperatures. a-e Typical OM images of the synthesized MoS₂ under different reaction temperature of 850 °C, 800 °C, 790 °C, 780 °C, 770 °C, and 760 °C. The molar ratio of NaCl to MoO₃ is fixed to 20, and the gas flow rate is fixed to 50 sccm. Scale bars: 100 μm for the OM under 20X objective, 10 μm for the small area under 50X objective.

Comment 11:

Line 409, the pressure during the growth is missing.

Response:

Thanks for the reviewers' comment. In this work, atmospheric pressure was adopted in the CVD process. All the TB-MoS₂ samples were synthesized under the atmospheric pressure. We have added the missing information in the revised manuscript.

Changes to the revised manuscript are shown below (In Page 22).

The TB-MoS₂ is grown by atmospheric pressure CVD method on SiO₂/Si substrate in a single temperature tube furnace as depicted in Fig. 1a

We here would like to thank all reviewers again for the detailed review and constructive suggestions that help us improve the manuscript.

REVIEWER COMMENTS

Reviewer #1 (Remarks to the Author):

The authors have carefully addressed most of my comments. One of the remaining minor concerns is the response to my Comment 5. The authors made statements like "Introducing NaCl can affect the thermodynamics and kinetics in the CVD system (the detailed analysis can be found in Supplementary Note 5)." Citations are required here.

For other statements like: "Supplementary Note 6. Kinetics analysis of confined space.", "introducing a confined space provides controllability of kinetics", etc, can not be fully supported by the provided data. I can not find any kinetics analysis or study showing the controllability of kinetics. The authors should be careful when making such statements.

Reviewer #2 (Remarks to the Author):

The author added more discussion of the growth mechanism in the revised manuscript, including the effect of NaCl, confined space, temperature, as well as turbulent flow. New simulation results are added to prove the turbulent flow.

However, in the revised manuscript, there is still no evidence for why and how to avoid the thermodynamically favorable AB-stacked MoS₂ and increase the proportion of twisted bilayer MoS₂ to AB-stacked MoS₂. The revised supplementary Fig. 17 and Supplementary Fig. 21 show some growth conditions corresponding to good TB-MoS₂, monolayer MoS₂, as well as thicker MoS₂, but I didn't find a growth condition corresponding to AB-stacked bilayer MoS₂ in the revised manuscript and supplementary materials. It seems like that the optimized growth condition of bilayer MoS₂ is the same of TB-MoS₂, which is confused me. So I have some concerns on the mechanisms on tuning the twist structure and optimizing the yield of TB-MoS₂. In another words, as the authors would like to highlight this method break the thermodynamically unfavourable structure, they should provide more contrast experimental results, including normal bilayer MoS₂.

The revised manuscript highlights the confined space and turbulent air flow, but it's lack of the bridge between CFD simulation to the thermodynamic influence to twisted structures. How does the turbulent flow influence the nuclei orientation?

I also suggest the authors to give an insight picture illustrating the growth mechanism in the main text, such as the effect of confined space/turbulent flow on twisting MoS₂. Current Fig.1a is somewhat unscientific.

Reviewer #3 (Remarks to the Author):

The authors addressed most of my concerns. Their reply about the “vibration intensity” seems wrong. I personally never see such a definition, nor is it mentioned in those citations given by the author. It needs to be carefully corrected before being published.

Response to referees

Response to Reviewer 1

The authors have carefully addressed most of my comments. One of the remaining minor concerns is the response to my Comment 5. The authors made statements like "Introducing NaCl can affect the thermodynamics and kinetics in the CVD system (the detailed analysis can be found in Supplementary Note 5)." Citations are required here.

For other statements like: "Supplementary Note 6. Kinetics analysis of confined space.", "introducing a confined space provides controllability of kinetics", etc, can not be fully supported by the provided data. I can not find any kinetics analysis or study showing the controllability of kinetics. The authors should be careful when making such statements.

Response:

We sincerely appreciate the reviewer for carefully reading our manuscript and providing invaluable suggestions to further improve it. The point-by-point response has been listed below.

(1) Citations on the NaCl affect the thermodynamics and kinetics.

In our previous works, the related references were cited in the section of "Supplementary Note 5. Thermodynamically and kinetics analysis of NaCl.". To address the reviewer's concern, we have added corresponding references in the revised manuscript.

Changes to the revised manuscript are shown below (In Page 18).

Introducing NaCl can affect the thermodynamics and kinetics in the CVD system (the detailed analysis can be found in Supplementary Note 5) [*Nature* **556**, 355-359 (2018)].

(2) Controllability of kinetics.

In effect, the kinetics analysis of confined space in the revised manuscript and Supplementary Note 6 indicated that introducing a confined space can affect the growth kinetics. As the reviewer pointed out, the kinetics was uncontrollable and could only be affected by the confined space. Therefore, we have changed the corresponding statements in the revised manuscript to address the reviewer's concern.

Changes to the revised manuscript are shown below (In Page 19).

Here, introducing a confined space alters the kinetics and consequently realizes anisotropic growth of the TB-MoS₂ with the thermodynamically unfavourable twisted stacking structures.

Response to Reviewer 2

The author added more discussion of the growth mechanism in the revised manuscript, including the effect of NaCl, confined space, temperature, as well as turbulent flow. New simulation results are added to prove the turbulent flow.

However, in the revised manuscript, there is still no evidence for why and how to avoid the thermodynamically favorable AB-stacked MoS₂ and increase the proportion of twisted bilayer MoS₂ to AB-stacked MoS₂. The revised supplementary Fig. 17 and Supplementary Fig. 21 show some growth conditions corresponding to good TB-MoS₂, monolayer MoS₂, as well as thicker MoS₂, but I didn't find a growth condition corresponding to AB-stacked bilayer MoS₂ in the revised manuscript and supplementary materials. It seems like that the optimized growth condition of bilayer MoS₂ is the same of TB-MoS₂, which is confused me. So I have some concerns on the mechanisms on tuning the twist structure and optimizing the yield of TB-MoS₂. In another words, as the authors would like to highlight this method break the thermodynamically unfavourable structure, they should provide more contrast experimental results, including normal bilayer MoS₂.

The revised manuscript highlights the confined space and turbulent air flow, but it's lack of the bridge between CFD simulation to the thermodynamic influence to twisted structures. How does the turbulent flow influence the nuclei orientation?

I also suggest the authors to give an insight picture illustrating the growth mechanism in the main text, such as the effect of confined space/turbulent flow on twisting MoS₂. Current Fig.1a is somewhat unscientific.

Response:

We sincerely appreciate the reviewer for carefully reading our work and providing invaluable suggestions to improve it. Our point-by-point responses are shown below.

(1) Optimized growth condition of bilayer MoS₂ and TB-MoS₂.

In this work, the optimized growth condition of bilayer MoS₂ is indeed similar to the TB-MoS₂. As noticed in Fig. 5 b-c and Supplementary Fig. 26, we can find that a higher yield of TB-MoS₂ can be obtained under a higher proportion of bilayer MoS₂, which is consistent with the reported twisted bilayer materials. For example, the disoriented stacking graphene (10%)

can be observed under a high coverage of bilayer graphene (99%) [*ACS Nano* **6**, 8241-8249 (2012)], 20-30% twisted graphene obtained at a high coverage of bilayer graphene of 90% [*Chem. Mater.* **28**, 4583-4592 (2016)], 7% twisted graphene obtained at a high coverage of bilayer graphene of 77% [*Chem. Mater.* **21**, 7852-7859 (2018)], and 5% twisted MoS₂ can be synthesized under a high yield of bilayer MoS₂ of 30% [*Nat. Commun.* **5**, 4966 (2014)].

As we discussed in the manuscript and Supplementary Information, the bilayer MoS₂ with high proportions are obtained under gas flow rates of 30, 50, 70, and 90 sccm, or molar ratio of NaCl to MoO₃ of 10 and 20. Meanwhile, with these parameters, we also obtain abundant TB-MoS₂ on SiO₂/Si substrate, which indicates that the TB-MoS₂ was synthesized at high proportions of bilayer MoS₂. The suitable growth conditions of bilayer MoS₂ are preferred to form the twisted structures under the turbulent flow caused by a confined space setup.

As we know, the synthesis conditions, such as reaction temperature, gas flow rate, the molar ratio of salt to metal precursors, and the setup might be the important parameters affecting the sample morphology in the CVD system [*Nature* **556**, 355-359 (2018); *Nature* **605**, 69-75 (2022); *Nano Res.* **7**, 511-517 (2014); *J. Am. Chem. Soc.* **143**, 18103-18113 (2021); *Cryst. Growth Des.* **18**, 1012-1019 (2018)]. Here, we have carried out a series of contrast experiments with/without confined space under different synthesis conditions to support this phenomenon, as shown in Supplementary Fig. 21 and Supplementary Fig. 22. In our work, with the increase in the gas flow rate, the morphology of the MoS₂ can be evolutive from monolayer to bilayer without the assistance of confined space (Supplementary Fig. 21). Additionally, the layer of MoS₂ increased with the introduction of a confined space, as summarized in Table R1. The consistent phenomenon can also be observed in the growth conditions in terms of the molar ratio of NaCl to MoO₃, as shown in Supplementary Fig. 22 and Table R1. Therefore, high-proportioned bilayer MoS₂ can be synthesized in the wide-growth window with the assistance of a confined space. The backflows introduced by confined space will increase the collision rate of molecules to molecules and molecules to substrate. As a result, there is enough energy for the *in-situ* twisted nucleation under the gas phase, resulting in the formation of TB-MoS₂. In other words, under the optimized conditions for growing high proportion of bilayer MoS₂, there are higher collision rate for the increased molecules in the confined space, resulting in a higher yield of Tb-MoS₂. Thus, it is hard to obtain pure AB-stack bilayer MoS₂ in the confined space due to the existence of the turbulent flow. Hence, from the experimental results and corresponding analysis, we can better understand the optimized growth conditions of bilayer MoS₂, which is similar to the TB-MoS₂.

The related figures and table are listed below for your easy reference.

Supplementary Fig. 26 Proportion of monolayer MoS₂, bilayer MoS₂, multi-layer MoS₂, and non-MoS₂ in the as-grown samples.

Fig. 5 b-c Key parameters for CVD grown TB-MoS₂. a, b Density and yield of TB-MoS₂ under different gas flow rates and molar ratio of NaCl to MoO₃, respectively.

Table R1 Summary of the experiment results with/without assistance of confined space.

Gas flow rate (sccm)	Dominating morphology of MoS ₂		Molar ratio of NaCl to MoO ₃	Dominating morphology of MoS ₂	
	Without confined space	With confined space		Without confined space	With confined space
10	Monolayer	Multi-layer	0	Non	Non
30	Monolayer	Bilayer	3	Monolayer	Monolayer
50	Monolayer	Bilayer	10	Monolayer	Bilayer
70	Monolayer	Bilayer	20	Monolayer	Bilayer
90	Monolayer Bilayer	Bilayer Multi-layer	30	Monolayer Multi-layer	Multi-layer
110	Monolayer Bilayer	Multi-layer			

Supplementary Fig. 21 Typical BF-OM of synthesized products under different gas flow rates. **a-f** The typical OM images synthesized under different gas flow rates without confined space. **g-l** The typical OM images synthesized under different gas flow rates with confined space. Scale bars: 100 μm for the OM under 20X objective, 10 μm for the small area under 50X objective. The TB-MoS₂ was synthesized with fixed ratio of NaCl to MoO₃ of 20.

Supplementary Fig. 22 Typical BF-OM of synthesized products under different molar ratios of NaCl to MoO₃. **a-e** The typical OM images of the synthesized MoS₂ under different molar ratios of NaCl without confined space. **f-j** The typical OM images of the synthesized MoS₂ under different molar ratios of NaCl with confined space Scale bars: 100 μm for the OM under 20X objective, 10 μm for the small area under 50X objective.

(2) The turbulent flow influences on nucleation process.

We believe that the turbulent flow influenced the twisted nucleation process at the gas phase nucleation stage. The deposited MoO_x molecular clusters are relatively small, which are easily

reduced into MoS₂ clusters in the initial growth stage. It can also be seen in some previous works [*Nat. Commun.* **7**, 12206 (2016); *Chem. Mater.* **33**, 3241-3248 (2021); *Cryst. Growth Des.* **18**, 1012-1019 (2018); *ACS Appl. Mater. Interfaces* **13**, 59154-59163 (2021)]. Then, the MoS₂ clusters can be twisted under the effect of turbulent flow caused by the confined space. Thus, the twisted nucleus is deposited on the SiO₂/Si substrate, and further continuous edge growth, leading to the formation of layered MoS₂ flakes. Here, we can notice some twisted bilayer, trilayer, and multilayer MoS₂ on SiO₂/Si substrate, as shown in Fig. R1. As noticed, all the MoS₂ samples exhibited twist structures, and the twist structures were formed, which are sharing the same nucleation site. Therefore, we believe that the nuclei orientation occurs **randomly** due to the turbulent flow introduced by the confined space CVD system. Besides, as noticed from Supplementary Fig. 21 and Supplementary Fig. 22, we can conclude that various layer numbers of MoS₂ flakes were synthesized and showed a random distribution on the SiO₂/Si substrate, which indicated the gas phase nucleation process of MoS₂ in the CVD synthesis of TB-MoS₂. Therefore, based on the above analysis, we believe the nucleation process of TB-MoS₂ mainly occurred **randomly** due to the turbulent flow influences on the gas phase nucleation stage.

Fig. R1 The BF-OM and DF-OM of the twisted bilayer, twisted trilayer, and twisted multilayer MoS₂ on SiO₂/Si substrate.

Apart from the experiment results, we have tried to find more evidence to describe the nucleation process based on the simulation results. However, there are some challenges in the investigation of the nucleation process as follows.

For the CFD simulations, the backflow under macroscopical results can reflect the gas states in the CVD system. However, the accurate location of backflow and the corresponding gas state (such as velocity vector maps) under nanoscale can not be obtained with the fixed mesh size of $1\text{ mm} \times 1\text{ mm}$. Although the higher precision modes can be established with the mesh size to the nanoscale (same order as the nucleation size), the computing resource and time will be increased exponentially. In addition, the distribution rules of the physical field and the flow field are almost consistent both at the nanoscale and millimeter-scale when the mesh size reaches a certain density. Therefore, in this work, we mainly prove the backflow existed under the CVD system with the confined space. The evidence for the twisted nucleation influenced by a turbulent flow should be further investigated by other simulation methods, such as Molecular Dynamics.

We attempted to simulate the gas phase nucleation and growth process of TB-MoS₂ using molecular dynamics. However, the traditional molecular dynamics enable spatial scale simulations with tens of thousands of atoms for a long time scale. Therefore, it falls short in describing the formation of chemical bonds during nucleation. On the other hand, ab initio molecular dynamics allow for the simulation of chemical bond breaking and formation. But, its limitation lies in the nanosecond time scale, making it challenging to accurately depict the full nucleation and growth processes of TB-MoS₂. Additionally, the periodicity of TB-MoS₂ results in a significantly increased unit cell compared to pristine MoS₂, which adds the complexity to the simulation.

As a consequence, we are unable to provide more precise results based on molecular dynamics simulations to substantiate the exact twisted nucleation during the gas phase nucleation process at the current stage.

(3) Growth mechanism diagram on the effect of the turbulent flow on TB-MoS₂.

In our previous works, Fig. 1a is the illustration of the CVD setup for the synthesis of TB-MoS₂, which can help to understand the confined-space CVD setup. Based on our above

mentioned analysis, we have added another picture illustrating the growth mechanism for the affection by the turbulent flow, as shown in Fig. 5e. Besides, to address the reviewer's concern, we have added the corresponding discussion on the growth mechanism about turbulent flow effect on the twisted nucleation process in the revised manuscript.

Fig. 5e Growth mechanism diagrammatic of twisted nucleation process under turbulent flow.

Changes to the revised manuscript are shown below (In Page 5 and Page 22).

The deposited precursor molecular clusters are relatively small and limited, which are easily reduced into MoS₂ clusters in the initial growth stage. With the help of the turbulent flow induced by the confined space, the random motion of MoS₂ clusters will be enhanced under the gas phase (Fig. 5e). The collision rate of molecules to molecules and molecules to substrate will further be increased, which produces enough energy for the *in-situ* twisted nucleation under the gas phase. Then, the MoS₂ clusters can be twisted under the turbulent flow. The size of the twisted nucleus is further enlarged by the interaction with vapor atoms, and consequently deposited randomly on the SiO₂/Si substrate. The twisted MoS₂ nucleus will continuously grow along the edges, leading to the formation of layered MoS₂ flakes (Fig. 5e).

Fig. 5 | Key parameters for CVD grown TB-MoS₂. **a** Statistical distribution of twist angles based on OM of as-grown TB-MoS₂. **b-c** Density and yield of TB-MoS₂ under different gas flow rate and molar ratio of NaCl to MoO₃, respectively. **d** Flow chart of the salt-assisted growth process for the reconfiguring nucleation and growth of TB-MoS₂ via the space-

confined CVD method. **e** Growth mechanism diagrammatic of twisted nucleation process under turbulent flow. Scale bars: 100 μm in **d**.

Changes to the revised Supplementary Information are shown below (In Page 35-37).

Supplementary Fig. 21 Typical BF-OM of synthesized products under different gas flow rates. a-f The typical OM images synthesized under different gas flow rates without confined

space. **g-i** The typical OM images synthesized under different gas flow rates with confined space. Scale bars: 100 μm for the OM under 20X objective, 10 μm for the small area under 50X objective. The TB-MoS₂ was synthesized with fixed ratio of NaCl to MoO₃ of 20.

Supplementary Fig. 22 Typical BF-OM of synthesized products under different molar ratios of NaCl to MoO₃. **a-e** The typical OM images of the synthesized MoS₂ under different molar ratios of NaCl without confined space. **f-j** The typical OM images of the synthesized MoS₂ under different molar ratios of NaCl with confined space Scale bars: 100 μm for the OM under 20X objective, 10 μm for the small area under 50X objective.

Response to Reviewer 3

The authors addressed most of my concerns. Their reply about the “vibration intensity” seems wrong. I personally never see such a definition, nor is it mentioned in those citations given by the author. It needs to be carefully corrected before being published.

Response:

We sincerely appreciate the reviewer for carefully reading our work and providing invaluable suggestions to improve it. The point-by-point responses are shown below.

Firstly, we apologize for our ambiguous expressions. In our initial manuscript, the ambiguous description "*Compared to the monolayer MoS₂, the TB-MoS₂ shows a higher vibration intensity, while the frequency of E_{2g}¹ mode decreases ($\Delta\omega = 2.4 \text{ cm}^{-1}$) and the A_{1g} mode increases ($\Delta\omega = 2.1 \text{ cm}^{-1}$)*" can not express our correct meaning. Therefore, in the previous comment (Comment 3), the reviewer raised a question "*Please clarify the “vibration intensity” and its relation with the Raman shift*". Indeed, in our initial manuscript, we want to point out the independent experiments phenomenon, including both "bilayer MoS₂ shows a higher Raman intensity than monolayer" and "Raman shift occurred between the monolayer and bilayer MoS₂". We are not implying a relationship between "vibration intensity" and "Raman shift".

Therefore, we clarified the statements in the revised manuscript as shown below:

*"Compared to monolayer MoS₂, TB-MoS₂ shows a higher Raman intensity attributed to the strongly increased Raman intensity resulting from the increase in layer number [Phys. Rev. B **97**, 165409 (2018); Nano Lett. **15**, 1356-1361 (2015); Phys. Rev. B **80**, 125422 (2009); Nanoscale **7**, 8135-8141 (2015); ACS Nano **6**, 7381-7388 (2012)]."*

*"Besides, the frequency of the E_{2g}¹ mode decreases ($\Delta\omega = 2.4 \text{ cm}^{-1}$), while the frequency of the A_{1g} mode increases ($\Delta\omega = 2.1 \text{ cm}^{-1}$). This is caused by the long-range Coulombic interaction between the Mo atoms. Additionally, the enhancement of interlayer interactions leads to restoring forces on the increased S atoms [J. Raman Spectrosc. **44**, 92-96 (2013)]."*

Thus, we have explained the independent experiment's phenomenon "bilayer MoS₂ shows a higher Raman intensity than monolayer" and "Raman shift occurred between the monolayer

and bilayer MoS₂" in the last response to reference (Comment 3). Besides, based on the reviewer's comment, we have tried to establish a correlation between "Raman intensity" and "Raman shift" but our attempts were unsuccessful. However, the Raman intensity of MoS₂ has been defined to be $I = \int_0^d |E_{ab}(x)E_{em}(x)|^2 dx$, and some related definitions of Raman intensity of graphene and TMDCs are shown in Table R2. Thus, we can conclude that the Raman intensity increased with the increase of the layer number from this equation. Therefore, the Raman intensity of the bilayer is higher than the monolayer MoS₂.

Table R2 Summary of the definition of Raman intensity.

Materials	Definition of Raman intensity	Ref.
Graphene	$I = \int_0^{d_1} t\gamma ^2 \Delta y$ d_1 : Thickness of graphene; t : Total amplitude of the electric field; γ : Summation of infinite transmitted light.	The complete derivation process is given in the main text. Appl. Phys. Lett. 92, 043121 (2008)
Graphene	$F = N \int_0^{d_1} F_{ab} F_{sc} ^2 dx$ d_1 : Thickness of graphene; N : Normalization factor; F_{ab} : Net absorption term; F_{sc} : Net scattering term.	The complete derivation process is given in the main text. Phys. Rev. B 80 , 125422 (2009)
Graphene	$I \propto \int_0^{d_i} F_L(Z_i)F_R(Z_i) ^2 dZ_i$ d_i : Thickness; F_L, F_R : Respective enhancement factors for laser excitation and Raman signals, respectively.	— Nanoscale 7 , 8135-8141 (2015)
MoS ₂	$I = \int_0^{d_i} F_{ex}(x)F_{sc}(x) ^2 dx$ d_i : Total thickness; $F_{ex}(x), F_{sc}(x)$: Electric field amplitudes excitation and scattering light, respectively.	The complete derivation process is given in SI. ACS Nano 6 , 7381-7388 (2012)
MoS ₂	$I = \int_0^d E_{ab}(x)E_{em}(x) ^2 dx$ d : Thickness of the flake; $E_{ab}(x)$: Electric field amplitudes within the flakes; $E_{em}(x)$: Light emitting out the flakes.	— Nano Lett. 15 , 1356-1361 (2015)
WS ₂	$F = \int_0^{t_1} F_{abs} F_{sc} ^2 dy$ t_1 : Thickness; F_{abs} : net amount of laser absorption; F_{sc} : enhancement factor of the scattered light.	— Nanoscale 12 , 6064–6078 (2020)

Again, we would like to thank all reviewers for the detailed reviews and constructive suggestions that help us improve the manuscript.

REVIEWERS' COMMENTS

Reviewer #2 (Remarks to the Author):

The authors have addressed most of my comments. A new schematic diagram is also added in new Fig. 5e. The authors explained the growth conditions and the relationship between bilayer MoS₂ and TB-MoS₂ with a logic that “a higher yield of TB-MoS₂ can be obtained under a higher proportion of bilayer MoS₂”. If so, as far as I can see, confined space's role in breaking the thermodynamically unfavorable is not so strong. The roles and mechanisms of turbulent flow on growing twist structures still need to be explored (maybe next work). Nevertheless, the authors carefully demonstrated an efficient method and growth windows for TB-MoS₂, a step forward for the controlled growth of twisted bilayer 2D materials.

I recommend the publication of this manuscript in Nature Communications after addressing the following minor concerns.

The authors should give the corresponding expression on the logic of “a higher yield of TB-MoS₂ can be obtained under a higher proportion of bilayer MoS₂” in the manuscript, which would help readers understand the key point on how to get high proportion of TB-MoS₂. The authors should also provide a clearer expression on the denominator of the % of the TB-MoS₂ in the abstract, introduction, and main conclusions. Is the 17.2% to all the MoS₂ domains? or just to bilayer MoS₂?

Response to referees

Response to Reviewer 2

The authors have addressed most of my comments. A new schematic diagram is also added in new Fig. 5e. The authors explained the growth conditions and the relationship between bilayer MoS₂ and TB-MoS₂ with a logic that “a higher yield of TB-MoS₂ can be obtained under a higher proportion of bilayer MoS₂”. If so, as far as I can see, confined space's role in breaking the thermodynamically unfavorable is not so strong. The roles and mechanisms of turbulent flow on growing twist structures still need to be explored (maybe next work). Nevertheless, the authors carefully demonstrated an efficient method and growth windows for TB-MoS₂, a step forward for the controlled growth of twisted bilayer 2D materials.

I recommend the publication of this manuscript in Nature Communications after addressing the following minor concerns.

The authors should give the corresponding expression on the logic of “a higher yield of TB-MoS₂ can be obtained under a higher proportion of bilayer MoS₂” in the manuscript, which would help readers understand the key point on how to get high proportion of TB-MoS₂. The authors should also provide a clearer expression on the denominator of the % of the TB-MoS₂ in the abstract, introduction, and main conclusions. Is the 17.2% to all the MoS₂ domains? or just to bilayer MoS₂?

Response:

We sincerely appreciate the reviewer for carefully reading our manuscript and providing invaluable suggestions to further improve it.

In this work, the yield was defined by the proportion of TB-MoS₂ samples in bilayer MoS₂. Therefore, we have added the expression “a higher yield of TB-MoS₂ can be obtained under a higher proportion of bilayer MoS₂” and “the yield of TB-MoS₂ in bilayer MoS₂” in the revised manuscript.

Changes to the revised manuscript are shown below (In Page 2, 18, 20, 22,).

Moreover, the yield of TB-MoS₂ in bilayer MoS₂ and density of TB-MoS₂ are significantly improved to 17.2% and 28.9 pieces/mm² by tailoring gas flow rate and molar ratio of NaCl to

MoO₃.

As shown in Fig. 5c and Supplementary Fig. 22, with the increase in the molar ratio from 0 to 20, the density increases from 0 to 28.9 pieces/mm², while the yield in bilayer MoS₂ increases from 0 to 17.2%.

The high proportion of bilayer MoS₂ (77.6% and 60.9%) can be achieved under the gas flow of 30 sccm and 50 sccm, respectively. We can find that a higher yield of TB-MoS₂ can be obtained under a higher proportion of bilayer MoS₂.

Although the high yield of TB-MoS₂ in bilayer MoS₂ (17.2%) and high density (28.9 pieces/mm²) of TB-MoS₂ have been achieved, the twist angles are still random and unpredictable.

Again, we would like to thank all reviewers for the detailed reviews and constructive suggestions that help us improve the manuscript.